# Reproducibility study on how to find Spurious Correlations, Shortcut Learning, Clever Hans or Group-Distributional non-robustness and how to fix them

## Abstract

Deep Neural Networks (DNNs) are increasingly utilized in high-stakes domains like medical diagnostics and autonomous driving where model reliability is critical. However, the research landscape for ensuring this reliability is terminologically fractured across communities that pursue the same goal of ensuring models rely on causally relevant features rather than confounding signals. While frameworks such as distributionally robust optimization (DRO), invariant risk minimization (IRM), shortcut learning, simplicity bias, and the Clever Hans effect all address model failure due to spurious correlations, researchers typically only reference work within their own domains. This reproducibility study unifies these perspectives through a comparative analysis of correction methods under challenging constraints like limited data availability and severe subgroup imbalance. We evaluate recently proposed correction methods based on explainable artificial intelligence (XAI) techniques alongside popular non-XAI baselines using both synthetic and real-world datasets. Findings show that XAI-based methods generally outperform non-XAI approaches, with Counterfactual Knowledge Distillation (CFKD) proving most consistently effective at improving generalization. Our experiments also reveal that the practical application of many methods is hindered by a dependency on group labels, as manual annotation is often infeasible and automated tools like Spectral Relevance Analysis (SpRAy) struggle with complex features and severe imbalance. Furthermore, the scarcity of minority group samples in validation sets renders model selection and hyperparameter tuning unreliable, posing a significant obstacle to the deployment of robust and trustworthy models in safety-critical areas.

## 1 Introduction and Motivation

Over the past decade, machine learning has made tremendous progress in many different domains, with landmark results such as large performance jumps across computer vision, speech recognition, and natural language processing. Deep Neural Networks (DNNs), in particular, have achieved state-of-the-art performance in the fields of visual recognition (image classification, object detection, image segmentation) (He et al., 2016a; Liu et al., 2022; Tan & Le, 2021; Zhang et al., 2023; Cheng et al., 2022), speech recognition and synthesis (Gulati et al., 2020; Baevski et al., 2020; Kim et al., 2021), natural language processing (He et al., 2023; Costa-jussà et al., 2024), and playing games (Silver et al., 2016; Schrittwieser et al., 2020). These advances accelerated the adoption of machine learning models in practical systems across industry and science. Consequently, their deployment is no longer limited to low-risk consumer applications, but starts to expand into high-stakes, safety-critical domains, such as medical diagnostics (Tiu et al., 2022; Esteva et al., 2017), autonomous driving (Hu et al., 2023; Li et al., 2022), fault detection and traffic management in aviation (Al-Haddad et al., 2024; Pinto Neto et al., 2023; Du et al., 2025), risk and market modeling in finance (Kesharwani & Shukla, 2024; Cheng et al., 2025; Zhou et al., 2025), cybersecurity (Nazim et al., 2025; Ataa et al., 2024), and legal technology (Ansari et al., 2024; Medvedeva & Mcbride, 2023).

In these contexts, predictive accuracy alone is insufficient. Model reliability and accountability are equally critical, since reliance on spurious correlations, often referred to as the Clever Hans effect, can not only intro-

duce systematic bias, but also harm patients or compromise safety. This creates an urgent need for methods that detect and mitigate the effect of such spurious correlations to ensure that models base their predictions on causally relevant features rather than confounding signals. While a wide range of correction strategies has been proposed over the last years, each approach carries distinct assumptions, supervision requirements, and computational costs, which complicates the design of fair comparative evaluations. Furthermore, the research landscape is fractured by different terminology, e.g. distributionally robust optimization (DRO), invariant risk minimization (IRM), shortcut learning, simplicity bias, or Clever Hans. While all communities try to address the same fundamental goal of ensuring that models prioritize causally relevant features over confounding signals, researchers too often remain confined to their own field, predominantly referencing work within their own community. This reproducibility study aims to unify their perspectives by providing a comprehensive comparative analysis of correction methods under the realistic, challenging constraints often found in safety-critical domains.

Specifically, we contribute:

- A re-implementation and comparative evaluation of a diverse selection of correction strategies for imaging tasks, including recent XAI-based methods that have not yet been comprehensively tested independently.

- An evaluation design that mimics practical constraints in safety-critical applications, i.e. highly limited data availability and strongly pronounced spurious correlations, using both synthetic and real-world datasets with varying complexity of the causal and the confounding feature.

- A practical approach to subgroup labeling by applying Spectral Relevance Analysis (SpRAy), removing the common but unrealistic assumption of prior access to group labels.

- A discussion of advantages and possible pitfalls practitioners might come across during the implementation and application of the evaluated correction methods.

## 1.1 From identifying towards correcting Clever Hans

To raise trust and support experts in their decision-making, model behavior should be transparent to humans. For a long time, however, neural networks were considered *black boxes*, inherently difficult to interpret at the level required for operational assurance. This has motivated a large body of work in the field of XAI, which aims to unveil the decision strategies learned by models and make them easier to understand. Local XAI methods explain individual model decisions by revealing what input features or human-level concepts the model considered to be important in a specific input sample. Important representatives are attribution-based methods such as Layerwise Relevance Propagation (LRP) (Bach et al., 2015), Integrated Gradients (Sundararajan et al., 2017), or Concept Relevance Propagation (CRP) (Achtibat et al., 2023), which assign real-valued attribution scores to each input feature that indicate how strongly the feature contributed to the decision. These attributions can be visualized as heatmaps, quickly highlighting relevant regions in the input sample. Alternatively, counterfactual explainers (e.g. (Karimi et al., 2020; Rodríguez et al., 2021; Jeanneret et al., 2023; Weng et al., 2025; Bender et al., 2026)) construct a minimally changed counterfactual that would flip the model's decision. By comparing the counterfactual to the original sample, we can again uncover important features on which the model relies for its decision. In contrast to these local techniques, global XAI methods characterize model behavior at a more general level, not with respect to one individual decision. Important approaches include activation maximization (Erhan et al., 2009; Nguyen et al., 2016) and Relevance Maximization (Achtibat et al., 2023), which synthesize inputs that maximize neuron or class scores, thereby informing us about the model's idea of a prototypical example for a specific class or concept. Other noteworthy approaches are Concept Activation Vectors (CAVs) (Kim et al., 2018; Pahde et al., 2025) that quantify sensitivity to human-defined concepts, and related global analyses such as Network Dissection (Bau et al., 2017) that links internal units to semantic concepts. SpRAy (Lapuschkin et al., 2019) combines both local and global XAI by clustering attribution maps for a large amount of samples to reveal distinct decision strategies learned by a model.

XAI audits have repeatedly uncovered non-obvious failure modes in otherwise high-accuracy models. A central shortcoming is simplicity bias: when multiple predictive cues are available, DNNs tend to prefer simple

features that are easy to fit under empirical risk minimization (ERM). In images, they often rely on low-level signals such as background textures, color casts, watermarks, or other dataset-specific artifacts while ignoring more complex but semantically relevant features. Such models can achieve high accuracy by exploiting the simple features as shortcuts (*shortcut learning*) without actually learning a valid decision strategy for solving the underlying task. Hence, they will usually fail under distribution shifts, especially when the simple features are confounders lacking a causal link to the target and only correlating spuriously. Because validation and test splits often contain the same spurious correlations as the training data, high performance persists during model selection and evaluation with standard metrics such as average accuracy over all samples (referred to as *empirical accuracy* hereinafter), overestimating the model's ability to generalize. When the model is deployed to a new environment where the confounder changes distribution or disappears, performance can drop sharply, especially on minority subpopulations. This behavior is known as the Clever Hans effect, by analogy to the historical case where a horse appeared to answer arithmetic questions by exploiting signals unrelated to the task (Shah et al., 2020; Geirhos et al., 2020; Valle-Pérez et al., 2019; Yang et al., 2024; Beery et al., 2018; Taori et al., 2020; Johnson, 1911).

Such spurious correlations are common across popular datasets and tasks (Geirhos et al., 2020; Lapuschkin et al., 2019; Anders et al., 2022; Sagawa et al., 2020; Steinmann et al., 2024; Kirichenko et al., 2023; Scimeca et al., 2021). In image classification, popular examples include predicting *horse* because of a copyright tag in the corner (Lapuschkin et al., 2019), or *wolf* because of snow in the background (Ribeiro et al., 2016), rather than relying on object shape. Alarmingly, Clever Hans predictors also commonly occur in safety-critical medical tasks (Oakden-Rayner et al., 2020; Banerjee et al., 2023; Badgeley et al., 2019; Klauschen et al., 2024; Zech et al., 2018). For instance, a deep model for pneumonia detection on chest X-rays was found to base its predictions on hospital-specific markers in the images (which differed between hospitals and correlated with pneumonia prevalence) rather than on the actual lung pathology. Likewise, DNNs have "detected" cancer by picking up scanner- or lab-specific staining patterns associated with malignant samples, instead of the histological signs of cancer. Data limitations amplify these risks in medical imaging and other high-stakes areas: available training data are often scarce, costly to obtain, and acquired with varying protocols and equipment (Litjens et al., 2017; Kelly et al., 2019; Varoquaux & Cheplygina, 2022). Such constraints increase the chance that a single non-causal shortcut dominates learning, while minority subgroups (lacking that shortcut) are underrepresented, resulting in overfitting, domain shift, and evaluation pitfalls.

In these settings, recognizing and mitigating shortcut learning is crucial for building DNNs that generalize reliably under real-world distribution shifts. While methods for evaluating model behavior are by now well-established and commonly used, it is apparent that there is also a need for interventions that can prevent or correct Clever Hans models without requiring large, perfectly curated datasets. In this context, work on robust optimization in statistics and operations research has laid an early foundation (Kouvelis & Yu, 1997; Ben-Tal & Nemirovski, 1997; 1998; El Ghaoui & Lebret, 1997; El Ghaoui et al., 1998). Building on this, a large body of research has refined the distributionally robust optimization (DRO) framework (Duchi & Namkoong, 2021; Delage & Ye, 2010; Ben-Tal et al., 2013; Hashimoto et al., 2018; Duchi et al., 2021; Oren et al., 2019; Duchi & Namkoong, 2019; Shafieezadeh Abadeh et al., 2015; Zhang et al., 2020; Jin et al., 2021; Zhang et al., 2025). Rather than following the standard approach of optimizing the average loss via empirical risk minimization (ERM), DRO defines an uncertainty set around the empirical training distribution and minimizes the worst-case loss. A prominent instance, which is often considered to be state-of-the-art in the field of robust learning, is Group Distributionally Robust Optimization (Group DRO) (Sagawa et al., 2020; Hu et al., 2018). Group DRO optimizes the worst group loss by assuming access to subgroup annotations for training samples. Another approach is used by Deep Feature Reweighting (DFR) (Kirichenko et al., 2023): while similarly relying on group labels, it uses them to assemble a small balanced set for fine-tuning the final linear layer, thereby reducing reliance on confounding features.

Methods based on Invariant Risk Minimization (IRM) (Arjovsky et al., 2020; Wang et al., 2025; Lin et al., 2022; Liu et al., 2021b; Krueger et al., 2021; Zhou et al., 2022; Tan et al., 2023) modify the learning objective so that the model learns to only extract features that are stable across different environments, where an environment denotes a subset of data that shares a distinct data-generating process. By promoting invariance across environments, IRM aims to suppress features that occur only in specific subgroups, including

spuriously correlated confounders. Other approaches, such as Just Train Twice (JTT), first train a weaker auxiliary model, treat its wrongly classified samples as proxies for underrepresented groups, and then up-weight those samples when training the final predictor (Liu et al., 2021a; Nam et al., 2020; Yaghoobzadeh et al., 2019; Dagaev et al., 2023; Zhang et al., 2022).

A recent line of work tries to directly leverage results from XAI for correcting Clever Hans predictors. Weber et al. (Weber et al., 2023) offer a detailed overview as well as a qualitative comparison of XAI-based techniques to improve various model qualities beyond generalization capabilities. They also provide a framework to categorize different approaches depending on which step of the training loop they target, e.g. data augmentation or modification of the training objective. For instance, Right for the Right Reasons (RRR) (Ross et al., 2017) and similar methods (Shao et al., 2021; Rieger et al., 2020) adapt the loss function to penalize large attribution scores for features in regions of the input sample deemed irrelevant to the task, given sufficient domain knowledge. The Class Artifact Compensation (ClArC) family uses CAVs to capture the direction associated with a confounder. Augmentative ClArC (A-ClArC) and Projective ClArC (P-ClArC) then either add or suppress this component in each sample through projection (Anders et al., 2022), while Right Reason ClArC (RR-ClArC) adds a penalty to the objective that discourages model sensitivity towards the confounder (Dreyer et al., 2024). Finally, Counterfactual Knowledge Distillation (CFKD) (Bender et al., 2023) integrates a counterfactual explainer that systematically augments the training dataset with counterfactual examples in order to remove spurious correlations in the dataset.

This is just a small overview of the plethora of techniques that have been proposed in recent years under umbrella terms such as robust learning or domain generalization. Each approach carries distinct assumptions, supervision requirements, and computational costs, which complicates the design of fair comparative evaluations. Original method papers typically compare against a limited set of baselines and use different datasets, metrics, architectures, and training protocols. As a result, reported improvements are often sensitive to the chosen experimental setup. Independent studies have introduced standardized benchmarks to reduce this variability and to evaluate many methods side by side, often reporting mixed results and sometimes even finding that standard ERM can match or exceed the performance of current state-of-the-art mitigation methods (Gulrajani & Lopez-Paz, 2021; Koh et al., 2021; Yang et al., 2023; Yu et al., 2024; Qiao & Low, 2024). However, to our knowledge, these benchmark suites do not yet include CFKD and RR-ClArC, both of which are plausible contenders to reach or surpass the performance of currently popular approaches. A systematic and fair comparison of both XAI-based and non-XAI-based correction strategies under realistic data constraints is therefore needed to assess their effectiveness and guide their practical adoption.

## 1.2 Research scope

We aim to systematically re-evaluate and compare current state-of-the-art methods for mitigating Clever Hans behavior in image classification under standardized and fair test conditions. In contrast to prior studies, our comparative analysis is designed to approximate practical constraints in safety-critical applications such as computer-aided diagnosis. Specifically, we evaluate methods in a scenario where the training and validation data are highly limited and the spurious correlation between the confounder and the class labels is extremely strong. Importantly, this spurious correlation also persists in the validation split, with the consequence that minority subgroups are severely underrepresented during hyperparameter tuning. Previous evaluations, in particular those presented in original method papers (e.g. (Sagawa et al., 2020; Liu et al., 2021a)), often validate on splits with ample minority group coverage, either through manual curation or simply by using large datasets. Such setups (artificially) simplify hyperparameter tuning and overestimate the practical robustness of correction methods. By contrast, our design mirrors realistic deployments in domains where only scarce and biased data are available. This ensures that the comparative analysis not only measures theoretical potential but also provides insights into the practical feasibility of applying these correction methods in high-stakes domains.

We assess performance on both synthetic and real-world datasets that vary in the complexity of the causal and the confounding feature. For each dataset, we create two poisoned versions: in the first version, the confounder is symmetrically distributed across the target classes, and in the second version (visualized in Figure 1), the distribution is asymmetrical. The setting is described in detail in Section 3.

In addition, a lot of popular correction methods (Kirichenko et al., 2023; Sagawa et al., 2020; Anders et al., 2022; Pahde et al., 2025; Dreyer et al., 2024) require group labels for at least a subset of samples, which are usually just assumed to be known in advance. Because such annotations are rarely available in practice, each of these methods were evaluated twice: once with ground truth group labels and once with group labels obtained via SpRAy. This design tests robustness to a realistic degree of label noise and simultaneously assesses how well SpRAy can support data group discovery in a setting with extremely small minority groups. To our knowledge, apart from manual annotation, there is currently no broadly suitable alternative to SpRAy for this purpose, so the applicability of these methods in a practical scenario also depends on the quality of SpRAy results.

This work focuses on recently proposed XAI-based correction methods that have not yet been extensively evaluated, but seem to be promising candidates to achieve state-of-the-art performance. Accordingly, we include CFKD and RR-ClArC in our method selection. Due to its comparatively low computational costs, we further add P-ClArC, as it can give perspective to the trade-off between computational expenses and correction quality. Moreover, implementing RR-ClArC already provides most components required to implement P-ClArC, which limits additional engineering effort. As non-XAI baselines, we select Group DRO and DFR, both well-established methods in robust learning research. Although Group DRO is typically applied during initial model training, we employ it post hoc on a fixed biased predictor to ensure comparability across methods. While Group DRO is often considered to be a seminal contribution in the field, DFR is particularly attractive in practice, as it is fast and easy to implement.

Together, the methods in our selection represent a broad spectrum of strategies for mitigating Clever Hans behavior. They differ fundamentally in how they intervene in the learning process: by modifying the dataset to remove imbalance, either through sub-sampling (DFR) or data augmentation with counterfactuals (CFKD); by altering latent representations through projection to suppress confounder directions (P-ClArC); or by adapting the learning objective, either prioritizing minority-group performance (Group DRO) or penalizing sensitivity to confounding features (RR-ClArC). Figure 1 visualizes these conceptual differences and also provides a compact overview of our evaluation methodology.

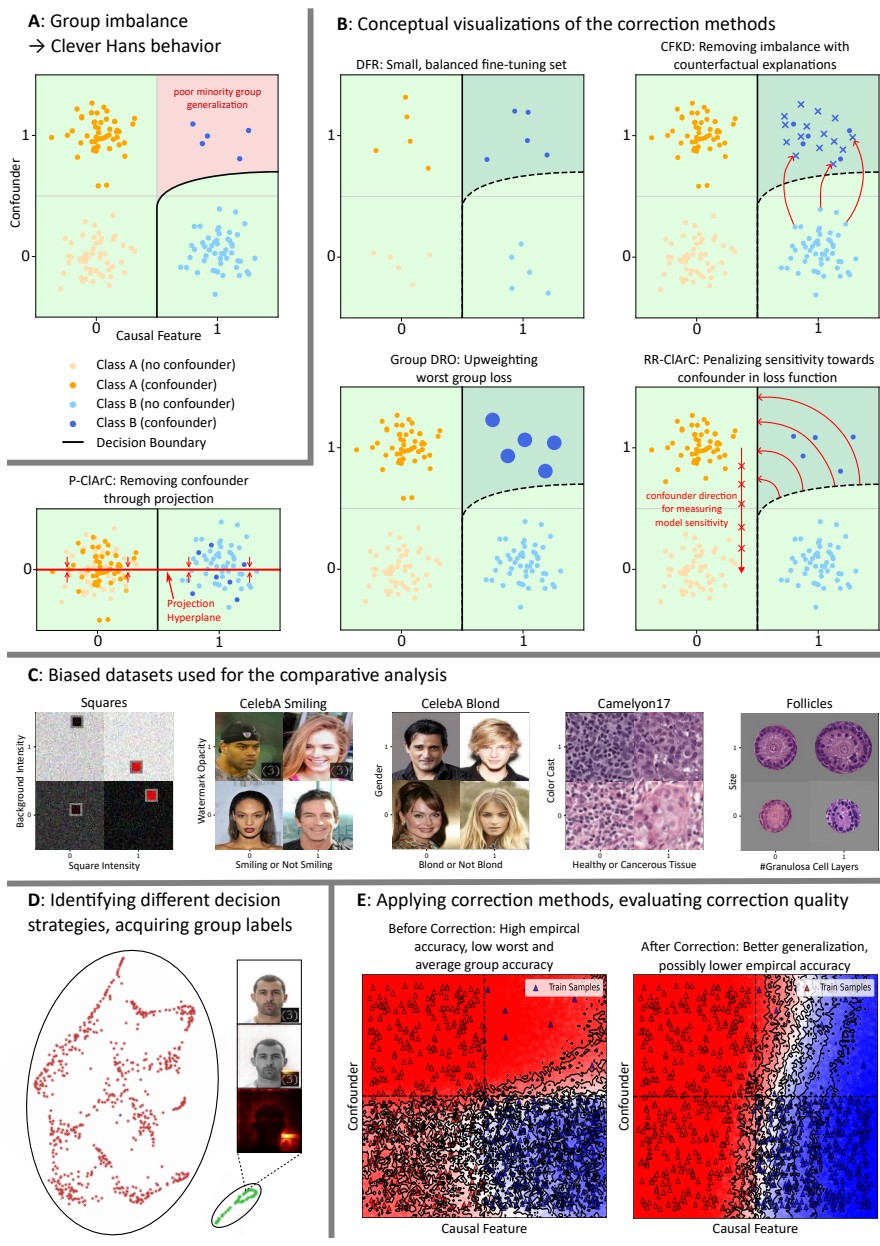

Figure 1: Research scope overview. **A**: Sketch illustrating how a confounder that is spuriously correlated with the class labels due to group imbalance can lead to Clever Hans behavior. Majority groups (green area) generalize well, while unseen minority group data (red area) is systematically misclassified. The few training samples in the minority group can easily be overfitted and hence do not prevent the classifier from learning a Clever Hans strategy. **B**: Conceptual visualization of the selected correction methods. **C**: Overview of the datasets used in our experiments, highlighting the distinction between the causal (x-axis) and the confounding (y-axis) feature. **D**: Identification of valid and Clever Hans decision strategies with the help of LRP and SpRAy; corresponding annotation of whole data point clusters with group labels. **E**: Application of the correction methods, followed by an evaluation and comparison of the correction quality. This specific example shows the decision boundary and confidence regions of the classifier trained on biased Squares before and after applying RR-ClArC. The ideal decision boundary would be a vertical line in the center, perfectly separating the samples by the value of the causal feature.

## 2 Method Review

Building on the method selection outlined in Section 1.2, this section provides a comprehensive overview of the approaches examined in this study. We first describe techniques for detecting spurious correlations that ML models may exploit, and then present correction strategies designed to prevent or mitigate the resulting Clever Hans behavior.

Since all correction methods except for CFKD used in our comparative analysis require group labels, we will provide a brief explanations how the terms *confounder label* and *group label* are used throughout this study: In addition to a class label $t_i \in C$ (with $C$ denoting the set of class indices), each sample $i$ is also characterized by a confounder label $q_i \in \{0, 1\}$. For confounders such as watermarks or other dataset artifacts, the confounder label indicates whether the respective artifact is present ($q = 1$) or absent ($q = 0$). For concepts like background color or object size, where there is no natural way of assigning the confounder labels, we arbitrarily assign to all samples with a specific color or size the label $q = 1$, and to all other samples $q = 0$. Although confounding features may be continuous, they are discretized for the experiments in this study so that the resulting confounder label is a binary variable. We will also only regard a single confounder at a time, even though it is possible for multiple confounders to be active simultaneously. The group label of a sample $i$ is defined as the tuple containing both its class and confounder label $(t_i, q_i)$. Accordingly, samples with an equal group label form a data group. Note that the original ClArC paper (Anders et al., 2022) uses the term *artifact label*, but we prefer the more general term *confounder label*, since the confounding features in our experiments are not necessarily artifacts.

### 2.1 Layer-Wise Relevance Propagation (LRP)

Layer-Wise Relevance Propagation (LRP) (Bach et al., 2015) is an attribution-based method that assigns each input feature, e.g. each pixel of an image, an attribution score (*relevance*) indicating its contribution to a model's output. The result can be visualized as a heatmap, highlighting the regions of the input that the model relied on most strongly. Such visualizations offer insights into the model's decision process and can reveal spurious correlations indicative of Clever Hans behavior, as can be seen in Figure 2.

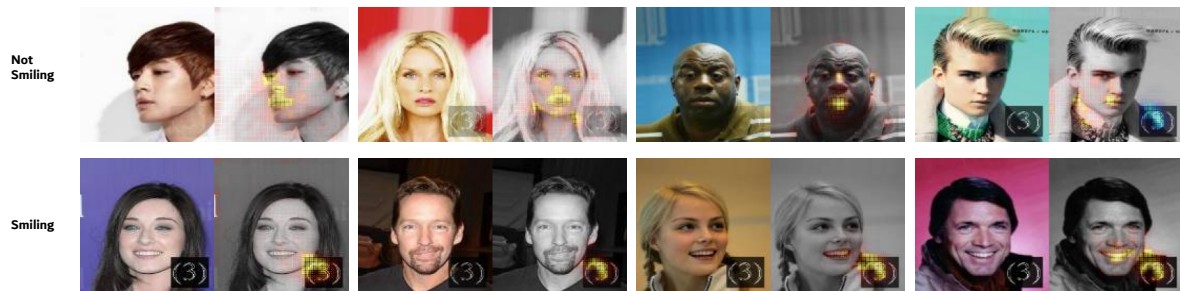

Figure 2: Selected samples from the CelebA dataset paired with their respective heatmaps as an overlay over the original image. The classifier was trained to distinguish between images of smiling and non-smiling persons. A watermark was added in the bottom right of each image, but the transparency of the watermark is correlated with the class labels. For images with the label *Smiling*, the watermark is bolder in most cases, and for images with the label *Not Smiling*, it is usually more transparent. Thus, the model learns to exploit the watermark as a Clever Hans feature, which can clearly be seen in the heatmap overlays: For the image on the very right in the *Not Smiling* row, we can see that the model correctly considered the mouth of the person to contribute to the *Not Smiling* class. However, the bold watermark actually had a negative contribution, possibly leading to a false classification. Conversely, for the *Smiling* class, a bold watermark acts as a strong positive indicator, sometimes leading the model to ignore the shape of the mouth completely, as e.g. in the first and second images in the bottom row.

Formally, LRP redistributes relevance from each subsequent layer of the neural network to its predecessor, starting at the model output $f(x)$ and ending at the individual pixels of input $x$. The relevance of the output

layer $L$ is defined as $R^L = f(x)$. Then, for a single neuron $j$ in the previous layer $L-1$, its relevance can be computed as

$$R_j^{L-1} = \frac{z_j}{\sum_{j'} z_{j'}} R^L \tag{1}$$

with $z_j$ being the neurons pre-activation (i.e. its activation is $a_j = \sigma(z_j)$ with activation function $\sigma$). As we can see, we normalize the relevance by dividing by the sum of the pre-activations of all neurons in $L-1$ that our output neuron is connected to. Analogous, the relevance a neuron $i$ in layer $L-2$ receives from a neuron $j$ in layer $L-1$ is:

$$R_{ij}^{L-2} = \frac{z_{ij}}{\sum_{i'} z_{i'j}} R_j^{L-1} \tag{2}$$

To compute the total relevance of neuron $i$, we have to sum up the relevances that are distributed from all the neurons in layer $L-1$ to neuron $i$:

$$R_i^{L-2} = \sum_j R_{ij}^{L-2} \tag{3}$$

This process is continued until we arrive at the input layer, where the relevances of the individual input neurons (i.e. pixels) can be visualized as a heatmap. An important property of LRP is that relevance is conserved between layers, so that:

$$f(x) = \sum_i R_i^{L-1} = \sum_i R_i^{L-2} = \ldots = \sum_i R_i^1 = \sum_i R(x_i) \tag{4}$$

This means the network decision is directly rewritten as a sum of easy-to-interpret attributions from each pixel (instead of writing $R_i^1$ for the relevance of pixel $i$, we can also write $R(x_i)$).

Conditional Relevance Propagation (CRP) also makes use of the same backpropagation, but additionally allows to condition the relevance not only on a specific class (e.g. *Smiling*), but also on human-interpretable concepts (such as *Mouth* or *Watermark*) by supplying a set of conditions $\theta = \{c_1, c_2, \ldots, c_n\}$ to the process ($R(x|\theta)$). These conditions can be thought of as rules that direct the flow of the backpropagation only through specific parts of the neural network. In practice, this is done by selecting one or more channels in a layer that we are interested in. All relevance is then only distributed through these channels. Especially in deeper layers, channels are assumed to encode for high-level, human-understandable concepts (see e.g. (Bau et al., 2020; Zhou et al., 2015)), so this procedure allows for the generation of heatmaps regarding these individual concepts. By repeating the process with alternating condition sets, we can systematically search for concepts important to the model's decision making. It is also possible to select only a certain output neuron in the case of multiclass classification, thereby receiving attribution values supporting or opposing a model's decision with respect to a particular class.

## 2.2 Spectral Relevance Analysis (SpRAy)

While LRP explanations can be helpful to discover undesirable decision strategies, examining heatmaps for individual predictions does not allow us to conclude that a model consistently avoids Clever Hans behavior. Explanations for some samples may appear valid even when the model relies on spurious correlations, because not all inputs contain the confounding feature. Without prior knowledge of the confounder's presence or distribution, a large number of LRP explanations would need to be examined manually until we can deduce with sufficient certainty that a model did not learn any invalid decision strategies.

SpRAy (Lapuschkin et al., 2019) addresses this limitation by combining local and global approaches. Instead of analyzing explanations sample by sample, SpRAy is able to process large amounts of attribution maps and arrange them into clusters by similarity. With sufficient domain knowledge, this allows us to identify

all distinct decision-making strategies learned by the model, including Clever Hans strategies. This procedure reveals at scale for which samples the model uses valid features and for which it relies on spurious confounders, enabling the assignment of confounder labels for all samples in a dataset. These labels are essential for correction techniques such as DFR, Group DRO, or ClArC methods. When ground-truth labels are unavailable, SpRAy allows us to avoid the extensive human labor that comes with manual annotation.

The first step of SpRAy is to generate relevance heatmaps $h_1, \ldots, h_n \in \mathbb{R}^d$ using LRP or CRP for all samples of interest. As these heatmaps indicate which input features are important for the model's classification decision, clusters of similar heatmaps intuitively represent the different decision strategies learned by the model. In order to identify these clusters, SpRAy uses Spectral Clustering, which is described in detail in (von Luxburg, 2007). In short, Spectral Clustering is performed by creating a similarity graph, where each heatmap is modeled as a vertex, and the similarity between two heatmaps is used as the weight of the edge connecting them. This graph can be represented by a weighted adjacency matrix $S$, with each entry $s_{ij}$ containing the pairwise similarity between heatmaps $i$ and $j$. There are different methods to construct the similarity graph. One possibility is to apply the (sparse) k-Nearest Neighbors algorithm to the heatmaps. If a heatmap is not among the k nearest neighbors of another heatmap, the respective entry is set to 0. Otherwise, we set it to their similarity score, which can e.g. be calculated with the gaussian kernel:

$$s_{ij} = \exp(-\frac{\|h_i - h_j\|^2}{2\sigma^2}) \tag{5}$$

The hyperparameter $\sigma$ can be used to modify the widths of the neighborhood. From the weighted adjacency matrix $S$, we compute the (normalized) graph Laplacian $L := I - D^{-1}S$, with $D$ denoting the degree matrix of the graph. The next step is to compute the first $k$ eigenvectors $u_1, \ldots, u_k \in \mathbb{R}^n$ (i.e. the eigenvectors corresponding to the first $k$ eigenvalues ordered by size, starting from the smallest). The eigenvectors form the columns of a matrix $U \in \mathbb{R}^{n \times k}$, whose rows $y_1, \ldots, y_n$ can be used as low-dimensional ($k << d$) representations for our heatmaps. Accordingly, $U$ constitutes the Spectral Embedding of the heatmaps. Now, we can apply conventional clustering algorithms like k-Means or DBSCAN to the representations, and optionally use further embedding methods such as t-SNE to visualize and explore the distribution of the heatmaps to hopefully identify the decision strategies learned by our model. Figure 3 shows a t-SNE visualization of the Spectral Embedding using real data from a colored version of the MNIST dataset.

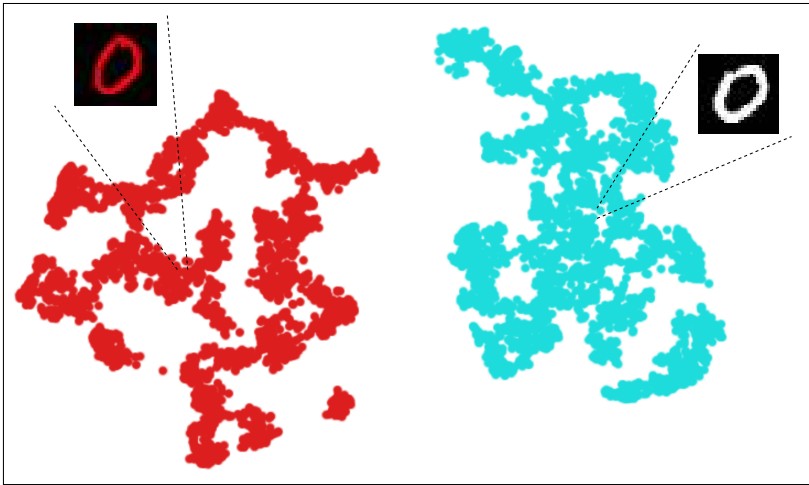

Figure 3: Result of a Spectral Relevance Analysis on Class '0' samples of the Colored MNIST dataset, where 50% of samples from Class '0' are colored red, while all other samples are colored white. The spectral embedding is visualized with the help of t-SNE. It is clearly visible that the samples are arranged in two distinct clusters, representing the two decision strategies learned by the model: classification by coloring (left side), which constitutes a Clever Hans effect, and classification by the shape of the depicted digit (right side), which is the correct, causal concept that we actually want our model to base its decision on.

Lapuschkin et al. (Lapuschkin et al., 2019) don't explicitly specify a reason why they apply Spectral Clustering to the relevance maps instead of applying other clustering techniques. One reason might be that Spectral Clustering is a well-established method that also works well in cases where the clusters don't have convex shapes, as opposed to e.g. directly using k-Means, and can also be used with large, complex datasets (von Luxburg, 2007).

In practice, it can be beneficial to downsize the relevance maps before applying SpRAy, because it can speed up the process and make it more robust (Lapuschkin et al., 2019). In addition, it is not necessary to use the relevance maps of the input layer. As high-level concepts usually are better captured and become more disentangled in the deeper layers of the model (i.e. captured by distinct channels (Bau et al., 2020; Zhou et al., 2015; Zeiler & Fergus, 2014; Anders et al., 2022)), using the attribution maps of these layers can produce better results. This is also another advantage over using LRP without SpRAy, because only attribution scores in input space (or for the first few shallow layers) can be visualized as heatmaps in a way that is meaningful to and interpretable by humans. This makes it hard to assess the importance of concepts that overlap in input space, especially (but not exclusively) if we have multiple non-local concepts such as background texture and color cast. By choosing an appropriate layer for computing attribution scores, SpRAy can still successfully cluster samples by the importance of the overlapping concepts, allowing us to identify distinct strategies learned by the model that would be hard or impossible to detect when manually examining heatmaps.

### 2.3 Concept Activation Vectors (CAVs)

Concept Activation Vectors (CAVs) were first introduced by Kim et al. (Kim et al., 2018) as a global XAI-tool that measures a model's response to a specific concept, i.e. it quantifies how important a concept $c$ is for reaching a particular classification. Given our trained model as a composite of its individual layers $f = f_L \circ \ldots \circ f_1$, we can also view the model as the composite of 1) a feature extractor $f_{\text{fe}}^{(l)} = f_l \circ \ldots \circ f_1$ that extracts informative, high-level features from the raw input features, i.e. it computes meaningful latent representations for the input samples, and 2) a downstream head $f_{\text{dh}}^{(l)} = f_L \circ \ldots \circ f_{l+1}$ which performs the actual classification task based on these latent representations, so that $f = f_{\text{dh}}^{(l)} \circ f_{\text{fe}}^{(l)}$. Then, the CAV of a concept is defined as the vector orthogonal to a linear decision boundary (i.e. a hyperplane) separating the representations $a_l = f_{\text{fe}}^{(l)}(x)$ (layer $l$ activations) of inputs containing the concept and the representations of inputs that do not. Layer $l$ at which the model is "split" will usually be one of the deeper layers of the model, e.g. the penultima layer, because as already mentioned in section 2.2, deeper layers often extract and disentangle the high-level concepts we are interested in. However, the exact choice highly depends on the nature of the concept and the model architecture, and some concepts may be better represented in the activations of shallower layers (see e.g. (Kim et al., 2018; Anders et al., 2022)).

In our case, the concept we are interested in is the respective confounder. Given that each sample is annotated with a confounder label indicating the presence of a Clever Hans feature, computing a CAV for that Clever Hans feature is simple: We train a linear classifier, e.g. a linear SVM, to differentiate between the representations $a_l$ of Clever Hans samples and clean samples by using their respective confounder labels as the classification targets. The classifier weights then constitute a CAV for the Clever Hans feature: $v_{\text{svm}}^{c,l} = w_{\text{svm}}$. Since we want the CAV to only be aligned with the confounding concept's direction and, accordingly, be orthogonal to the directions of all other causal concepts, it is usually necessary that all samples used to train the linear classifier are from the same class. If both the Clever Hans samples and the clean samples are sufficiently representative for the general data distribution in the chosen target class, it can be assumed that the causal concepts do not influence the decision boundary of the linear classifier and are therefore not captured by the CAV. Now, we can compute the *conceptual sensitivity* $S_{c,l}$ with

$$S_{c,l} = \nabla_{a_l}[f_{\text{dh}}^{(l)}(a_l)] \cdot v_{c,l} \tag{6}$$

to get a measurement of how sensitive the model is towards the concept of interest.

However, according to Pahde et al. (Pahde et al., 2025), using the weight vector of a linear classifier as a CAV might make it susceptible to noise, potentially leading to poor alignment with the direction of the

concept in latent space. Instead, they propose a different kind of CAV called Pattern Concept Activation Vector (PCAV). An estimation of the PCAV for concept $c$ at layer $l$ is given by

$$v_{\text{pat}}^{c,l} = \frac{\text{cov}[a_l, q]}{\sigma_q^2}, \tag{7}$$

i.e. the (empirical) covariance between latent representations $a_l$ and their confounder labels $q$ divided by the (empirical) variance of $q$. Pahde et al. experimentally show that PCAVs better capture the confounder's direction in latent space in the presence of noise compared to CAVs acquired by training a linear classifier (Pahde et al., 2025). Again, we only want to consider activations of samples from the same target class. Otherwise, since the confounder labels are also correlated with the actual causal concepts, the PCAV would not be aligned with the confounding concept's direction in latent space, but with the direction of a superposition of confounding and causal concepts.

Alternatively to directly using the activations $a_l$ as latent representations of $x$ when computing a (P)CAV, Dreyer et al. also try to first either perform max-pooling or take the mean over the spatial dimensions, i.e. the width $w$ and the height $h$, of the activations (Dreyer et al., 2024). Given that layer $l$ has $k$ channels, $a_l$ is in $\mathbb{R}^{m \times w \times h}$. After max-pooling or taking the mean, the new dimensionality would simply be $\mathbb{R}^m$, so we end up with a single score per channel. There is no explicit reason stated for this in (Dreyer et al., 2024), but presumably it was done as a way to reduce noise in the representations. Also, when assuming that each channel carries information about one specific concept, the new representation can be seen as a set of indicator values for each individual concept. In addition, reducing the dimensionality can reduce memory requirements and computation times. Whether max-pooling or averaging should be performed will be determined in our experiments by a hyperparameter that will be referred to as the *CAV mode*.

## 2.4 Class Artifact Compensation (ClArC)

Class Artifact Compensation (ClArC) refers to a family of correction methods developed to mitigate Clever Hans behavior in ML models. First, Anders et al. introduced Augmentative ClArC (A-ClArC) and Projective ClArC (P-ClArC) (Anders et al., 2022), which were later further refined by Pahde et al. (Pahde et al., 2025). Subsequently, Dreyer et al. introduced Right Reason ClArC (RR-ClArC) (Dreyer et al., 2024), which generally achieves superior performance compared with A-ClArC and P-ClArC. The most recent variant, Reactive ClArC (R-ClArC) (Bareeva et al., 2024), further advances this approach but appeared too recently to be included in this study. All ClArC variants are applied post-hoc to an already trained predictor rather than modifying the training process itself.

ClArC requires access to confounder labels $q \in \{0, 1\}$ for a representative subset of samples. When ground-truth labels are unavailable, they can be inferred automatically using SpRAy. Based on these labels, we choose a target class $y$ and partition the data into two disjoint subsets, $X^- = \{x_i \mid q_i = 0 \wedge t_i = y\}$ and $X^+ = \{x_i \mid q_i = 1 \wedge t_i = y\}$. A CAV $v_c$ is then computed to capture the direction associated with the confounding concept $c$ in latent space. As discussed in section 2.3, PCAVs are generally preferred over weight vectors obtained from a linear classifier because they are more reliable under noisy conditions. From this step onward, the procedure diverges between the different ClArC variants.

**P-ClArC.** For P-ClArC, we construct a *suppressive artifact model* that removes the confounding component from each sample representation. The model is defined as

$$h_{\text{sup}}(x) = (I - v_c v_c^T)x + v_c v_c^T z_{\text{sup}} \tag{8}$$

where $z_{\text{sup}}$ represents the *non-artifact reference point* computed as the mean of all non-confounder samples,

$$z_{\text{sup}} = \frac{1}{|X^-|} \sum_{x^- \in X^-} x^- \tag{9}$$

As already established, it can often improve the quality of the CAV when it is derived from activations acting as latent representations $a_l = f_{\text{fe}}^{(l)}(x)$ instead of the raw inputs $x$. In that case, the suppressive model operates on the feature space of layer $l$: $h_{\text{sup}}(a_l)$.

Figure 4b shows a sketch of the original setting for P-ClArC as described in (Anders et al., 2022): The task is a typical classification problem, but one class contains a spurious artifact, such as a watermark or rounded image corners, that never appears in the other class(es). To correct the resulting Clever Hans behavior, the suppressive model $h_{\text{sup}}$ uses the CAV to project all data points onto a hyperplane orthogonal to the direction related to the confounding feature. This projection removes the confounder from each representation, preventing the classifier from exploiting it as a shortcut. The hyperplane is anchored at reference point $z_{\text{sup}}$.

The suppressive model, and thereby the projection, is integrated as an additional layer within the model architecture. When defined on the inputs $x$ directly, we place it before the first model layer: $f_{\text{corrected}} = f \circ h_{\text{sup}}$. When instead defined on the activations of layer $l$, the suppressive model has to be inserted between layer $l$ and layer $l+1$: $f_{\text{corrected}} = f_{\text{dh}}^{(l)} \circ h_{\text{sup}} \circ f_{\text{fe}}^{(l)}$. After the projection, all representations approximate non-confounder samples, effectively neutralizing the bias in the model's decision function. The intention behind defining reference point $z_{\text{sup}}$ as the mean of all non-confounder samples is that they should ideally remain (nearly) unchanged by the projection.

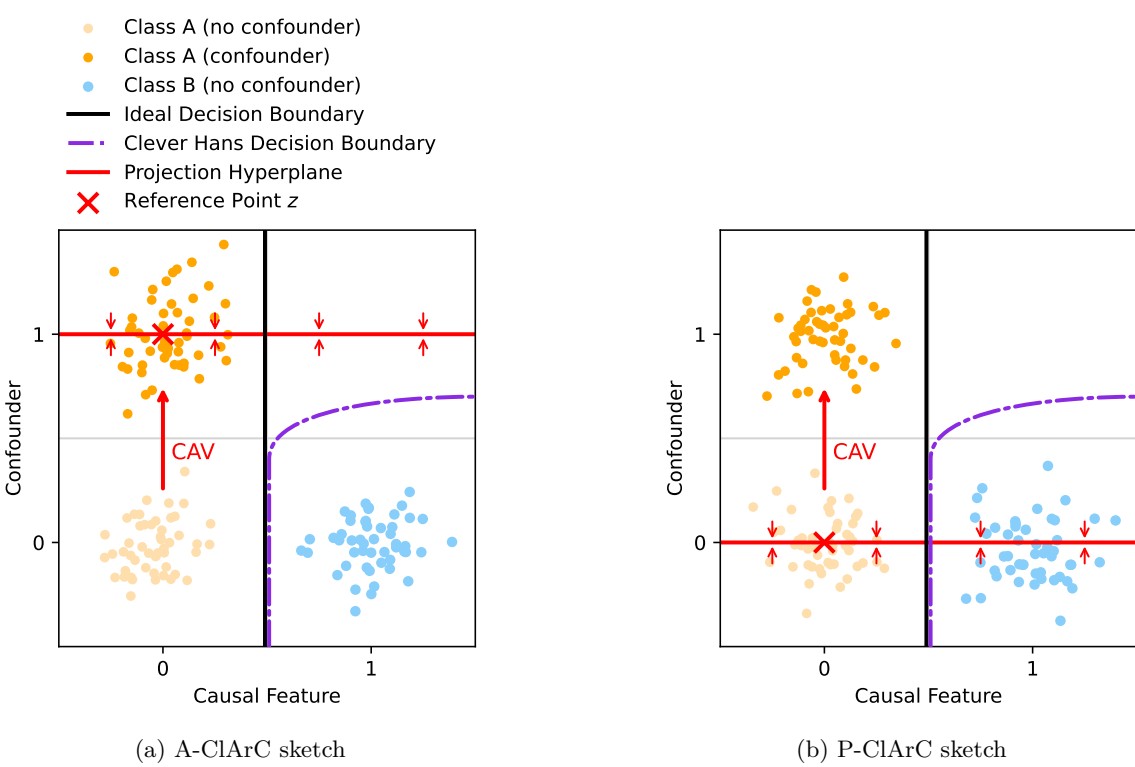

(a) A-ClArC sketch                                 (b) P-ClArC sketch

Figure 4: Sketch illustrating how A-ClArC and P-ClArC use a CAV to (a) add the confounder to or (b) remove the confounder from all samples. During training, there are no examples from class B containing the confounder available, so the model strongly associates the confounder with class A, resulting in a Clever Hans decision boundary. During inference, if there are new data points from class B that do contain the confounder, they will get misclassified. To fix this Clever Hans behavior, both A-ClArC and P-ClArC project all data points onto a hyperplane that is orthogonal to the direction associated with the confounder (captured by the CAV), so that it can no longer be used by the model as a discriminative feature between the two classes. However, in this setting, P-ClArC comes with the advantage that no additional fine-tuning is necessary, because for data points on the hyperplane, the true decision boundary and the confounded decision boundary already result in the same classification. A-ClArC requires fine-tuning, because otherwise, all projected data points from class B would get misclassified, since the true decision boundary and the Clever Hans decision boundary deviate for class B samples containing the confounder.

In this particular setting, additional fine-tuning is not required, because the ideal and the Clever Hans decision boundaries produce identical predictions for non-confounder samples (cf. Figure 4b). Whether fine-tuning is necessary depends on the specific shape of the learned decision boundary, and hence on the distribution of the confounder in the dataset. When fine-tuning, only the layers following the projection should be updated (i.e. only fine-tune $f_{\text{dh}}^{(l)}$). The earlier layers must remain frozen. Otherwise, the network may learn a new representation in which the direction of the confounder is no longer aligned with the computed Concept Activation Vector, thereby undoing the suppression effect.

A notable limitation of this approach is that it projects all samples, regardless of whether they contain the confounding feature. Preserving performance on clean samples therefore requires that causal and confounding features are approximately orthogonal in latent space. This assumption may not hold in practice, especially when the dataset is small or the feature representations are highly entangled. To prevent this, choosing the correct projection layer $l$ is crucial, but finding the optimal value might require extensive hyperparameter tuning. Moreover, a concept that acts as a confounder for one class can be part of a valid decision strategy for another class. Because P-ClArC applies a single projection across all classes, it cannot make this distinction, which may degrade overall classification accuracy.

**A-ClArC.** A-ClArC is conceptually very similar to P-ClArC. The main difference lies in the direction of projection: instead of removing the confounding feature, A-ClArC projects it onto all samples. The method defines an *inductive artifact model*:

$$h_{\text{ind}}(x) = (I - v_c v_c^T)x + v_c v_c^T z_{\text{ind}} \tag{10}$$

with the *artifact reference point* $z_{\text{ind}}$ as the mean of the confounder samples

$$z_{\text{ind}} = \frac{1}{|X^+|} \sum_{x^+ \in X^+} x^+ \tag{11}$$

Mathematically, A-ClArC differs from P-ClArC only in this choice of reference point $z$. The general idea remains the same: after projection, the classifier can no longer discriminate between confounder and non-confounder samples because the confounding component has been equalized across all representations. Consequently, the confounder can no longer be used as a shortcut feature. However, since the model originally associates the presence of the confounder with a specific class, all projected samples tend to be classified as that class (see Figure 4a). Fine-tuning the model layers following the projection layer is therefore necessary, so the network learns to rely on causal features, while the confounding signal (ideally) remains constant across all inputs.

Because A-ClArC is conceptually almost identical to P-ClArC, we restricted our comparative analysis to P-ClArC. In addition, for the datasets we use in our experiments (described in Section 3.1), there is not always a natural way to allocate confounder labels $q = 0$ and $q = 1$. For example, we use confounding concepts like *gender* or different types of color casts. In such cases, the differentiation between A-ClArC and P-ClArC becomes blurry, as we arbitrarily decide for one of the two possible confounder values (e.g. *male*) to be considered non-confounders and the other value (e.g. *female*) as confounders for computing the CAV and reference point $z$.

**RR-ClArC.** RR-ClArC adopts a different strategy from A-ClArC and P-ClArC by modifying the objective function rather than the model's architecture. As before, we compute a CAV $v_{c,l}$ that captures the direction of confounder $c$ using representations $a_l = f_{\text{fe}}^{(l)}(x)$. But instead of applying a fixed projection to these representations, RR-ClArC fine-tunes the model with an additional *Right-Reason loss term* $L_{\text{RR}}$ that penalizes sensitivity to the confounder. The total loss becomes

$$L = L_{\text{CR}} + \lambda L_{\text{RR}} \tag{12}$$

where $L_{\text{CR}}$ denotes the standard cross-entropy loss and $L_{\text{RR}}$ enforces insensitivity to the Clever Hans feature through

$$L_{\text{RR}} = \left( \nabla_{a_l} \left[ m \cdot f_{\text{dh}}^{(l)}(a_l) \right] \cdot v_{c,l} \right)^2. \tag{13}$$

This new loss term strongly resembles the square of the *conceptual sensitivity* (Equation 6) used to measure how strongly the model reacts to a certain concept. The only difference is that RR-ClArC adds a regularization parameter $m \in \mathbb{R}^{|C|}$, which is a class-specific weighting vector that also enables selective correction: setting entries at class-specific indices to zero disables correction for these classes. Alternatively, we can randomly fill $m$ with values from $\{-1, 1\}$, which Dreyer et al. have found to improve regularization (Dreyer et al., 2024). The regularization strength $\lambda$ acts as a weighting factor for the Right-Reason loss and controls the trade-off between prediction accuracy and confounder suppression during training.

Alternative formulations replace the squared dot product for computing the Right-Reason loss with the cosine similarity or the L1 norm, or substitute the logits with the sum of log-probabilities as the gradient target. The choice of loss variant is treated as a hyperparameter in our experiments.

During fine-tuning, minimizing the combined loss $L$ reduces the model's gradient alignment with the CAV. This process forces the model to become less sensitive to the confounder and to base its predictions on causal features instead. In contrast to P-ClArC, fine-tuning is thus not an optional step for RR-ClArC. Nevertheless, RR-ClArC brings the advantage of class-specific unlearning, providing more flexibility in multi-class scenarios, while P-ClArC and A-ClArC suppress the spurious correlation across all classes equally. Empirically, Dreyer et al. report that RR-ClArC, in most cases, outperforms both A-ClArC and P-ClArC, mitigating Clever Hans behavior the strongest of the examined ClArC methods (Dreyer et al., 2024). However, the correction quality for all ClArC methods critically depends on the accuracy of the computed CAV. If $v_{c,l}$ does not correctly represent the confounder's direction, the projection might not sufficiently suppress the confounder, or may even suppress important causal information.

An important distinction between our experimental setups and those used in the original ClArC papers concerns the type and distribution of confounders. While (Anders et al., 2022) and (Dreyer et al., 2024) primarily investigate discrete, artifact-like confounders that are either present or absent, we generalize this setting and additionally regard confounders that can take on any value on a continuous range.

Another difference lies in the confounder distribution within the training and validation data. In the original setups for P-ClArC and RR-ClArC, the confounder appears only in one class, so that one data group is completely empty (as illustrated in Figure 4). In contrast, our datasets intentionally include both confounder and non-confounder samples for each class, as Group DRO and DFR require that sufficient instances from all data groups are available. Additionally, Dreyer et al. fine-tuned their models for a fixed number of ten epochs (Dreyer et al., 2024) when applying P-ClArC and RR-ClArC, whereas we determined the optimal number of epochs dynamically by monitoring the average group accuracy on the validation split. This strategy also required that every data group is represented by at least a few samples. More information about our specific implementations of the correction methods is provided in section 3.2.

## 2.5 Counterfactual Explanations

Counterfactual Explainers offer an alternative method for investigating model behavior to feature attribution techniques such as LRP. Instead of assigning each input feature of a sample a score that indicates how much it contributed to the model's output, Counterfactual Explainers aim to alter the sample in a minimal, but semantically meaningful way so that the model changes its classification decision. By comparing the original sample with its counterfactual, we learn what features the model considers to be important for the classification.

Should the altered feature have a causal connection to the source class and target class, this can be an indicator that the model follows a valid decision strategy. If instead the altered feature is non-causal, this might indicate the model learned to shortcut the decision process due to a spurious correlation in the training data. An example for both cases can be found in Figure 5. In this study, we will only focus on so-called Visual Counterfactual Explainers for image classification (e.g. (Jeanneret et al., 2022; 2023; Bender et al., 2026; Bender & Morik, 2026; Zeid & Bender, 2026)), even though it is also possible to create counterfactuals in other domains, e.g. for tabular data Mothilal et al. (2020), natural language Sarkar et al. (2024), graphs Chen et al. (2023); Bechtoldt & Bender (2026) or proteins Kłos et al. (2026).

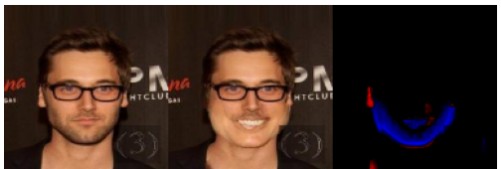
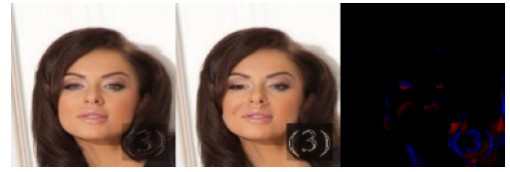

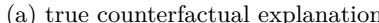

(a) true counterfactual explanation          (b) false counterfactual explanation

Figure 5: Counterfactual Explanations generated for source class *Not Smiling* and target class *Smiling*. From left to right, each panel shows the original image, its counterfactual, and the per-pixel difference between the original and the counterfactual. (a) shows a *true* counterfactual explanation, i.e. the Counterfactual Explainer changed the causal feature that distinguishes the source and target class by altering the mouth area in such a way that the depicted person is now actually smiling. (b) shows a *false* counterfactual, as the causal feature remains unchanged and instead the opacity of the watermark is increased, hinting at the model possibly being affected by Clever Hans.

Generating counterfactual explanations is not a trivial task, as there are multiple, sometimes contradictory, qualities that a good counterfactual explanation has to fulfill. First, changes to the original sample should be small, i.e. the counterfactual should be located close to the opposite side of the decision boundary, because we want to ensure that the changes mainly affect features important to the decision strategy. Simultaneously, the counterfactual should lie on the same data manifold $\mathcal{M}$ from which the original sample originated. Model behavior, in particular behavior of deep Neural Networks, for out-of-distribution data points is highly unpredictable, so if this requirement is not fulfilled, we would instead create an adversarial example. Compared to the original image, an adversarial example contains a minimal perturbation which is often barely visible and can resemble random noise patterns. While it also changes the model's decision, it does not provide any information about the concepts on which the model relies (Szegedy et al., 2014; Nguyen et al., 2015).

In general, changes to the original image must also not be too subtle, i.e. they need to be noticeable and understandable for human examiners. In addition, we also want there to be diversity when creating counterfactuals: for a single sample, we want to be able to create a set of counterfactual explanations that together reveal all relevant features and decision strategies. To make the counterfactual explanations more easily interpretable, they should also be sparse: ideally, only a single concept should be modified per generated counterfactual.

Since it is hard to fulfill all those requirements, as there are trade-offs between them, there has been a lot of research on the topic, leading to the invention of multiple different approaches to counterfactual creation, each focusing on different qualities, e.g. Diffeomorphic Counterfactuals (DiffeoCF) (Dombrowski et al., 2024), Diverse Valuable Explanations (DiVE) (Rodríguez et al., 2021), LatentShift Cohen et al. (2023), Adversarial Visual Counterfactual Explanations (ACE) (Jeanneret et al., 2023), Diffusion Visual Counterfactual Explanations (DVCE) (Augustin et al., 2022), DiME (Jeanneret et al., 2022), FastDiME (Weng et al., 2025), Global Counterfactual Directions (GCD) (Sobieski & Biecek, 2025), CDCT Varshney et al. (2024), DiffAE-CF Ha & Bender (2025), LeapFactual Cao et al. (2025). The method we will concentrate on in this study, and that will also be used for our CFKD experiments, is called Smooth Counterfactual Explorer (SCE) (Bender et al., 2026). SCE tries to address all described qualities by combining and improving traits from existing methods with some own additions. Like other methods, SCE iteratively adapts the original sample $x$ by Gradient Descent, propagating the Gradients through the decoder part of a Denoising Diffusion Probabilistic Model (DDPM) (Ho et al., 2020) for a better alignment with the data manifold $\mathcal{M}$. However, unlike the other methods, SCE does not compute the gradients using the original model $\nabla f$, but instead distills the original model into a surrogate model with smoother gradients $\nabla \hat{f}$. The motivation for this is that adversarial examples can also exist within $\mathcal{M}$ as local generalization errors (Stutz et al., 2019), and, according to Bender et al., smoothing the classifier's gradient field helps to prevent getting stuck in these local minima during the creation of counterfactual explanations. SCE also includes a diversifying mechanism that, when creating multiple counterfactuals, "locks" directions already used, so the following ones must find a

different route, i.e. they cannot again modify a feature that was already adapted in previous counterfactual explanations. Lastly, SCE uses a sparsifier that focuses the explanation on an individual feature, i.e. it prevents multiple features from being adapted simultaneously in a single counterfactual.

### 2.6 Counterfactual Knowledge Distillation (CFKD)

Counterfactual Knowledge Distillation (CFKD) (Bender et al., 2023; 2025; Hackstein & Bender, 2025) follows a different paradigm than the ClArC methods described above. It corrects biased models through an iterative student-teacher framework by distilling knowledge from an unbiased teacher onto a biased student model with the help of counterfactual explanations.

As with the ClArC methods, we start with a biased student model that was trained on a dataset containing a spurious correlation between some confounding feature and the class labels. The teacher would in practice be a human domain expert, but can be substituted with an unbiased oracle model, making it easier to conduct large-scale experiments and reproduce results. Based on the student, we generate counterfactual explanations for each sample in the dataset using the SCE algorithm described above. Since the student is affected by Clever Hans, a portion of the counterfactual explanations will be *false*, i.e. the modified feature is a non-causal confounder. The teacher now evaluates all generated counterfactuals. If, according to the teacher, the counterfactual still belongs to the source class, we know it is a *false counterfactual* and it is assigned the source class label (i.e. same class label as the factual). If the teacher instead agrees that the counterfactual belongs to the target class, it is considered a *true counterfactual*. The identified false counterfactuals are then added to the original dataset, helping to reduce the bias in the dataset by re-balancing the sizes of the different data groups. This process is illustrated in Figure 6: Creating counterfactual explanation can be imagined as "pushing" a sample just across the model's decision boundary. Since the student's boundary is skewed by the Clever Hans effect, this process frequently generates false counterfactuals that land within the sparse, underrepresented minority groups, effectively increasing their population. True counterfactuals, i.e. counterfactuals that actually modify the causal feature, are discarded. Due to the shape of the Clever Hans decision boundary, they typically fall within already well-populated majority groups and would only increase the existing imbalance.

Using the augmented dataset, we can either choose to fine-tune the student or train a new model from scratch. As the different data group sizes are now (more) balanced, the spurious correlation between the

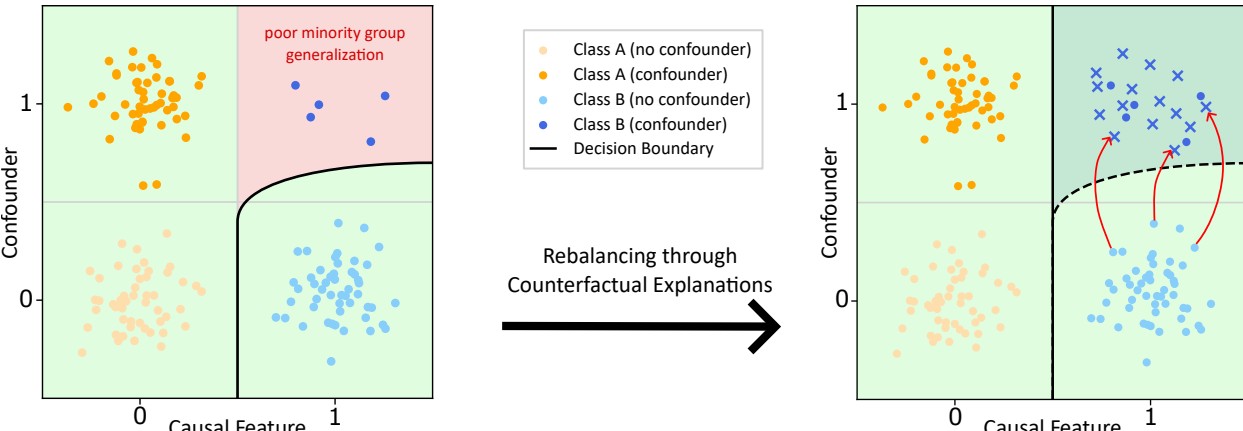

Figure 6: Conceptual illustration of dataset rebalancing using CFKD. (Left) An imbalanced training set leads to a biased Clever Hans decision boundary, where the model incorrectly relies on a spurious correlation between the confounder and class labels. (Right) CFKD generates 'false' counterfactuals (indicated by red arrows), for which only the value of the confounder is modified and the causal feature remains unchanged. By augmenting the dataset with these new samples, CFKD populates the sparse minority group, consequently weakening the spurious correlation. Fine-tuning on the augmented dataset hence allows the model to learn a more robust decision boundary.

confounder and the class labels no longer holds or is at least weakened, so the student has to learn to rely on causal features for classification. Should the model still display Clever Hans behavior, this process can be repeated for multiple iterations. However, getting a meaningful assessment for this is not trivial, as the validation data will generally come from the same, biased data distribution as the training data, so the empirical accuracy on the validation set is not a good estimator for the model's ability to generalize. In contrast to ClArC and the other methods we examine, CFKD does not assume access to group labels, so we also cannot compute the average group accuracy, which would be a more reliable estimator. As an alternative, Bender et al. propose a new metric called *feedback accuracy*, which is defined as the proportion of true counterfactual explanations among all counterfactual explanations created. The intuition is that, by incorporating the teacher's feedback in the form of counterfactual explanations labeled by the teacher into the dataset for the next iteration, the student's decision boundary will more and more resemble the decision boundary of the teacher. The teacher will therefore more often agree with the student that the class label actually changed when creating a counterfactual explanation, so the feedback accuracy increases, approaching a value of 1. This process can be viewed as the teacher transferring its knowledge to the student, hence the name Counterfactual Knowledge Distillation. Since we assume that the teacher (human expert or oracle model) is an unbiased classifier, aligning the student's decision boundary with the teacher results in the student unlearning the bias.

Bender et al. successfully apply CFKD for varying degrees of dataset poisoning, and show that, because CFKD is able to augment the dataset using false counterfactuals, it can even mitigate model bias when the correlation between the confounder and the class targets equals 1, i.e. when the minority groups are completely empty (Bender et al., 2023). All other correction methods we examine can not be applied in such a scenario, as Group DRO and DFR require samples from all data groups, and the ClArC methods need at least one class represented by both confounder and non-confounder samples for computing an accurate CAV.

If there are multiple confounding features present in the dataset, CFKD theoretically has no problems to correct for them simultaneously, as SCE creates a set of counterfactual explanations for a single sample, each modifying a different feature that the student thinks is relevant for classification. However, its application to multiclass classification remains challenging, as counterfactual explanations are typically generated for a single class transition at a time. Scaling to a multiclass setting would require generating multiple times more counterfactual explanations per sample, which could quickly become infeasible.

## 2.7 Deep Feature Reweighting (DFR)

Deep Feature Reweighting (DFR) (Kirichenko et al., 2023) is a correction method that, in contrast to the previously described methods, is not XAI-based. As for the other methods, the starting point is a student model trained with plain ERM on a dataset $D$ exhibiting a spurious correlation. Again, the student will be able to achieve high accuracy on the majority groups thanks to Clever Hans, but will fail for unseen minority group samples where the spurious correlation breaks. To correct the model with DFR, we create a new dataset $\hat{D}$, in which all data groups contain the same number of samples, so there no longer is a spurious correlation between the confounder and the class labels. Then, we fine-tune the last layer of the student model for a few epochs on $\hat{D}$, which, according to Kirichenko et al., is enough to remove or at least mitigate the bias learned by the model (Kirichenko et al., 2023). Similarly to the ClArC methods, DFR can be seen as splitting the model into a feature extractor, which extends from the input to the penultimate layer, and a downstream head, i.e. the last, fully connected layer acting as the classifier. The assumption is that the feature extractor still learns to extract the causal features when it is trained on the poisoned dataset $D$, but that the confounding feature is simply heavily upweighted in the downstream head, resulting in Clever Hans behavior. Fine-tuning the last layer on unbiased $\hat{D}$ therefore leads to a reweighting of the extracted features in favor of the causal features. Unlike the ClArC method, where layer $l$ after which to split the model is defined as a hyperparameter, DFR always splits the model after the penultimate layer.

To create the unbiased dataset $\hat{D}$, we need a sufficient amount of samples annotated with groups labels. Similarly to the ClArC methods, these either have to be known in prior or can be obtained via SpRAy. According to Kirichenko et al., a small number of samples for $\hat{D}$ is sufficient to noticeably reduce reliance on the confounder. In their own experiment, the size of $\hat{D}$ was a quarter of the size of the original training dataset

$D$, which was sufficient to yield similar results as state-of-the-arts methods like Group DRO. Preferably, $\hat{D}$ should not be a subset of $D$, as this is reported to reduce the improvements that can be achieved with DFR. Instead, one should construct a hold-out set from the data before training the student that is reserved for the DFR fine-tuning.

One possible drawback of DFR is that, since all data groups should have the same number of samples in $\hat{D}$, the size of the smallest minority group also dictates the size of all other groups. Depending on the total number of samples and the degree of poisoning, this number can be very small, especially in scenarios with very limited data availability and strong spurious correlations. As a consequence, $\hat{D}$ might only contain very few samples, independent of whether $\hat{D}$ is a subset of $D$ or a held-out set. Multiclass settings or the presence of multiple confounders might make this problem even more severe, because the dataset is fractured into even smaller groups. The few samples are possibly no longer representative of the general data distribution within their respective groups, so using them for fine-tuning the student might fail to improve generalization.

## 2.8 Group Distributionally Robust Optimization (DRO)

Group Distributionally Robust Optimization (Group DRO) (Hu et al., 2018; Sagawa et al., 2020) is a specialized, structured variant of Distributionally Robust Optimization (DRO) (Delage & Ye, 2010; Ben-Tal et al., 2013; Duchi et al., 2021). DRO itself is an optimization framework that acts as an alternative to standard ERM optimization: Instead of minimizing the average loss for the training data, DRO tries to minimize the worst-case loss over an uncertainty set of distributions. This uncertainty set should be representative for all possible data distributions that the model should be able to generalize to. This helps to prevent Clever Hans, as a spurious correlation that is present in the original training data will ideally not be present in all of the distributions of the uncertainty set. Since we optimize for the worst-case loss, the model is forced to learn a decision strategy that is valid for all the distributions, which should only be possible by relying on actual causal features.

There are different strategies for creating the uncertainty set, e.g. using $\phi$-divergences, such as the KL divergence, or the Wasserstein distance (Ben-Tal et al., 2013; Mohajerin Esfahani & Kuhn, 2018; Gao & Kleywegt, 2023). Instead, Group DRO assumes that group labels are available to us for both the training and validation data, and simply creates the uncertainty set by grouping the samples by their group labels. But in place of optimizing by only selecting samples from the group with the highest loss at each iteration step, we use an improved version of the training algorithm by Sagawa et al. (Sagawa et al., 2020) that updates the model for all groups at each step, but maintains a weighting factor $w_g$ per group $g$ used to scale the individual group losses, and which is larger for groups that had larger loss in the past. According to the authors, this increases stability and convergence, while still focusing the model's training on the group(s) for which it performs poorly.

Following the (Group) DRO optimization framework in theory ensures low training loss for the minority groups. However, this does not guarantee high test accuracy for these groups as well. Powerful models such as large DNNs are capable of simply overfitting minority group samples, so even when they mainly rely on Clever Hans features, they still achieve perfect (or nearly perfect) minority group performance on the training data. Focusing on the group with the highest loss during optimization therefore can still yield a similarly biased classifier as ERM that generalizes poorly under distribution shifts. Sagawa et al. address this issue by applying strong regularization (in particular weight decay) and early stopping, preventing the model from overfitting minority group samples for decreasing worst group loss during training, so that it instead has to rely on actually causal concepts (Sagawa et al., 2020).

Sagawa et al. also introduce an additional hyperparameter called model capacity $C$ (Sagawa et al., 2020). For each group size $n_g$, they compute $\frac{C}{\sqrt{n_g}}$ and add the result to the loss of group $g$ before calculating the overall loss and updating the model weights. This adds additional weight to the minority groups during optimization, giving even more priority to fitting them correctly (while preventing overfitting through regularization), as they usually generalize worse.

Using ground-truth group labels, Group-DRO is reported to significantly improve worst-group performance on unseen data compared to standard ERM training on biased datasets, both in imaging (Waterbirds (Sagawa

et al., 2020), CelebA (Liu et al., 2015)) and natural language processing (MultiNLI (Williams et al., 2018)) tasks. However, contrary to DFR and the ClArC methods, Group DRO requires group labels for all training and validation samples. Again, since group labels often are not available in practice, SpRAy or manual labeling will often be necessary to be able to use Group DRO.

## 3 Experimental Design

This section outlines the experimental setup that forms the basis of our comparative analysis. We begin by introducing the datasets used in the evaluation and emphasize the underlying data distributions regarding both the causal and confounding features and how they correlate with the classification targets. We then provide information about our implementation of each correction method, including hyperparameter tuning procedures and model selection criteria, to enable a transparent and reproducible comparison. Finally, we present the baseline student classifiers before correction and quantify the severity of their Clever Hans behavior.

### 3.1 Datasets

We evaluate the correction methods on five datasets: Squares, CelebA Smiling, CelebA Blond, Camelyon17, and Follicles. This selection includes both synthetic and real-world data and allows us to evaluate the correction methods for varying degrees of complexity regarding the true causal feature and the confounding feature. By including Camelyon17 and Follicles, we also incorporate possible practical scenarios in the medical field, a domain where robust generalization and reliance on exclusively true, causal features are especially important.

The task to be solved on all datasets is a binary classification problem, i.e. the ML model has to correctly predict the class label $t \in \{0, 1\}$ associated with an input sample. In conjunction with two possible values for the true confounder label $q \in \{0, 1\}$, there are consequently four different data groups within each dataset:

1. $A^- = \{x_i \mid t_i = 0 \wedge q_i = 0\}$ — non-confounder instances from class A

2. $A^+ = \{x_i \mid t_i = 0 \wedge q_i = 1\}$ — confounder instances from class A

3. $B^- = \{x_i \mid t_i = 1 \wedge q_i = 0\}$ — non-confounder instances from class B

4. $B^+ = \{x_i \mid t_i = 1 \wedge q_i = 1\}$ — confounder instances from class B

For all datasets except Follicles, we construct two poisoned variants, one with a symmetric and one with an asymmetric distribution of the confounding feature across the two classes. In the symmetric version, 98% of samples in class A will have the true confounder label $q = 0$, while only 2% of samples have $q = 1$. Conversely, 2% of samples in class B will have $q = 0$, and 98% will have $q = 1$. Hence, the respective relative group sizes are $[A^- : 49\%; \ A^+ : 1\%; \ B^- : 1\%; \ B^+ : 49\%]$, so the distribution of the confounding feature within class A is inversely symmetrical to the distribution in class B. Nevertheless, individually the true class and confounder labels are evenly distributed across the dataset, i.e. half of the samples belong to class A and the other half belongs to class B, and analogous, half the samples have the true confounding label $q = 0$ and the other half $q = 1$.

In the asymmetric version, class A consists of 50% samples with $q = 0$ and 50% samples with $q = 1$, while class B consists of 98% samples with $q = 0$ and 2% samples with $q = 1$. This leads to relative group sizes of $[A^- : 25\%; \ A^+ : 25\%; \ B^- : 49\%; \ B^+ : 1\%]$, making the distributions of the confounder in the two classes asymmetrical. In addition, while still half of the samples belong to class A and half of the samples belong to class B, the confounding feature is no longer evenly distributed across the dataset: Now, 74% of samples have the true confounder label $q = 0$, while only 26% of samples have $q = 1$.

Figure 7 shows a sketch of the data distributions of both the symmetric and the asymmetric dataset versions. Additionally, it depicts the theoretically optimal decision boundary as well as the Clever Hans decision boundary that a model will learn due to the spurious correlation of the confounder with the class labels. Note

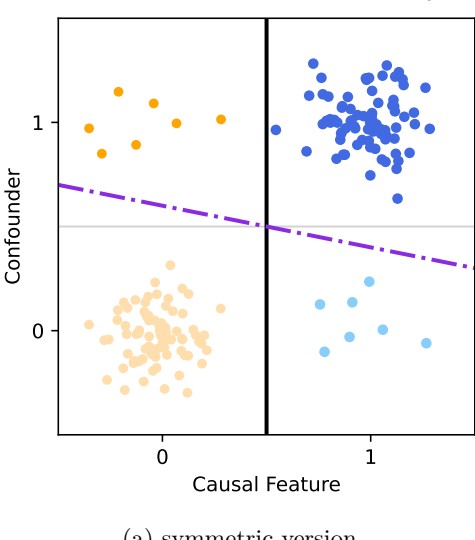

(a) symmetric version

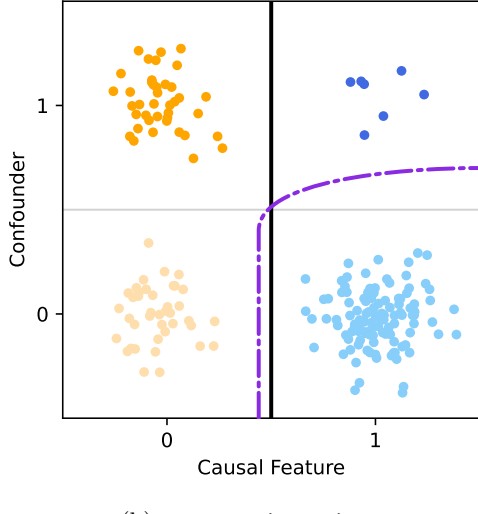

(b) asymmetric version

Figure 7: Generalized depictions of (a) the symmetric and (b) the asymmetric dataset distributions in the training and validation splits. The perfect decision boundary is a vertical line in the center, separating the two classes based on the value of the true, causal feature. Because of the simplicity bias, models trained on these datasets will prefer exploiting the confounder as a shortcut feature instead of basing their decision on the causal feature. Clever Hans "pushes" the decision boundary horizontally into the area of the minority group samples, thereby sacrificing accuracy on the minority groups in exchange for easily classifying majority group samples correctly. Due to their small number, minority group samples can be overfitted in order to achieve perfect training accuracy, with poor generalization for the minority groups as a consequence.

that the severity of the Clever Hans effect, and therefore the deviation between the learned decision boundary and the optimal solution, will vary for the different datasets. Datasets with more complex confounding features and simpler causal features likely lead to less biased models, as complexity makes the confounder harder to be exploited as a shortcut feature.

Both the symmetric and the asymmetric dataset versions are limited to only 1000 samples, of which 800 are allocated to the train split and 200 samples are allocated to the validation split. As a consequence, the minority groups will be represented by only 8 samples in the train split, and only 2 samples in the validation split. There are multiple reasons why we chose this specific setting for our experiments. First, we wanted to keep the number of samples small, because in practical scenarios (e.g. in the medical domain), where preventing Clever Hans behavior is especially relevant, training data often will also be limited. In addition, methods like DFR and Group DRO have already been shown to yield good results when a sufficient amount of samples annotated with true confounder labels is readily available (i.e. minority groups were still represented by multiple hundreds of instances in the training data (Kirichenko et al., 2023; Izmailov et al., 2022; Koh et al., 2021; Sagawa et al., 2020)). We think it is more interesting to evaluate their performance in more challenging scenarios, with small group sizes and a very strong spurious correlation between the confounder and the target label.

Compared to the training and validation splits, the test splits are much larger and contain a representative amount of minority group samples so that the reported results reflect the true generalization capabilities of the corrected models. The test splits of Squares, CelebA Smiling, and Camelyon17 are unpoisoned, i.e. all

data groups contain the same amount of samples, with the smallest being the Squares test split with a size of 1600 samples (400 per group). The test split for Camelyon17 contains 6800 samples (1700 per group) and the one for CelebA Smiling contains 19400 samples (4850 per group). For CelebA Blond, we use 20260 of the remaining images to create the test set, in which the minority group is represented by 165 samples.

As already mentioned, the only exception to the pattern described above is the Follicles dataset. The reason for this is that Follicles is a real-world medical dataset with a highly limited number of samples, so it is not possible to conduct further sub-sampling for reaching the same symmetric and asymmetric data distributions used with the other datasets. Instead, we will use the Follicles dataset as is. This way, we cover a real-world classification task containing a real spurious correlation that was not artificially created or strengthened by our setting. The Follicles data distribution and the sizes of the train, validation, and test split will be described in detail in section 3.1.5.

### 3.1.1 Squares

The *Squares* dataset is a synthetic dataset that consists of $64 \times 64$ images with only two important features: The *Foreground Intensity*, which refers to the brightness of the small, red square, and the *Background Intensity*, which refers to the brightness of the background. Both are continuous and can take on values in the interval $[0.0, 1.0]$. The x and y coordinates of the foreground square are determined uniformly at random. For solving the binary classification problem, the classifier must learn to distinguish between images of Class A with a dark foreground (values in $[0.0, 0.5)$) and images of Class B with a bright foreground (values in $[0.5, 1.0]$), meaning the foreground intensity is the causal feature that the classifier should rely on. In the poisoned versions of this dataset, the background intensity is correlated with the class labels, so it acts as a Clever Hans feature that can be exploited by the classifier. Samples with a background intensity $< 0.5$ are considered to have a dark background and are assigned the true confounder label $q = 1$. Conversely, samples with a background intensity $\geq 0.5$ have a bright background and are assigned the true confounder label $q = 0$. To make the task more challenging, mild gaussian noise is added on top of all images. Figure 8 provides randomly selected samples from the Squares dataset, illustrating the distinction between the four data groups.

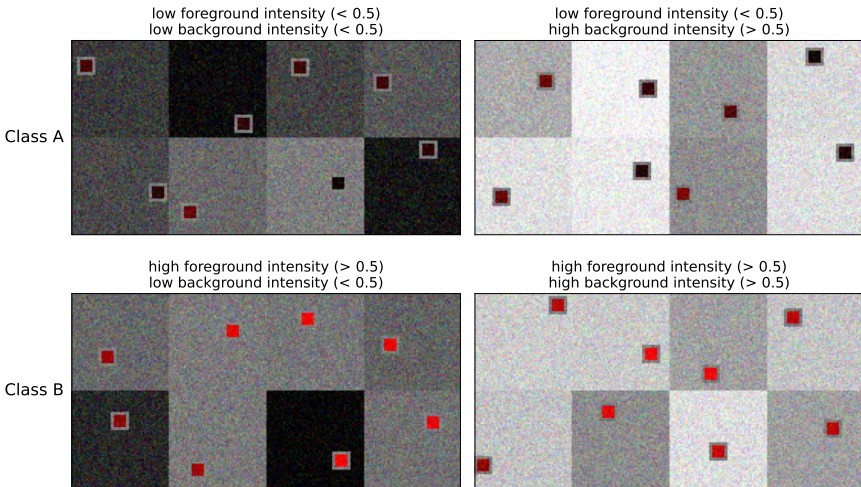

Figure 8: Randomly selected samples of the *Squares* dataset grouped by their respective foreground and background intensities. Class A contains samples with low foreground intensity ($[0.0, 0.5]$) and Class B contains samples with high foreground intensity ($[0.5, 1.0]$). In the poisoned versions of this dataset, the background intensity acts as a Clever Hans feature.

### 3.1.2 CelebA Smiling

The original CelebFaces Attributes (CelebA) dataset (Liu et al., 2015) comprises 202599 images of faces of celebrities, each annotated with 40 binary labels to indicate the presence of specific facial attributes of the depicted person. For our experiments, we only regard the *Smiling* label, i.e. we convert the task to a binary classification problem so that the classifier has to distinguish between the two classes *Not Smiling* (Class A) and *Smiling* (Class B). We also downsize the images to a uniform size of $128 \times 128$ pixels. In addition, we artificially add a watermark in the lower right corner of each sample. The confounder, i.e. the feature spuriously correlated to the target label, is the opaqueness of this watermark. Similar to the confounder for the Squares dataset, the opaqueness of the watermark is continuous and can take on values in the range $[0.0, 1.0]$, with 0.0 being completely transparent and 1.0 being completely opaque. Samples with a watermark opaqueness of $< 0.5$ are annotated with the true confounder label $q = 0$, and samples with a watermark opaqueness of $\geq 0.5$ will have confounder label $q = 1$. The four distinct data groups are visualized in Figure 9.

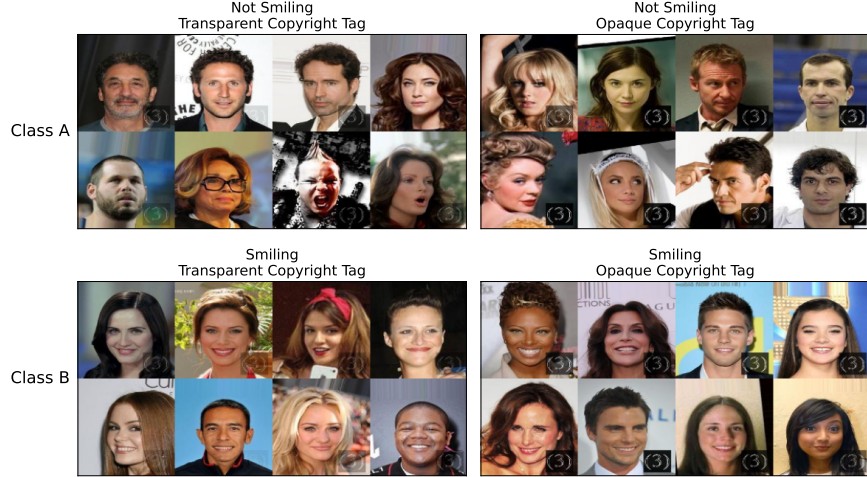

Figure 9: Randomly selected samples of the *CelebA* dataset, where we added a watermark of varying opacity to each sample in order to artificially introduce a confounding feature. The samples are grouped by the two target classes, *Smiling* and *Not Smiling*, and by the opacity of their watermark ($< 0.5$ as transparent and $\geq 0.5$ as opaque)

### 3.1.3 CelebA Blond

Like CelebA Smiling, CelebA Blond is based on the original CelebA dataset. Again, we downsize the images to $128 \times 128$ pixels and create a binary classification task by selecting the attribute label *Blond* as the classification target our model should learn to predict. Class A is comprised of images showing non-blond persons, and Class B contains the blond persons' images.

To introduce a spurious correlation for the model to exploit, we no longer artificially create a confounder, such as a watermark of varying opaqueness. Instead, we simply use another of the provided attribute labels: the *Male* label, with $q = 0$ indicating a female person and $q = 1$ indicating a male person. While the labels *Blond* and *Male* are already correlated in the original dataset, we still perform sub-sampling in a controlled way until the sizes of the data groups match the specifications described earlier. Randomly selected samples from each data group are shown in Figure 10.

Compared to the confounders in the other datasets, the spurious feature in CelebA Blond, i.e. the gender, is arguably the most complex. When deriving the gender from facial attributes, a multitude of simpler features, which individually are not necessarily associated with one specific gender, have to be examined and weighted. Among others, the simpler features include the presence of a beard, hairstyle, shape of the chin, shape of the eyebrows, and presence of makeup. This complexity presumably makes it harder for a

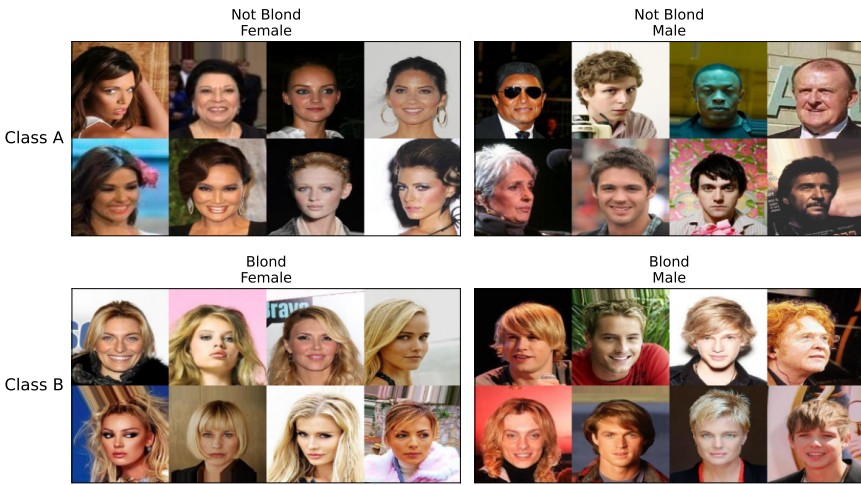

Figure 10: Randomly selected samples of the CelebA dataset, grouped by the two target classes *Not Blond* (Class A) and *Blond* (Class B), and by the confounding feature, which is the gender of the depicted person.

model trained on CelebA Blond to exploit *gender* as a shortcut feature, so we expect the student models to be affected less severely by Clever Hans. As we will see, it also negatively impacts the quality of SpRAy results, because the different decision-making strategies represented by LRP attribution maps cannot be as easily and unambiguously clustered. Further difficulty is introduced by the fact that the ground-truth attribute labels provided with CelebA are known to be inconsistent and even inaccurate to some degree (see e.g. Lingenfelter et al. (2022); Wu et al. (2023)), so our classifiers and the applied correction methods need to possess a certain robustness to noisy labels.

### 3.1.4 Camelyon17

Camelyon17 is a histopathological dataset that contains microscopic images of either healthy tissue (Class A) or tissue containing breast cancer metastases (Class B). Before conducting the experiments, we scale the images to sizes of $128 \times 128$ pixels. In the original dataset, there are four different possible hospitals from which the images can originate, but for our experiments, we only use the samples from two of the hospitals. Figure 11 shows that, depending on their hospital of origin, there is a clear distinction in the coloring of the images. This coloring is spuriously correlated with the presence of cancer metastases in the image, i.e. it is correlated with the target label, and therefore acts as a Clever Hans feature.

Note that this is a scenario that could reasonably occur when applying computer-aided diagnosis in practice. One could imagine that one of the hospitals mainly conducts preventive medical examinations, so most microscopic images from this hospital would depict healthy tissue. The other hospital could be specialized in the treatment of breast cancer and therefore have mostly patients who already tested positive for the disease. Tissue samples from this hospital would show signs of cancer metastases at a much higher rate. When images from both hospitals are used in the same dataset, and the hospitals do not produce the images under the same conditions, there is a risk that equipment-specific artifacts, such as e.g. a color cast, strongly correlate with the task-relevant histological signs of cancer. A model trained on this dataset will use this confounding feature as a shortcut to distinguish between healthy and cancerous tissue, while actually learning to distinguish between different lab equipment. This kind of model would probably fail in the event of a distribution shift, e.g. when confronted with images from hospitals not represented in the dataset. As the test split for assessing the model's generalization capabilities would likely come from the same distribution as the training and validation data, this Clever Hans effect might easily be overlooked, with possibly disastrous consequences for the patients. Similar behavior has been observed already in ML models that were trained to diagnose pneumonia (Zech et al., 2018).

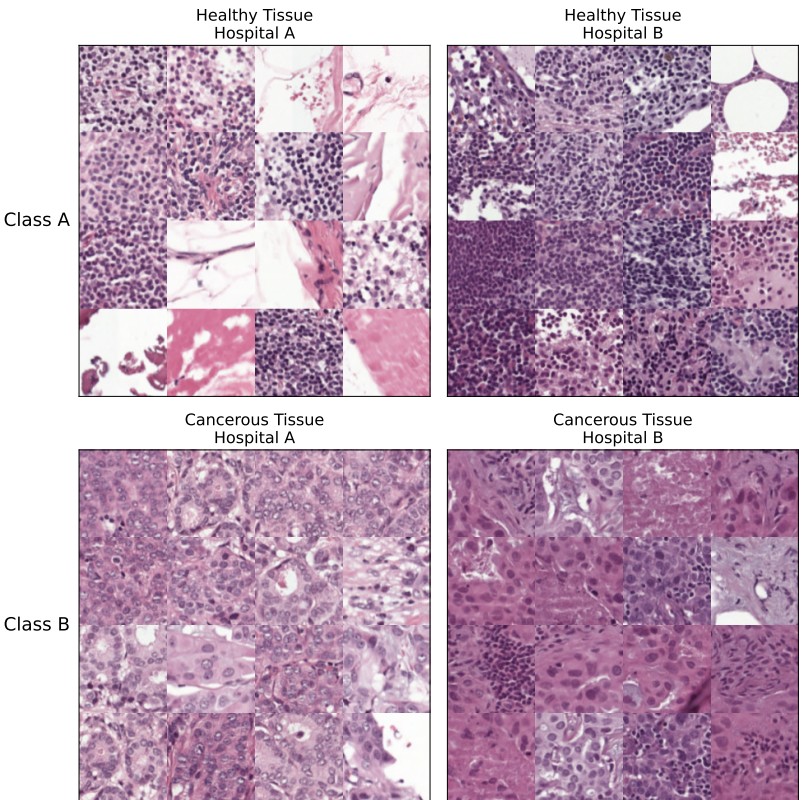

Figure 11: Randomly selected samples of the *CAMELYON17* dataset grouped by presence of cancer metastases (classification target) and hospital of origin (confounder). Images from hospital A depict a pink color cast, while images from hospital B tend to be more purple.

### 3.1.5 Follicles

The Follicles dataset consists of microscopic images of ovarian follicles and was originally introduced in Bender et al. (2023). The task is to classify the maturation stage of each follicle as either *primordial* (Class A) or *growing* (Class B). The correct causal feature for this distinction is the number of granulosa cell layers: a single ring-layer indicates a primordial follicle, while additional cells inside the ring or multiple ring-layers indicate a growing follicle. However, the maturation stage is strongly correlated with the size of the follicle, even though the size itself is not a valid criterion on which to rely for classification and therefore acts as a naturally occurring confounder. The four data groups can be seen in Figure 12

As already mentioned, unlike the other datasets in this study, Follicles is relatively small and therefore cannot be resampled into the symmetric and asymmetric poisoned variants described above. Instead, we will use the Follicles dataset in its original form. The full dataset contains 1507 images downsized to a shape of $256 \times 256$ pixels, with 931 images of primordial follicles and 576 images of growing follicles. Among the primordial follicles, 776 are small (confounder label $q = 0$) and 155 are large ($q = 1$). Among the growing follicles, 117 are small ($q = 0$) and 459 are large ($q = 1$). Hence, the relative group sizes within the dataset are approximately $[A^- : 51.5\%;\ A^+ : 10.3\%;\ B^- : 7.8\%;\ B^+ : 30.5\%]$. This distribution results in a strong group imbalance, with the majority of primordial follicles being small and the majority of growing follicles being large.

Because of the limited number of samples, we cannot construct large test sets like we did for the other datasets without impeding the training process by reducing the size of the training split. Instead, we adopt a more standard split strategy: 80% of the data for training, 10% for validation, and 10% for testing.

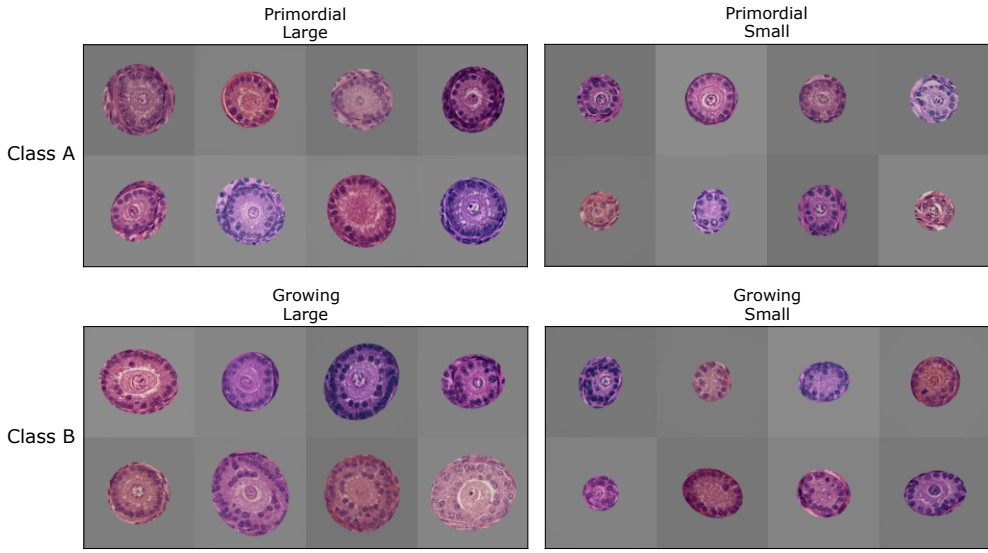

Figure 12: Randomly selected samples from the *Follicles* dataset, grouped by the two target classes *Primordial* (Class A) and *Growing* (Class B), and by the confounding feature, which is the size of the depicted follicle. Surrounding tissue is covered by a mask in order to help the model focus on the follicle.

## 3.2   Implementation details

For the comparative analysis, we apply the XAI-based correction methods CFKD, P-ClArC, and RR-ClArC to mitigate Clever Hans behavior in biased student models. Both P-ClArC and RR-ClArC are evaluated twice, once using the true confounder labels and once using labels obtained via SpRAy, to assess how label quality influences correction performance and to determine each method's robustness to imperfect supervision. As non-XAI-based baselines, we additionally correct with DFR and Group DRO, applying the same dual-evaluation protocol with true and SpRAy-derived group labels. All correction techniques are applied to the identical student models to ensure strict comparability and to isolate the effects of the correction procedures from architectural or initialization variance. The source code is provided alongside our study.

**Evaluation Metrics.**   Instead of relying on the empirical test accuracy, i.e. overall accuracy across all test samples, as the primary evaluation metric, we report the average group accuracy (AGA) on the test data. This metric is obtained by calculating the classification accuracy within each of the four data groups ($A^-$, $A^+$, $B^-$, $B^+$) and then taking the arithmetic mean of these values. In addition, we compute the worst group accuracy (WGA), defined as the minimum accuracy among the four groups. These metrics provide a more reliable measure of generalization under distribution shifts, as they explicitly account for subgroup performance. In contrast, empirical accuracy is often misleading in biased datasets because it is dominated by majority groups, which benefit from the spurious correlations that the correction methods are designed to remove. Consequently, models that achieve high empirical accuracy may still generalize poorly to minority subgroups or new environments where these correlations no longer hold. The main objective for all correction methods is therefore to maximize the average group accuracy, as it reflects balanced performance across all groups without making any assumptions about their individual importance.

**Training the students.**   We trained one student model for each of the poisoned datasets introduced in Section 3.1. All classifiers follow the ResNet-18 architecture (He et al., 2016b) as implemented in the PyTorch Torchvision library[1]. We did not use any precomputed weights for initialization, i.e. all weights were initialized randomly. Each model was trained using standard ERM for a maximum of 300 epochs, with the regularization strength gradually increasing over time. After each epoch, we computed the empirical

---

[1] https://docs.pytorch.org/vision/main/models/generated/torchvision.models.resnet18.html

accuracy on the validation set and eventually retained the model achieving the highest validation accuracy as the biased baseline. These baseline models serve as the starting point for all subsequent correction procedures, ensuring a consistent comparison of the effectiveness of each method in mitigating Clever Hans behavior.

**SpRAy Implementation.** When the true confounder labels were not assumed to be available, we first applied Spectral Relevance Analysis (SpRAy) for obtaining them. The first step is to compute attribution scores for all training and validation samples. Although we can theoretically use any attribution-based XAI technique for this purpose, we decided to use CRP (Achtibat et al., 2023) as implemented in the zennit-crp Python package[2]. This choice aligns with the implementation used in the RR-ClArC experiments provided by Dreyer et al., which can be found at `https://github.com/frederikpahde/rrclarc` and which we adopted as a reference because SpRAy had previously been applied only in the original P-ClArC paper (Anders et al., 2022), for which no official implementation was released.

To obtain high-quality labels with minimal misclassification, we experimented with different model layers $l$ (starting from the penultimate layer) for computing attribution scores and with varying numbers of eigenvalues $k$ for generating the spectral embedding. We also compared different preprocessing strategies, such as using raw attribution scores, channel-wise averaging, or downsampling for potentially improving robustness. We then apply different clustering techniques (k-Means, Agglomerative Clustering, DBSCAN, HDBSCAN) to the Spectral Embedding to automatically generate the SpRAy labels. Additionally, we manually explored the embedded data by computing t-SNE and UMAP visualizations, so that we can judge the quality of the automatic labeling. When required, these visualizations also allowed for manual annotation of clusters using ViRelAy (Anders et al., 2026). We assume enough prior knowledge about the data domain to be able to determine if the computed clusters reflect a valid or a Clever Hans decision strategy.

We generated confounder labels for both the training and validation sets. In case we were unable to achieve a satisfactory separation between confounders and non-confounders for a dataset, neither automatically nor manually, we conducted experiments only with the true confounder labels. For some datasets, acceptable clusters were eventually identified, but their discovery required extensive hyperparameter tuning. Consequently, it is questionable whether a practitioner without prior knowledge about the nature and distribution of the confounder, or even its presence, could reproduce the same results. We elaborate this problem in more detail in section 4.1.

**P-ClArC and RR-ClArC Implementation.** After acquiring group labels through SpRAy, we computed the Concept Activation Vector (CAV) required for both P-ClArC and RR-ClArC. When true confounder labels were assumed to be known, we directly began from this step. For both methods, we conducted an extensive grid search to identify optimal hyperparameters. P-ClArC and RR-ClArC share the hyperparameters regarding CAV computation. These included the layer $l$ at which the base model $f$ was divided into a feature extractor $f_{\text{fe}}^{(l)}$ and a downstream head $f_{\text{dh}}^{(l)}$, the CAV mode (max pooling, mean aggregation, or no reduction), the target class, and the CAV type (a linear SVM-based vector or a PCAV).

In datasets where there is no natural way to label samples as confounders or non-confounders (e.g. for the color cast due to different hospitals of origin in Camelyon17), we treated the label orientation, i.e. deciding which confounder variant corresponded to $q = 0$ or $q = 1$, as an additional hyperparameter, as it determines the CAV's direction and the reference point $z$ for the P-ClArC projection. P-ClArC introduced no further hyperparameters, as the position at which to add the projection layer (*projection location*) is already determined by $l$. For RR-ClArC, we additionally optimized the regularization parameters $\lambda$ and $m$ as well as the function type (square of the dot product or cosine similarity) for the RR-ClArC loss term.

Both methods involved fine-tuning the corrected model for up to 100 epochs. After completing the hyperparameter search, including the optimal number of epochs for fine-tuning, we selected the configuration that achieved the highest average group accuracy on the validation set. Preliminary comparisons of validation-based selection using average group accuracy versus average group loss showed no substantial differences.

---

[2]`https://github.com/rachtibat/zennit-crp`

**CFKD Implementation.**    For CFKD, we first trained a generative model based on a Denoising Diffusion Probabilistic Model (DDPM) on the same poisoned subset that was used for training the student model. Additionally, we trained an oracle model on an unpoisoned dataset version (i.e. all data groups contain the same number of instances) to act as a substitute for a human expert. The oracle solves the same classification task as the biased student but is unaffected by spurious correlations, allowing it to serve as a reliable reference for evaluating counterfactuals generated with the help of the DDPM. Accordingly, we assume a practitioner would have enough domain knowledge to manually classify samples, even those from minority groups, with high accuracy.

During the CFKD procedure, we generated four counterfactual explanations per training and validation sample using SCE (Bender et al., 2026). The oracle then evaluated each counterfactual: if the oracle's prediction matched the class label of the source sample (i.e. the factual), the counterfactual was labeled *false* and retained the source label; if the oracle's prediction switched, it was labeled *true* and assigned the target class label. Only the false counterfactual explanations were used to augment the dataset, as they primarily increase the representation of minority groups and thereby mitigate the spurious correlation between the confounder and the target class. But even if they should not belong to the minority group, they still live in regions of the data distributions for which the student and the teacher disagree, i.e. where the student classifies samples wrongly. Increasing training and validation data density for these regions improves model performance for these regions, thereby improving generalization for regions that were underrepresented in the training and validation data before.

We then fine-tuned the last layer of the student model on the augmented dataset for up to 20 epochs while monitoring the performance on the validation set. Because CFKD does not assume knowledge about group labels, we could not compute the average group accuracy for model selection as in other correction methods. Instead, we chose the optimal number of fine-tuning epochs based on the feedback accuracy. We conducted only a single iteration of the CFKD process, because this is usually sufficient to notably enhance generalization for minority groups, and additional iterations offer diminishing returns. Despite this restriction, CFKD remains the most computationally demanding method in our analysis due to the counterfactual generation process, so limiting the process to a single iteration also improved fairness regarding resource utilization.

This design choice, as well as all other hyperparameter settings for CFKD and SCE, closely followed the reference implementation provided by Bender et al. (2023), which can be found at `https://github.com/Explainable-AI-Berlin/pytorch_explain_and_adapt_library`.

**DFR Implementation.**    As described in section 2.7, it is highly recommended to use held-out data for constructing the balanced fine-tuning set when applying DFR. However, in our setting, the data is intentionally very limited, so reserving some samples for later use with DFR would mean training the original student on a smaller dataset with even fewer minority group samples. Since all correction methods are applied to the exact them students for comparability, they would be impacted by this decision as well, even though they have no use for the additional held-out data. As an alternative, we could provide DFR with additional samples just for fine-tuning, but having additional data available would also constitute an unfair advantage over the other methods. The task for all examined methods should be identical and simulate real-world scenarios: they should mitigate the student's Clever Hans behavior, receiving only the student model and the original training and validation data as input. Consequently, we decided to use a balanced subset of the student's training data for DFR fine-tuning. While ensuring methodological consistency, this design choice likely reduced DFR's effectiveness, which should be kept in mind.

For all datasets except Follicles, each of the four data groups in the DFR fine-tuning set consisted of eight samples, which corresponds to the size of the minority groups in the original training data, resulting in 32 samples in total. For Follicles, the smallest group contains 93 training samples, yielding a total of 372 samples for DFR fine-tuning. During fine-tuning, we monitored the AGA on the validation data after each epoch and selected the best-performing model.

DFR does not introduce any additional hyperparameters beyond those used for standard ERM training, e.g. learning rate and weight decay. We used the default Pytorch SGD optimizer[3] with a learning rate of 0.01 and no regularization.

**Group DRO Implementation.** While Group DRO is normally used as an alternative to ERM when training a model from scratch, we instead used it post hoc on the already biased students to ensure comparability with the other correction methods. In contrast to DFR, Group DRO updates the parameters in all model layers rather than only the final layer, enabling the method to adjust internal representations as well as the classification head.

The key hyperparameters for Group DRO are the weight decay factor $\lambda$ and the model capacity $C$. We evaluated several values for $\lambda \in \{1.0, 0.5, 0.1, 0.05, 0.01\}$ and found that $\lambda = 0.1$ generally yielded the best results. This might seem like an unusually large weight decay factor, but as explained in Section 2.8, Group DRO benefits from strong regularization to reduce the likelihood of overfitting minority group samples. For $C$, we adopted an algorithm by Sagawa et al. which dynamically adjusts $C$ during training. The algorithm itself is not described by Sagawa et al. (2020), but can be found in the provided implementation at `https://github.com/kohpangwei/group_DRO`. This code served as the main reference for our own implementation, also regarding other hyperparameter choices: Consequently, we used the PyTorch SGD optimizer and set the learning rate to 0.0001, together with a momentum factor of 0.9.

While applying Grup DRO to the student, we tracked the AGA on the validation data after each epoch and selected the model with the highest validation AGA as the final corrected version. The maximum validation performance was typically reached after several hundred epochs.

### 3.3 Uncorrected classifiers

Table 1 summarizes the performance of the uncorrected student models before applying any correction method. All students achieved high empirical accuracies on both the training and validation data. However, due to the spurious correlations present in the datasets, tuning by empirical accuracy on the validation set yielded models exhibiting pronounced Clever Hans behavior. Performance on minority groups, where the confounder cannot be exploited as a shortcut feature, is consistently poor, leading to low average group accuracies (AGA) and worst group accuracies (WGA) on validation and test data. Nevertheless, on the training set, students often still achieved perfect accuracy scores for minority groups due to overfitting. On unseen data, however, WGA was in several cases below 10%, illustrating the severity of the model bias.

This effect was particularly strong for the symmetric dataset versions, where the confounder–target correlation is more pronounced and there are two data groups underrepresented in the training data. In contrast, models trained on asymmetric versions, where the correlation is weaker and where there is only one minority group, tended to achieve higher AGA. The complexity of the confounding feature also influenced model behavior: when the confounder was simple (e.g., *background intensity* in Squares or *watermark opacity* in CelebA Smiling), models relied almost exclusively on it, resulting in steep accuracy drops for minority groups and hence especially low WGAs. Conversely, datasets with more complex confounders (e.g., *gender* in CelebA Blond) produced models that were less severely affected by Clever Hans effects. As described earlier, this aligns with findings by Shah et al. (2020) that neural networks often depict a simplicity bias, i.e. they tend to prefer simpler features and to ignore the more complex ones.

Figure 13 visualizes the decision boundaries learned by the uncorrected models trained on symmetric and asymmetric Squares. The Squares data are synthetic and defined by only two numerical features, so we can create new images in a grid pattern and plot the respective confidence scores (i.e. output logits), making it possible to represent the classifier's behavior in two-dimensional space. The plots show that both models strongly rely on the confounding feature. For the symmetric case (Figure 13a), the decision boundary is nearly horizontal, indicating that the model disregards the causal feature entirely. This finding is consistent with the scores in Table 1, as an AGA of approximately 50% and a WGA of almost 0% on the test split indicate nearly perfect accuracy for majority groups ($A^+$ and $B^-$) and complete failure for the minority groups ($A^-$ and $B^+$). Similarly, the model trained on asymmetric Squares (figure 13b) also classified nearly

---

[3]`https://docs.pytorch.org/docs/stable/generated/torch.optim.SGD.html`

Table 1: Accuracies achieved by the uncorrected student models on training, validation, and test data. The test splits for Squares, CelebA Smiling, and Camelyon17 are balanced, i.e. all data groups have the same size, so here the empirical accuracy and the AGA are equal. In contrast, the data distributions in the test splits for CelebA Blond and Follicles are more similar to the distribution on the validation and training splits.

| Dataset | Train accuracy | | | Validation accuracy | | | Test accuracy | | |
|---|---|---|---|---|---|---|---|---|---|
| | empirical | AGA | WGA | empirical | AGA | WGA | empirical | AGA | WGA |
| Squares symmetric | 100.0 | 100.0 | 100.0 | 97.5 | 49.7 | 0.0 | 51.1 | 51.1 | 1.8 |
| Squares asymmetric | 100.0 | 100.0 | 100.0 | 92.0 | 68.7 | 0.0 | 68.1 | 68.1 | 12.0 |
| Smiling symmetric | 89.8 | 55.0 | 12.5 | 94.0 | 48.0 | 0.0 | 51.3 | 51.3 | 7.3 |
| Smiling asymmetric | 100.0 | 100.0 | 100.0 | 77.5 | 56.7 | 0.0 | 59.4 | 59.4 | 1.0 |
| Blond symmetric | 100.0 | 100.0 | 100.0 | 92.0 | 71.4 | 50.0 | 80.3 | 72.7 | 38.2 |
| Blond asymmetric | 100.0 | 100.0 | 100.0 | 89.0 | 79.2 | 50.0 | 86.9 | 76.2 | 40.6 |
| Camelyon17 symmetric | 100.0 | 100.0 | 100.0 | 99.0 | 75.0 | 0.0 | 55.3 | 55.3 | 9.2 |
| Camelyon17 asymmetric | 100.0 | 100.0 | 100.0 | 88.5 | 77.5 | 0.5 | 75.4 | 75.4 | 42.1 |
| Follicles | 89.4 | 77.2 | 49.4 | 89.3 | 79.2 | 54.2 | 84.5 | 72.5 | 48.1 |

all minority group samples $(B^+)$ incorrectly. In addition, the unequal sizes of the majority groups also reduced the model's confidence for data in $A^-$ that lie close to the distribution of $B^-$.

These results confirm that models trained under standard ERM systematically exploit confounding features as shortcuts when strong spurious correlations are present. This baseline evaluation thus provides a reference point for assessing the effectiveness of the correction methods, which we evaluate and compare in the following section.

## 4 Results and Discussion

Building upon the experimental setup outlined in Section 3, this section presents the outcomes of our comparative evaluation of the selected correction methods. We report quantitative results for all datasets, analyze the impact of each method on group-balanced generalization, and discuss qualitative findings that provide further insight into the strengths and limitations of the individual approaches.

Performance was evaluated on the test splits introduced in Section 3.1 using two complementary metrics: average group accuracy (AGA) and worst group accuracy (WGA). Both metrics explicitly account for subgroup performance and thus provide a more reliable estimate of generalization under distribution shifts than empirical accuracy, which is dominated by majority groups.

Table 2 shows the results when providing DFR, Group DRO, P-ClArC, and RR-ClArC with ground-truth group labels. In nearly all cases, applying the correction methods improved AGA, indicating an overall enhancement in the ability to generalize across different data groups compared to the uncorrected students. However, the magnitude of improvement varied substantially between methods. Non-XAI-based approaches, particularly Deep Feature Reweighting (DFR), achieved only moderate gains in AGA. In contrast, the XAI-based methods P-ClArC, RR-ClArC, and CFKD produced considerably larger improvements in most settings. Except for asymmetric Camelyon17, the best-performing method on every dataset was one of these XAI-based approaches. Among them, CFKD achieved the strongest results most consistently, reaching the highest AGA in six out of nine cases and ranking second in two additional ones. It was also the only method that substantially improved generalization on CelebA Blond, the dataset with arguably the most complex

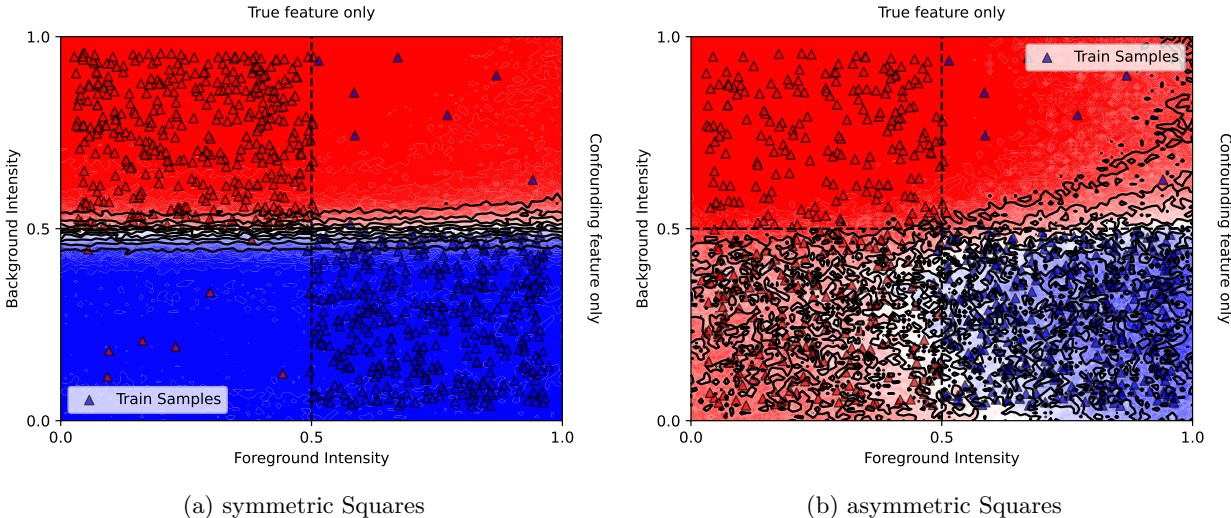

(a) symmetric Squares          (b) asymmetric Squares

Figure 13: Decision boundaries and confidence regions of the uncorrected student models trained on (a) symmetric and (b) asymmetric Squares. Training samples from Class A with low foreground intensity are depicted as red triangles, training samples from Class B with high foreground intensity as blue triangles. Likewise, new data in the red area is classified as Class A, and in the blue area as Class B. Color saturation reflects the model's confidence. A perfect decision boundary would be a vertical line at 0.5 foreground intensity, as that would separate the two classes by the true causal feature, rendering the model completely insensible towards the confounder (the background intensity). Due to Clever Hans, the actual decision boundaries deviate heavily from the perfect solution in both cases.

confounder *gender*. RR-ClArC performed second best overall and surpassed CFKD on two datasets. Both RR-ClArC and CFKD, however, failed to mitigate Clever Hans behavior on asymmetric Camelyon17, making P-ClArC the only XAI-based method that improved performance on every dataset. The poor performance of the XAI-based methods for asymmetric Camelyon17 led to DFR ranking best here, albeit its AGA increase is smaller than for the symmetric version. DFR also performed relatively well on Follicles, likely due to the larger size of the fine-tuning set.

The WGA results largely confirmed the findings for AGA. Again, XAI-based methods on average achieved the highest scores, with CFKD ranking first and RR-ClArC second. P-ClArC and Group DRO were more competitive in WGA than in AGA, but since model selection was based on AGA, this remains the primary evaluation metric. We still report WGA for completeness, as an increase in AGA may occur without a corresponding improvement in minority group performance or even with a decline. For example, applying DFR to the student trained on symmetric CelebA Blond increased AGA but slightly reduced WGA. Because no group was considered more important than another, an overall increase in mean group accuracy was still regarded as an improvement. In most cases, however, enhancements in AGA and WGA occurred simultaneously.

Moreover, no clear pattern emerged to suggest that some correction methods perform systematically better on either the symmetric or asymmetric dataset variants. The effectiveness of each method varied, with examples of stronger performance on both versions. Overall, the magnitude of improvement was largely consistent between the two configurations. Although the final AGA for asymmetric datasets was typically higher, this was proportional to the higher baseline AGA of the uncorrected student, resulting in a similar net performance gain.

We conducted the same experiments again with group labels provided by SpRAy to assess how label quality influences the correction performance, determining the robustness of each method to imperfect supervision. The corresponding results are summarized in Table 3. We do not report results for symmetric CelebA Smiling, both symmetric and asymmetric CelebA Blond, as well as symmetric Camelyon17, because for

Table 2: Average group accuracies (AGAs) and worst group accuracies (WGAs) on unseen test data before and after applying various correction methods, using **ground-truth group labels** for DFR, Group DRO, P-ClArC, and RR-ClArC. Model selection was based on validation AGA.

| Dataset | Uncorrected | DFR | Group DRO | P-ClArC | RR-ClArC | CFKD |
|---|---|---|---|---|---|---|
| *average group accuracy (AGA)* | | | | | | |
| Squares symmetric | 51.1 | 52.1 | 61.3 | 78.6 | 79.6 | **94.5** |
| Squares asymmetric | 68.1 | 73.9 | 77.1 | 85.2 | **92.4** | 91.5 |
| Smiling symmetric | 51.3 | 57.5 | 56.3 | 65.9 | 68.7 | **79.6** |
| Smiling asymmetric | 59.4 | 58.2 | 64.0 | 65.6 | 80.5 | **86.6** |
| Blond symmetric | 72.7 | 73.1 | 73.0 | 74.4 | 74.5 | **79.1** |
| Blond asymmetric | 76.2 | 76.9 | 77.2 | 77.1 | 78.6 | **87.2** |
| Camelyon17 symmetric | 55.3 | 63.2 | 72.6 | 72.1 | **81.3** | 78.7 |
| Camelyon17 asymmetric | 75.4 | **81.8** | 81.0 | 81.5 | 75.1 | 75.4 |
| Follicles | 72.5 | 82.7 | 79.2 | 76.6 | 79.8 | **83.9** |
| *worst group accuracy (WGA)* | | | | | | |
| Squares symmetric | 1.8 | 4.8 | 50.3 | 37.3 | 53.6 | **88.3** |
| Squares asymmetric | 12.0 | 50.0 | 58.8 | 63.0 | **81.0** | 78.0 |
| Smiling symmetric | 7.3 | 34.1 | 4.8 | 27.6 | 32.6 | **51.2** |
| Smiling asymmetric | 1.0 | 1.5 | 42.7 | 14.3 | 65.4 | **84.4** |
| Blond symmetric | 38.2 | 35.8 | 39.4 | 38.8 | **55.8** | 52.1 |
| Blond asymmetric | 40.6 | 41.8 | 43.0 | 55.8 | 66.1 | **67.9** |
| Camelyon17 symmetric | 9.2 | 23.0 | 53.9 | 62.6 | **68.7** | 45.9 |
| Camelyon17 asymmetric | 42.1 | 60.5 | **75.6** | 65.3 | 54.9 | 36.2 |
| Follicles | 48.1 | **70.8** | 66.7 | 66.7 | 66.7 | 66.7 |

these datasets, we were not able to acquire SpRAy labels of sufficiently high quality. In general, both AGA and WGA scores were lower compared to results obtained with ground-truth labels, which is expected due to the presence of label noise in the SpRAy-derived annotations. This further highlights the comparative advantage of CFKD, which does not rely on group labels at all. However, in some cases, performance improved compared to the runs with ground-truth labels, e.g. for Group DRO on symmetric Squares and symmetric CelebA Smiling, or P-ClArC on Follicles. These fluctuations suggest a lack of robustness, most likely because the validation splits contained only two samples per minority group, rendering model selection based on validation AGA inherently unreliable.

Table 3: Average group accuracies (AGA) and worst group accuracies (WGA) on unseen test data before and after applying various correction methods, using **group labels provided by SpRAy** for DFR, Group DRO, P-ClArC, and RR-ClArC. Datasets for which SpRAy did not yield labels of sufficient quality to conduct experiments are omitted. Test metrics were computed using ground-truth labels. Model selection was based on validation AGA.

| Dataset | Uncorrected | DFR | Group DRO | P-ClArC | RR-ClArC | CFKD |
|---|---|---|---|---|---|---|
| *average group accuracy (AGA)* | | | | | | |
| Squares symmetric | 51.1 | 52.1 | 64.2 | 76.1 | 86.8 | **94.5** |
| Squares asymmetric | 68.1 | 68.8 | 76.8 | 76.2 | 90.4 | **91.5** |
| Smiling symmetric | 51.3 | 54.0 | 59.6 | 64.6 | 57.0 | **79.6** |
| Camelyon17 asymmetric | 75.4 | **77.5** | 77.0 | 76.5 | 75.8 | 75.4 |
| Follicles | 72.5 | 76.1 | 73.5 | 78.3 | 68.8 | **83.9** |
| *worst group accuracy (WGA)* | | | | | | |
| Squares symmetric | 1.8 | 5.0 | 49.0 | 39.7 | 75.2 | **88.3** |
| Squares asymmetric | 12.0 | 45.2 | 56.0 | 32.2 | 73.8 | **78.0** |
| Smiling symmetric | 7.3 | 18.8 | 40.8 | 34.6 | 12.6 | **51.2** |
| Camelyon17 asymmetric | 42.1 | **57.5** | 38.6 | 32.2 | 34.7 | 36.2 |
| Follicles | 48.1 | **66.7** | 59.3 | **66.7** | 45.8 | **66.7** |

Among all methods, the largest deviations between results obtained with true and SpRAy-derived labels occurred for RR-ClArC, where performance differences exceeded 10 percentage points in some cases. Although RR-ClArC generally performed well relative to the other methods, this lack of robustness is concerning and cannot be solely attributed to label noise, since P-ClArC (which uses the same CAV) did not exhibit such large discrepancies. Moreover, even when the SpRAy labels were nearly identical to the ground truth, as for the symmetric Squares dataset, RR-ClArC's performance still diverged notably.

The following subsections provide a detailed discussion about the performance of the individual correction methods. We also include a description of our experience with SpRAy and quantify SpRAy label quality, providing additional context for the results in Table 3.

### 4.1 SpRAy

Many correction methods, including DFR, Group DRO, and the ClArC family, require that group labels are available for at least a representative subset of samples. Although these methods typically assume that group labels are known beforehand, such annotations are rarely available in practice. Manually identifying and labeling distinct data groups quickly becomes infeasible for larger datasets. Consequently, even though these methods do not formally depend on SpRAy, it effectively becomes a prerequisite for their practical use to obtain group labels at scale. To the best of our knowledge, there appears to be no broadly applicable alternative to SpRAy for this purpose. Because the downstream correction performance crucially depends on the label quality, we first discuss SpRAy's reliability and limitations.

Table 4 summarizes the quality of the group labels generated by SpRAy. For each dataset, the table lists the true group sizes, the sizes of the groups identified by SpRAy, and the alignment between SpRAy labels and ground-truth labels. These results show that the label quality quickly deteriorated as the confounding feature became more complex. The most accurate labels were obtained for Squares, where the confounder (background brightness) is simple and visually salient. In contrast, for CelebA Blond with a far more complex confounder *gender*, we could not derive any reliable SpRAy labels, neither for the symmetric nor the asymmetric version. While simpler than *gender*, the confounders in CelebA Smiling, Camelyon17, and Follicles are still more complex than the background intensity in Squares: For Camelyon17, the color-cast confounder is conceptually simple, but often subtle; for Follicles, the confounder (follicle size) can be ambiguous for irregular shapes, with many images lying close to the threshold between *small* ($q = 0$) and *large* ($q = 1$); and for CelebA Smiling, the watermark opacity occupies only a small image region and may blend with background textures, especially when more transparent. Nevertheless, we were able to acquire group labels for at least one version of these datasets, although with considerably lower accuracy than for Squares. In particular, label accuracy for minority groups was often poor, reaching as low as 20% in the worst case. These findings explain several of the performance declines observed in Table 3, where methods using SpRAy labels usually showed weaker correction compared to those using ground-truth annotations.

Table 4: Quality of the confounder labels generated by SpRAy. For each dataset, the table lists the true group sizes, the corresponding group sizes identified by SpRAy, and the alignment between SpRAy-derived and ground-truth labels. The reported SpRAy Label Accuracy denotes the proportion of samples within each group (as identified by SpRAy) whose SpRAy-assigned confounder label matches the true confounder label. Within each cell, values from left to right refer to the data groups $A^-$, $A^+$, $B^-$, $B^+$.

| Dataset | True Group Sizes | SpRAy Group Sizes | SpRAy Group Label Accuracy |
|---|---|---|---|
| Squares symmetric | 490/10/10/490 | 490/10/11/489 | 100.0/100.0/81.8/99.8 |
| Squares asymmetric | 250/250/490/10 | 248/252/491/9 | 99.2/98.4/99.8/100.0 |
| Smiling symmetric | 490/10/10/490 | 470/30/12/488 | 99.1/20.0/58.3/99.4 |
| Smiling asymmetric | 250/250/490/10 | — | — |
| Blond symmetric | 490/10/10/490 | — | — |
| Blond asymmetric | 250/250/490/10 | — | — |
| Camelyon17 symmetric | 490/10/10/490 | —/—/11/489 | —/—/72.7/99.6 |
| Camelyon17 asymmetric | 250/250/490/10 | 195/299/489/11 | 71.3/64.5/99.8/81.8 |
| Follicles | 401/125/114/660 | 434/92/145/629 | 87.3/76.1/63.4/96.5 |

Overall, the results in Table 4 illustrate that SpRAy performs reliably only when the confounding feature is conceptually simple and clearly separable in the relevance space. As soon as the confounder becomes more complex or less distinct, label quality declines sharply. This pattern is consistent across datasets and highlights SpRAy's sensitivity to both feature complexity and data scarcity.

Especially for CelebA Smiling, we expected to achieve better results, considering that in the original SpRAy paper (Lapuschkin et al., 2019), a watermark confounder was used as a demonstrative example. However, in our setting, the watermark is not a discrete feature that is either present or absent. Instead, it is present in every single sample, but with varying opacity along a continuous range. While this increases the complexity of the task, such poor label accuracy was still surprising. Of course, there are additional factors that might also have contributed to the low quality, most importantly, the extremely small number of elements in the minority groups. This size might be sufficient for the simple Squares dataset. But in more complex datasets, the few minority samples may have been too heterogeneous for SpRAy to identify consistent attribution patterns between them that also clearly distinguish them from the majority group. Using continuous instead of discrete confounders potentially made this problem even more severe. The large difference in group sizes also increases the risk that SpRAy captures unintended confounding features that arise by chance.

An example of this can be observed in Figure 14a: a small number of Camelyon17 samples from Hospital A contain an artifact in the form of a black bar that covers part of the image. In symmetric Camelyon17, Hospital A samples form the majority group in Class A and the minority group in Class B, leading to a higher probability for the artifact to appear in Class A. In fact, it exclusively appeared in Class A images in our experiment, and hence became strongly correlated with the corresponding class label. The artifact images stand out more prominently than genuine minority-group samples and can therefore "distract" SpRAy, which prevented the identification of the actual confounding pattern based on color cast.

A similar problem occurred in CelebA Blond: instead of clustering the samples by the designated confounder (*gender*), visualizations of the Spectral Embedding often showed them grouped by their background color (Figure 14b). This suggests that the background was spuriously correlated with the class labels *Blond* and *Not Blond* in our subsampled dataset.

While conducting experiments with these newly identified confounding features might have been interesting, such analyses are beyond the scope of this study. Moreover, no ground-truth labels are available for these

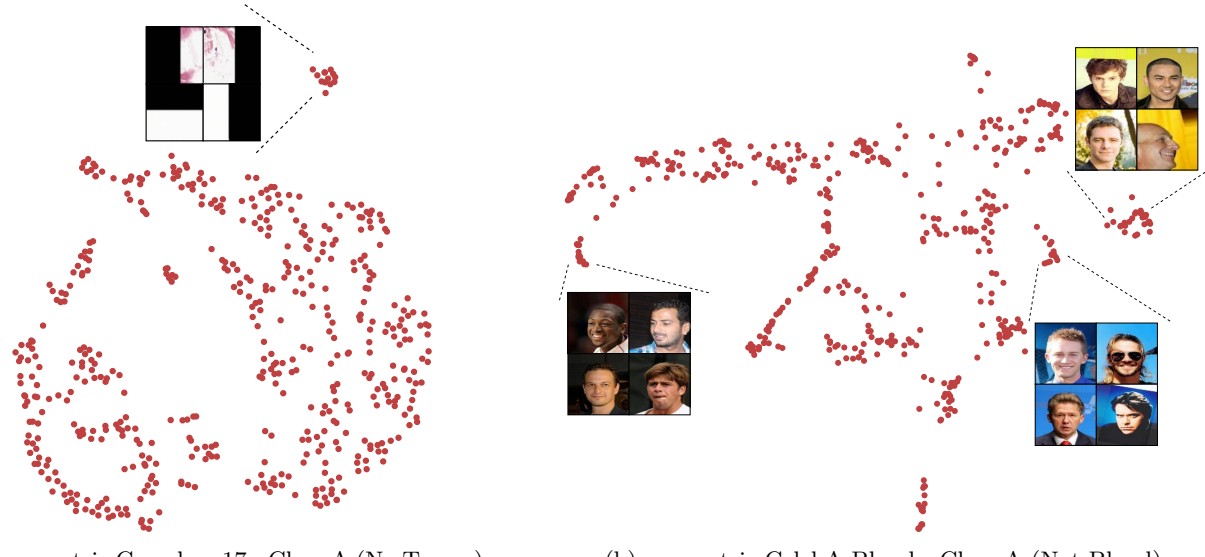

(a) symmetric Camelyon17 - Class A (No Tumor)          (b) symmetric CelebA Blond - Class A (Not Blond)

Figure 14: Examples of SpRAy clustering samples by unintended concepts. (a) In Camelyon17, SpRAy grouped images by the presence of an image artifact rather than by color cast. (b) In CelebA Blond, SpRAy clustered samples according to background color instead of gender.

features, which prevents a reliable evaluation of performance within individual data groups as done for the designated confounders. For them, SpRAy often produced unsatisfactory results. Even after extensive hyperparameter tuning and manual inspection of numerous spectral embeddings, Squares remained the only dataset for which SpRAy consistently generated accurate group labels. In the following, we provide additional insights into our experience with SpRAy.

**Automatic Clustering versus Virelay.** We obtained the SpRAy labels by computing the Spectral Embedding with varying hyperparameters, visualizing the result with t-SNE and UMAP, and then manually delineating clusters with Virelay (Anders et al., 2026). Notably, automatic clustering of the Spectral Embedding using algorithms such as k-Means, Agglomerative Clustering, DBSCAN, or HDBSCAN almost never produced usable labels. Their alignment with the ground truth was too poor to support subsequent correction. A possible explanation is that the clustering algorithms struggle to cope with the extreme size differences between the data groups. In contrast, visually inspecting the Spectral Embedding and then manually clustering the samples with Virelay proved far more reliable. Among the automatic methods, only DBSCAN occasionally produced acceptable results, as illustrated in Figure 15.

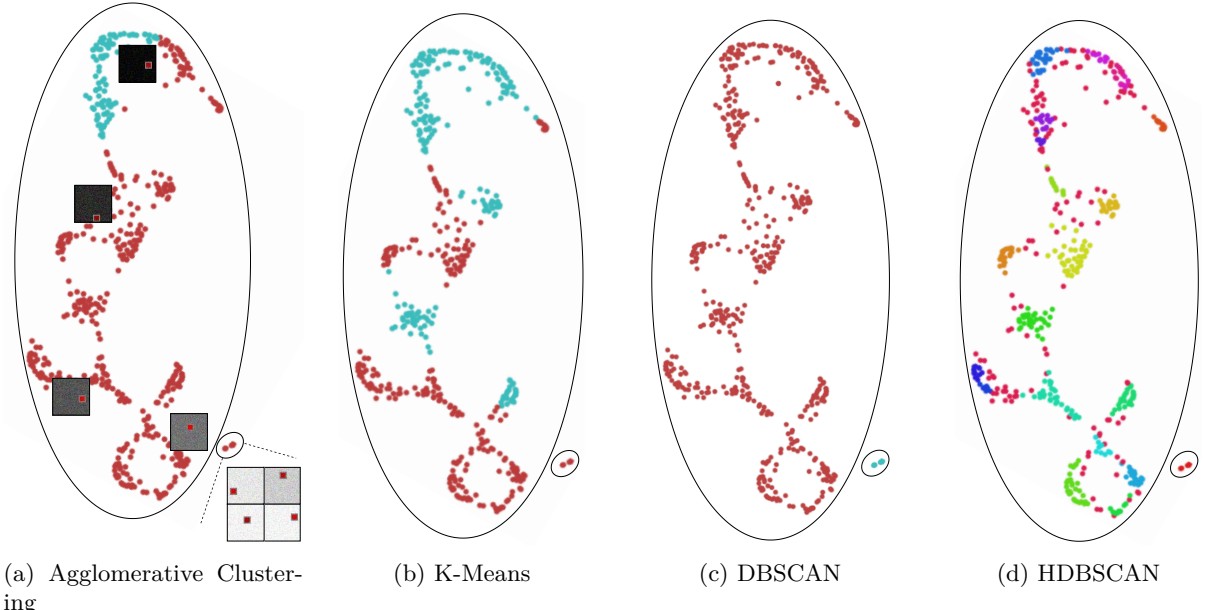

(a) Agglomerative Clustering     (b) K-Means     (c) DBSCAN     (d) HDBSCAN

Figure 15: T-SNE visualizations of the Spectral Embedding for Class B of symmetric Squares. The circled clusters correspond to those manually identified with ViRelAy and used for the results reported in Table 4. Data points are colored according to cluster assignments obtained through (a) Agglomerative Clustering, (b) K-Means, (c) DBSCAN, and (d) HDBSCAN. Except for DBSCAN, the automatically derived clusters align poorly with the true confounder labels, whereas the manually identified clusters match them almost perfectly.

**Visualization techniques.** As automatic clustering frequently failed, we had to rely on manual clustering with Virelay, which can only be successful with faithful visualization of the Spectral Embedding. We generated these visualizations using both t-SNE and UMAP.

We obtained the best results for *perplexity* values between 10 and 70 with t-SNE and a *number of neighbors* value between 5 and 60 with UMAP. Within these hyperparameter ranges, both techniques often yielded comparable outcomes: when the groups were clearly separated in one visualization, they were usually also separated in the others. Nevertheless, computing multiple visualizations with varying hyperparameter settings was helpful for identifying the different decision-making strategies. UMAP generally produced sharper separations between groups, which often facilitated the identification of distinct decision strategies. However, UMAP also tended to fragment the majority group into several subclusters. This can potentially be mis-

leading, especially in settings where it is unknown how many distinct decision strategies can be expected to be found. In contrast, t-SNE visualizations were typically smoother, often showing a continuous alignment of samples according to the confounder value rather than sharply delineated groups.

Figure 16 illustrates the differences between the two techniques. Overall, neither method was universally superior. We found that computing several visualizations and examining them jointly provided the most robust basis for identifying distinct decision strategies.

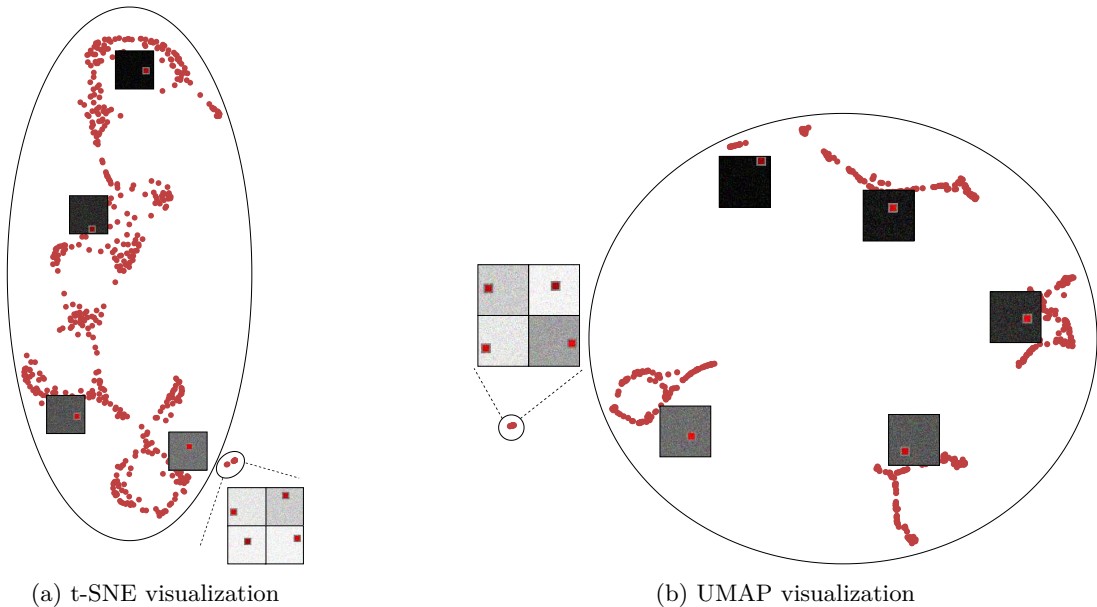

(a) t-SNE visualization          (b) UMAP visualization

Figure 16: Comparison of (a) t-SNE and (b) UMAP visualizations of the Spectral Embedding for Class B of symmetric Squares. With UMAP, the two groups tend to appear more distinctly separated, but the majority group often is more fractured. In contrast, t-SNE typically yields smoother, more continuous alignments of samples by confounder value, but the two groups are less clearly delineated. Both techniques provide complementary perspectives, and exploring multiple visualizations proved beneficial for reliable manual clustering with Virelay.

**Choosing the optimal layer.** Among all SpRAy hyperparameters, the choice of the layer $l$ from which attribution scores are computed had the strongest impact on label quality. Intuitively, selecting a deeper layer (such as the penultimate layer) often seems advantageous because high-level, human-interpretable concepts tend to be better captured there. However, for simple confounders, we found that intermediate layers closer to the input layer frequently yielded clearer and more stable results. For example, Figure 17 compares the two-dimensional visualizations of the spectral embeddings obtained for the symmetric Squares dataset when attribution maps were computed at layer 6 and at the penultimate layer 12 (See Appendix A for a detailed overview of the model architecture). When using layer 6, samples are well aligned along the confounding dimension (background brightness), with bright and dark samples forming two separated clusters. In contrast, the embedding derived from layer 12 exhibits greater fragmentation of the majority group and weaker alignment with the confounder. This suggests that the background-intensity feature is more distinctly represented in earlier layers of the model.

For the other datasets (Follicles, symmetric CelebA Smiling, and asymmetric Camelyon17), we usually obtained the best results from the penultimate or second-to-last layer. Determining the most suitable layer, however, required considerable trial and error and visual examination of each Spectral Embedding. This manual exploration became even more time-consuming when also tuning additional hyperparameters.

**Other considerations.** The performance of SpRAy is dependent upon a wide array of hyperparameters beyond the selection of a specific network layer l. Each of the underlying clustering algorithms applied

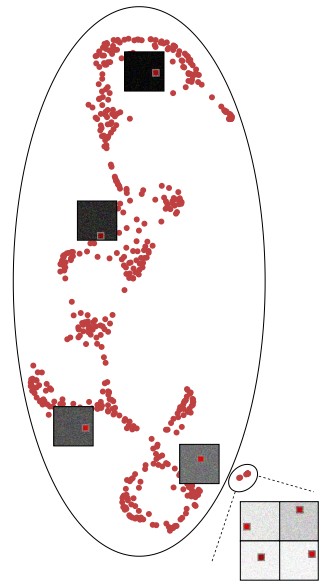
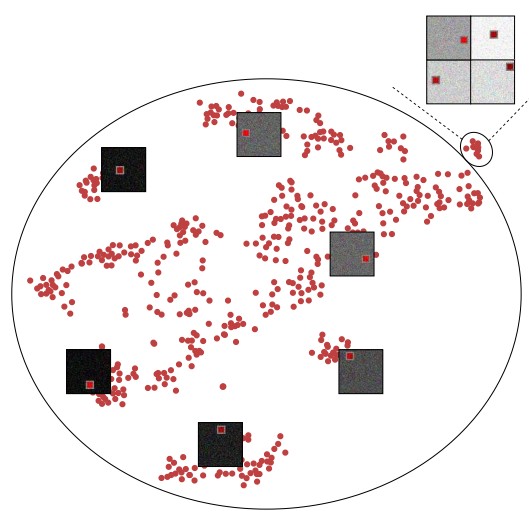

(a) SpRAy applied at intermediate layer (layer 6)  (b) SpRAy applied at penultimate layer (layer 12)

Figure 17: t-SNE visualizations of the spectral embeddings obtained by applying SpRAy to Class B samples of the symmetric Squares dataset using attribution maps from (a) an intermediate layer (layer 6) and (b) the penultimate layer (layer 12) of the biased model. Using layer 6 attribution maps leads to a better alignment of the Spectral Embedding with the confounder values.

to the spectral embedding (k-Means, Agglomerative Clustering, DBSCAN, or HDBSCAN) introduces its own set of parameters. Furthermore, the dimensionality of the spectral representation, determined by the number of eigenvectors k, is a critical factor. A lower value for k might effectively isolate a singular, dominant confounder, but may prove insufficient for capturing more complex, multifaceted confounders that are composed of multiple features.

In addition to these factors, the choice of the XAI technique and its own configuration for generating attribution maps is paramount. For instance, the *zennit* library offers numerous customizable components, including contributors, composites, and canonizers, each with their own hyperparameters. Although we tried several configurations, the relationship between the quality of the attribution maps and the final efficacy of SpRAy did not seem to be direct. While investigating the CelebA dataset, we observed that visually interpretable heatmaps did not necessarily yield successful SpRAy outcomes. Conversely, on the Squares dataset, attribution maps of lower visual quality often produced highly effective spectral embeddings. This suggests that the ultimate success of SpRAy hinges less on the absolute quality of individual attribution maps and more on a crucial condition: the consistency of attribution patterns within a group and the distinctiveness of these patterns between different groups. Likely, the low number of minority group samples in our setting made it, in some cases, impossible for SpRAy to identify these patterns.

Consequently, achieving usable results with SpRAy presented a significant challenge in our experimental work. The method's success was highly sensitive not only to the choice of hyperparameters, but also to the inherent properties of the dataset. However, the difficulty of this task is contextualized by the notable absence of alternative approaches for obtaining group labels besides manual annotation. The scarcity of such methods underscores the inherent complexity of the problem, even though group labels are a prerequisite for many common correction methods.

## 4.2 DFR

Among all examined correction methods, DFR was the easiest to implement and the fastest to converge. As it introduces no additional hyperparameters compared to a standard training regime, it circumvents the

need for extensive grid searches. However, DFR typically yielded only modest AGA improvements on the test data, particularly when compared to the XAI-based methods (cf. Table 2).

Surprisingly, DFR was most effective on the more complex medical datasets, achieving the highest accuracy of all methods on asymmetric Camelyon17 and the second-highest on Follicles. In contrast, its impact was minimal or even detrimental on datasets with simpler confounders, such as Squares and asymmetric CelebA Smiling. One plausible explanation for this pattern lies in DFR's architectural restriction: it always splits the model at the penultimate layer and only fine-tunes the final fully connected layer. This design can be advantageous when the confounding and causal features are complex and thus well captured in the deepest layers of the network. However, for datasets where the confounder and causal feature are simple, such as the foreground and background intensities in Squares, the relevant information may be better disentangled earlier in the network. In such cases, fine-tuning only the last layer may not suffice to remove the bias, limiting DFR's correction effect.

CelebA Blond presents an apparent exception to this pattern. While its causal feature, i.e. whether the hair of the depicted person is blond, is relatively simple, its confounder *gender* is arguably the most complex. Still, applying DFR did not notably improve the student. This outcome is likely attributable to the severe scarcity of fine-tuning data, which is exacerbated by the dataset's higher complexity. With only eight samples per group, it is improbable that such a small dataset could faithfully represent the distributions of each subgroup, rendering the optimization ineffective. This also further explains DFR's strong performance on the Follicles dataset, where a more substantial fine-tuning set of 93 samples per group provided a more robust representation of the data, thereby enabling a notable correction of the model's Clever Hans behavior.

Another challenge limiting DFR's efficacy is pre-fine-tuning accuracy saturation. In most cases, including for both symmetric and asymmetric CelebA Blond, the student model already achieved perfect accuracy on the 32 fine-tuning samples before DFR was applied. Consequently, the initial loss was negligible, which prevented meaningful updates to the final layer's weights and rendered the fine-tuning process largely inert. This issue may be particularly acute for datasets with simpler features, such as Squares, where the model's tendency to overfit on the few minority group samples during initial training is higher. This more severe overfitting results in an even lower starting loss when those same samples are used for fine-tuning. Follicles serves as a notable exception and reinforces this explanation, as it was the only dataset for which the student did not almost perfectly classify all fine-tuning samples already. In this case, the model's initial accuracy on the fine-tuning data was a lower 81.6%, which provided a sufficient loss signal for the correction to be effective. Using a held-out dataset for fine-tuning, as originally recommended by Kirichenko et al. (2023), would likely have improved results, as this would have prevented the students from overfitting on the fine-tuning data. However, as we explained in Section 3.2, we intentionally chose to use a subset of the original training data to maintain comparability across all methods.

Figure 18 illustrates the decision boundaries and the confidence regions after applying DFR to the models trained on symmetric and asymmetric Squares, both with true confounder labels and labels provided by SpRAy. Consistent with the negligible changes in AGA and WGA reported in Table 2 and 3 for symmetric Squares, the decision boundaries and confidence distributions (Figures 18a and 18b) remained virtually identical to those of the uncorrected models (Figure 13a). For the asymmetric variant, however, DFR led to some minor, yet visible improvements when using true group labels: While before correction nearly all minority group samples were wrongly classified, the corrected classifier is now accurate for a small portion of the minority group, and the overall decision boundary moved slightly closer to the theoretical optimum (cf. Figure 18c). Nevertheless, these improvements came at the expense of reduced confidence in some majority group regions, and even occasional misclassifications within the majority groups. When using SpRAy labels, similar changes were observed, although they were more pronounced, which led to the majority group $A^-$ becoming the weakest-generalizing group instead of the minority group $B^+$ (Figure 18d). However, since the SpRAy labels for the Squares dataset are almost perfectly accurate, these deviations might simply be due to random variance during optimization.

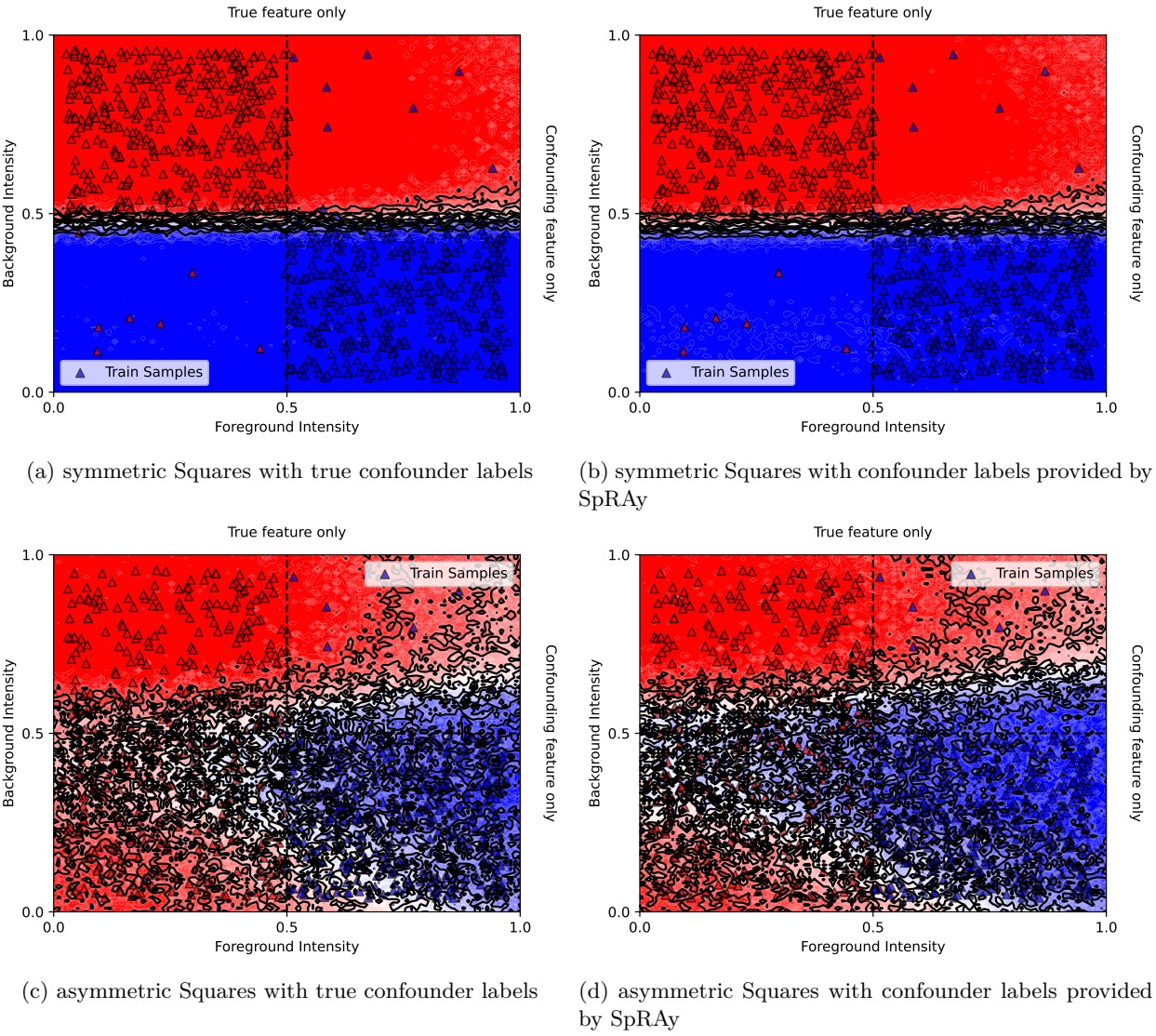

(a) symmetric Squares with true confounder labels

(b) symmetric Squares with confounder labels provided by SpRAy

(c) asymmetric Squares with true confounder labels

(d) asymmetric Squares with confounder labels provided by SpRAy

Figure 18: Decision boundaries and confidence regions for Squares after applying **DFR** to the student. The optimal decision boundary corresponds to a vertical line at 0.5 foreground intensity. Panels (a) and (b) depict results for symmetric Squares using true confounder labels and SpRAy-derived labels, respectively. Panels (c) and (d) show the corresponding results for asymmetric Squares.

## 4.3 Group DRO

Group DRO generally achieved stronger corrections than DFR, yielding a noticeable improvement in average group accuracy for most datasets. Only for asymmetric CelebA Blond, asymmetric Camelyon17, and Follicles did DFR perform marginally better, with Follicles being the only case where the difference exceeded one percentage point. These gains over DFR came at the cost of longer convergence times. Whereas DFR typically converged within a few epochs, Group DRO often required several hundred, or sometimes even over one thousand, epochs to reach stability. Even when accounting for the time that could have been saved by using Group DRO directly for training from scratch, instead of first training a student with standard ERM, Group DRO would have remained more computationally expensive, as ERM training usually converged well before 300 epochs. Although each epoch is only slightly slower than standard ERM, the total training time increased substantially because of the greater number of required iterations. Nonetheless, even the longest

runs never exceeded two hours on our hardware, given that each training set comprised only 800 samples (1205 for Follicles) and each Group DRO epoch was only slightly slower compared to ERM.

One advantage of Group DRO, which it shares with DFR, is its simplicity in hyperparameter tuning. Since the weight decay factor $\lambda$ was the only parameter requiring optimization, finding the best configuration was faster than for the XAI-based methods. But while Group DRO usually was able to outperform DFR, it generally produced smaller performance gains than RR-ClArC or CFKD. On average, these XAI-based methods achieved substantially higher improvements in both AGA and WGA scores. The two datasets where Group DRO matched or slightly exceeded their performance were again asymmetric Camelyon17 and Follicles, where RR-ClArC and CFKD performed below their usual level. Overall, Group DRO's results were relatively stable across datasets, typically improving the biased student model by a consistent margin, while other correction methods, e.g. DFR and RR-ClArC, showed higher variance in their effectiveness.

The advantage of Group DRO over DFR likely stemmed from the ability to update all model parameters, rather than being confined to adjusting only the weights of the final classification layer. Furthermore, Group DRO leveraged the entire training dataset, in contrast to DFR's reliance on a small subset, although both sets include the same number of minority group samples. Despite this, given Group DRO's prominence and its recognition as a SOTA correction technique, we would have anticipated more substantial performance improvements. However, the method likely encountered similar problems as DFR: First, since Group DRO was applied post-hoc to an already trained classifier, the initial loss values were often already minimal, resulting in only negligible weight updates during optimization. Second, the limited number of samples available for the minority groups might be insufficient to accurately represent the underlying group distribution, especially for more complex data, thus imposing an upper bound on the achievable generalization capabilities.

For the Squares dataset, Figure 19 visualizes the decision boundaries and confidence scores after correction with Group DRO. The corrected models show some improvements compared to the uncorrected students shown in Figure 13a. On the symmetric version, nearly all minority group samples were misclassified before correction, whereas after applying Group DRO, approximately half are now correctly classified (Figures 19a and 19b). However, this gain in minority group accuracy comes at the expense of lower confidence scores and even reduced accuracy for majority groups. In addition, the new decision boundary takes on a non-smooth, irregular shape.

For the asymmetric version, the results are more favorable (Figures 19c and 19d). Here, Group DRO substantially increases the number of correctly classified minority samples while maintaining high accuracy for the majority groups, even though confidence scores are also lowered. The resulting decision boundary lies closer to the theoretical optimum compared to the uncorrected student, indicating that the model relies less on the confounding feature. Nevertheless, it should be noted that, for the asymmetric setting, the uncorrected Clever Hans decision boundary is already more similar to the optimal solution.

Because the SpRAy-derived labels for the Squares dataset are almost perfectly accurate, the small deviations between results obtained with true labels and those using SpRAy labels are likely due to chance rather than label noise.

### 4.4 P-ClArC

As indicated in Tables 2 and 3, P-ClArC demonstrated consistent and reliable performance, yielding a noticeable improvement over the uncorrected student model in all experiments. Although it was generally outperformed by RR-ClArC and CFKD, P-ClArC was the only XAI-based method by which an increase in AGA could be achieved on asymmetric Camelyon17. It also surpassed the non-XAI-based correction methods in most scenarios.

P-ClArC's primary advantage is its computational efficiency, positioning it as the least resource-intensive of the evaluated XAI-based approaches. Computing the CAV for the confounder took only seconds when using a P-CAV, even on a CPU. Training a linear SVM to use its weight vector as a CAV could take slightly longer, as the time it takes for the classifier to converge depends on the linear separability of the sample representations and the layer from which they are extracted. Activations from deeper model layers are usually lower-dimensional, so using them as latent representations can increase convergence speed. In

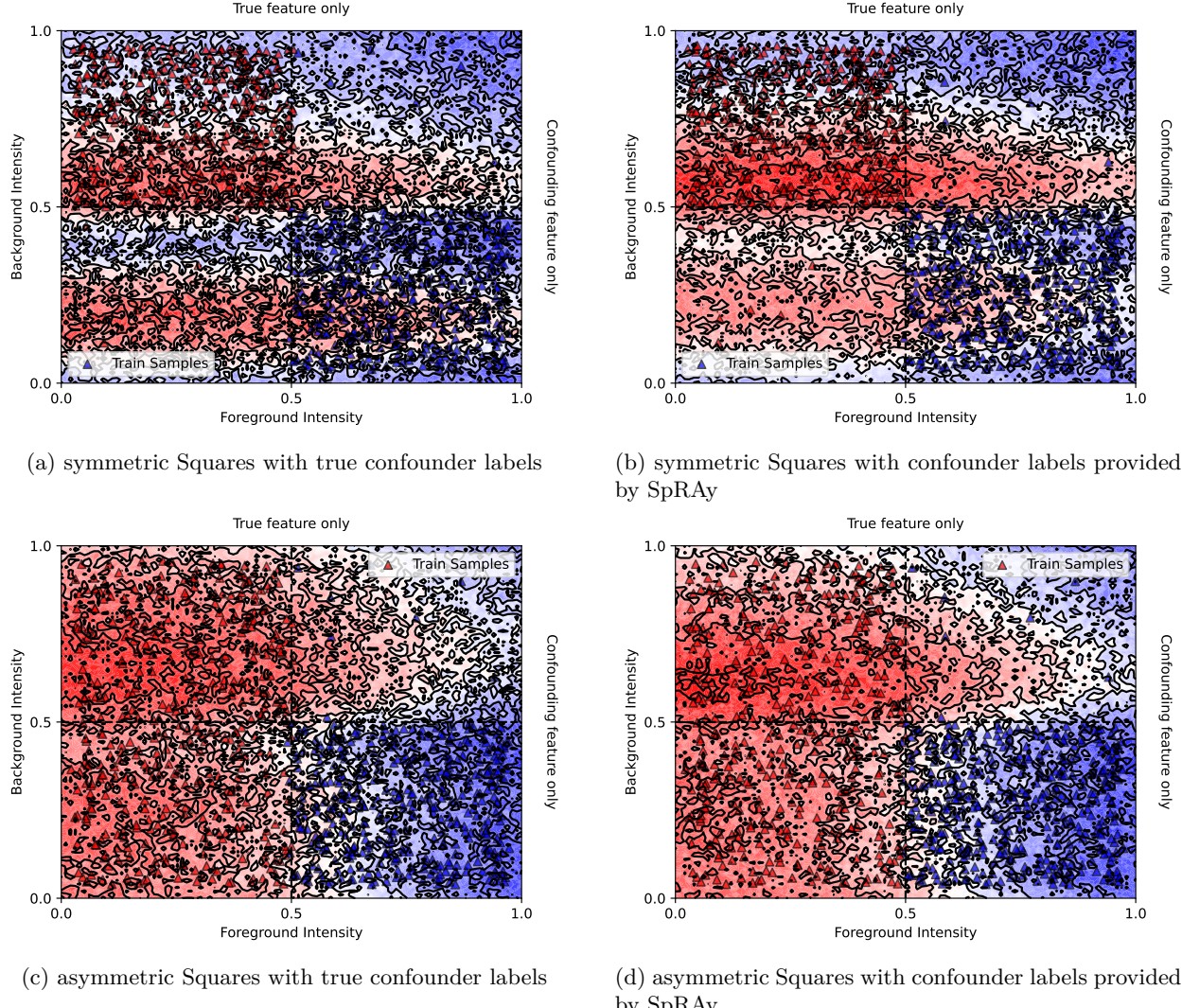

(a) symmetric Squares with true confounder labels

(b) symmetric Squares with confounder labels provided by SpRAy

(c) asymmetric Squares with true confounder labels

(d) asymmetric Squares with confounder labels provided by SpRAy

Figure 19: Decision boundaries and confidence regions for Squares after applying **Group DRO** to the student. The optimal decision boundary corresponds to a vertical line at 0.5 foreground intensity. Panels (a) and (b) depict results for symmetric Squares using true confounder labels and SpRAy-derived labels, respectively. Panels (c) and (d) show the corresponding results for asymmetric Squares.

our experiments, training the SVM was also completed in a few seconds in most cases. After the CAV was computed and the projection layer was integrated into the model, the final step involved fine-tuning to maximize the average group accuracy on the validation data. For our datasets, this process was quick as well, never taking more than 25 epochs and typically finishing in one to two minutes. The rapid convergence also allowed us to conduct extensive grid searches to identify the optimal hyperparameters within a practical timeframe. The hyperparameters we optimized included the projection layer $l \in \{0, 1, \ldots, 12\}$, the target class $c = \{A, B\}$ from which samples are used to compute the CAV, and the CAV type, selecting between a linear SVM-based vector ($v_{\mathrm{SVM}}$) and a PCAV ($v_{\mathrm{pat}}$).

As expected, PCAVs were superior to SVM-based CAVS most of the time. Regarding the CAV mode, we found that applying no reduction generally led to the largest improvements, although the difference to the results acquired with max pooling or mean aggregation was usually insignificant. Among the hyperparameters, the choice of the projection layer $l$ was the most important. Drawing a parallel with the observations from applying SpRAy, our intuition was that simpler confounders would be best addressed in shallower layers,

whereas more complex concepts would require choosing deeper layers to achieve the best results. Figure 20 illustrates the effect of the chosen layer $l$ on the AGA of the corrected model, both for the validation and the test set. For this analysis, we fixed layer $l$ and tuned the other hyperparameters, selecting the best model for each layer based on its performance on the validation data (see Appendix A for a layer overview). The results for the Squares dataset, which features the simplest confounder, support our assumption: for both the symmetric and asymmetric versions, optimal performance was achieved when setting $l = 1$, meaning the projection was most effective when applied directly after the first convolutional layer (cf. Figures 20a and 20b).

However, for CelebA Blond, the results did not align with the expectation that deeper layers would be optimal. In the symmetric version (Figure 20c), the AGA on the test data was relatively stable across most projection locations, with a notable performance drop at the shallowest ($l = 1$) and two deepest ($l = 11$, $l = 12$) layers. For the asymmetric version of CelebA Blond (Figure 20d), the test performance peaked

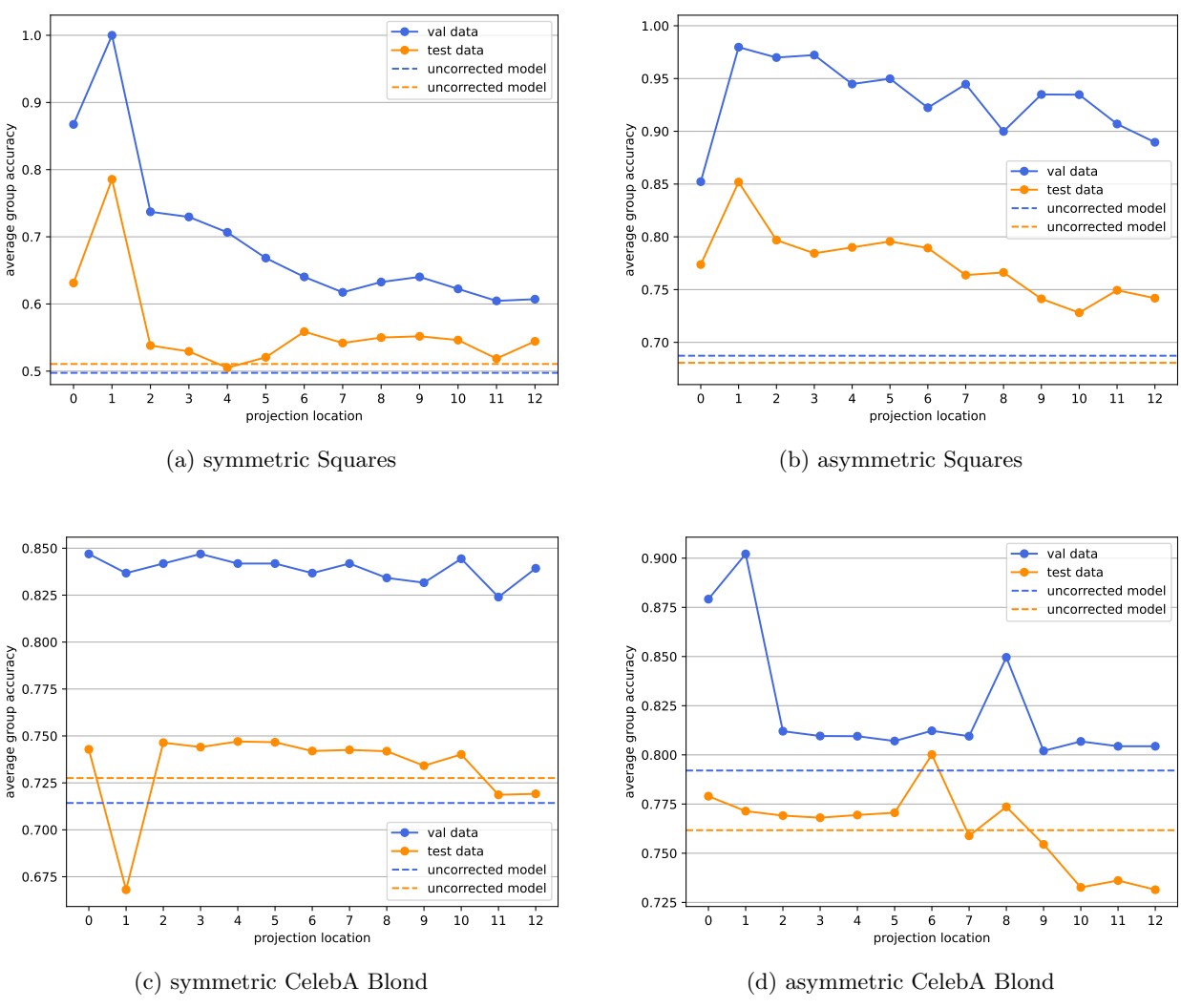

(a) symmetric Squares

(b) asymmetric Squares

(c) symmetric CelebA Blond

(d) asymmetric CelebA Blond

Figure 20: Impact of projection location $l$ on validation and test performance when correcting with P-ClArC. Each data point shows the validation and test AGA for the optimal model at a given layer $l$, selected based on the highest validation AGA. The layer $l$ indicates where the projection was inserted into the architecture, with $l = 0$ signifying a projection directly on the input image. The charts highlight a significant divergence between validation and test AGA, demonstrating that validation performance is an unreliable estimator for true generalization in our setting.

at an intermediate layer ($l = 6$) and was lowest when the projection was applied at the three deepest positions ($l = 10$, $l = 11$, $l = 12$). Figure 20 also reveals that the optimal projection location can differ substantially, even between dataset versions from the same domain with an identical confounder. While for symmetric CelebA Blond, the highest average group accuracy was achieved with the projection at $l = 4$, for the asymmetric version, the optimal location shifted to $l = 6$. Moreover, applying the projection at $l = 1$ yielded a small improvement for the asymmetric dataset but was the worst-performing option for the symmetric version, causing a noticeable AGA decrease.

This analysis of the layer-wise correction performance indicates that finding the optimal layer $l$ for computing the CAV and applying the projection remains challenging. Even with prior knowledge about the underlying dataset and the nature of the confounder, making educated guesses for finding a value for $l$ is highly unreliable, making costly grid search unavoidable to exploit P-ClArC's full potential.

Furthermore, it becomes apparent that in our experimental setting, the AGA on the validation data serves as a poor estimator for the model's true generalization ability on the test data. Across the board, the validation performance of the corrected models significantly overestimated the final test performance. Some degree of overestimation is expected, since we perform model selection based on validation AGA. Particularly for datasets with more complex confounders, however, the validation metric was not just optimistic, but sometimes even misleading about which hyperparameters were optimal. For asymmetric CelebA Blond, the test performance was highest at $l = 6$, but since the validation performance peaked at $l = 1$, our selection criteria led us to choose a sub-optimal model that generalized more poorly (see Figure 20d). In contrast, for the simpler Squares dataset, the validation AGA peaked at the same projection location as the test AGA ($l = 1$), allowing us to select the best model, as can be seen in Figures 20a and 20b.

The discrepancy between validation and test performance likely stems from the extreme data scarcity in the validation set. For all datasets except Follicles, the minority groups are represented by only two samples. Such a small number is often insufficient to capture the full data distribution of a group, a challenge that intensifies with feature complexity and that we also observed for DFR and Group DRO. For a dataset like Squares, where features are simple, these two samples might offer a reasonable proxy for test performance, though they are unlikely to cover edge cases. However, for a complex confounder like gender in CelebA Blond, it is highly improbable that two examples can represent the diversity of relevant attributes. This sparsity also elevates the risk of overfitting during model selection. An extensive hyperparameter search may produce a model that correctly classifies the two minority samples purely by chance, leading to its selection despite poor generalization. Conversely, a genuinely robust model could be discarded for failing to classify one of these samples, which is especially likely if the respective sample is an outlier. Because the coverage of the minority groups within the validation set is so low, validation AGA becomes an unreliable performance metric, potentially leading to the selection of models that are poorly generalized.

This issue of data scarcity also affects the computation of the CAV. P-ClArC's success heavily relies on a precise CAV to isolate and suppress the confounding feature. For this, we assume that the confounder can be represented by a linear direction in the model's latent space, orthogonal to causal concepts. While this assumption may not always hold in general, our setting makes it even more difficult to satisfy, since the CAV must be derived from only eight minority group samples in the training split. An unrepresentative sample set can produce an inaccurate CAV that is misaligned with the true confounder direction or even entangled with causal features. As a result, P-ClArC's ability to neutralize the bias without corrupting important causal signals could be decreased.

The described challenges are further intensified by potential mislabeling from SpRAy. Given the small size of the minority groups, each sample has considerable influence on the quality of the CAV, the average group accuracy, and, consequently, model selection. While the limited sample size already restricts the minority group's representativeness, incorrect labels from SpRAy can render the samples highly unrepresentative of their respective data group. The combined effect of low representativeness and high impact on the selection metric makes it substantially harder to produce and identify a model that genuinely generalizes well. Because of that, it is surprising that even when using SpRAy labels that are partially highly inaccurate, P-ClArC still always led to an increased AGA compared to the uncorrected model. For Follicles, test performance using SpRAy labels even surpassed test performance using the true labels. This is likely due to the problems during

model selection explained above: Using true labels, we achieved the highest AGA on the validation data with $l = 8$ using a SVM-based CAV. On the test data, this model yielded an AGA of 76.6%. If we instead selected a model with a PCAV projection and otherwise identical hyperparameter values, the resulting AGA on the test data would have been noticeably higher at 81.1%, surpassing the result for SpRAy labels. Consequently, the high variance in our model selection process posed the greatest challenge for P-ClArC. However, due to the lack of a better estimator for test AGA, selecting based on validation AGA remained the only option.

As before, we compute the confidence scores and decision boundaries for the corrected models on Squares (Figure 21). For both symmetric and asymmetric Squares, P-ClArC yields a noticeable improvement, correctly classifying a considerable portion of the minority groups that the original student failed on. Unlike the corrections made by Group DRO, these gains are achieved without degrading confidence or accuracy for the majority groups. This indicates that, at least for Squares, the few minority group samples in the training set were sufficient to compute a mostly accurate CAV for the confounder. However, the corrected

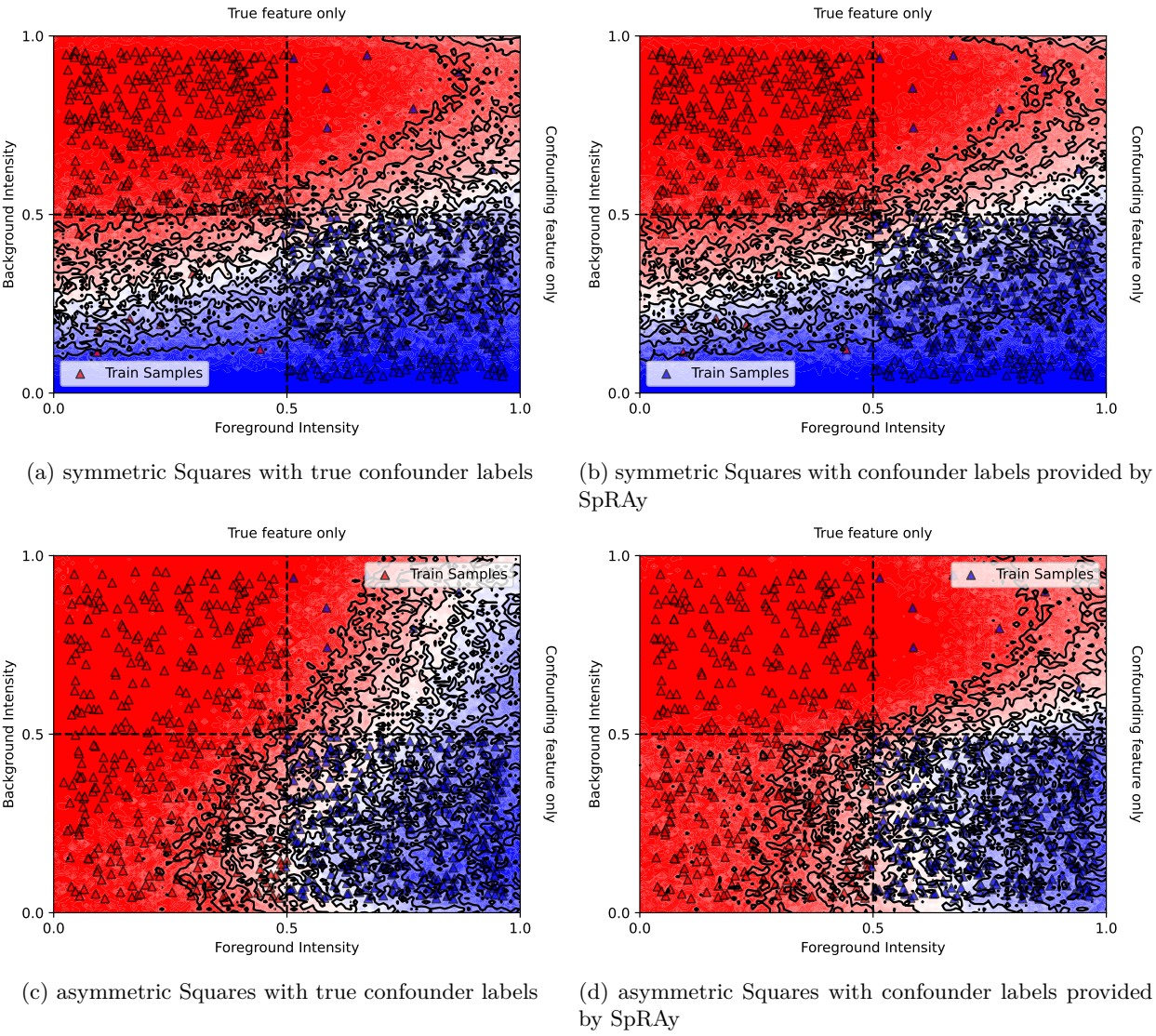

(a) symmetric Squares with true confounder labels

(b) symmetric Squares with confounder labels provided by SpRAy

(c) asymmetric Squares with true confounder labels

(d) asymmetric Squares with confounder labels provided by SpRAy

Figure 21: Decision boundaries and confidence regions for Squares after applying **P-ClArC** to the students. The optimal decision boundary corresponds to a vertical line at 0.5 foreground intensity. Panels (a) and (b) depict results for symmetric Squares using true confounder labels and SpRAy labels, respectively. Panels (c) and (d) show the corresponding results for asymmetric Squares.

decision boundaries still fall short of the theoretical optimum. The visualization also highlights the inconsistent results for asymmetric Squares, where the improvement with SpRAy labels is much smaller than with true labels, despite the near-perfect accuracy of the SpRAy annotations. This discrepancy is most likely an additional indicator that the flawed model selection process was a significant challenge in our experiments.

## 4.5 RR-ClArC

In our comparative analysis, RR-ClArC mostly achieved satisfactory results, usually ranking second behind CFKD and outperforming the other correction methods, especially when using ground-truth group labels (cf. Tables 2 and 3). A key advantage of RR-ClArC is its direct approach to mitigation. By incorporating a penalty term into the objective function, it explicitly minimizes the model's sensitivity to the confounding feature (given an accurate CAV). This contrasts with P-ClArC, which modifies sample representations with a projection layer rather than actually altering the student's behavior. Additionally, RR-ClArC supports class-specific unlearning, offering more granular control than the class-agnostic projection of P-ClArC. However, this particular benefit was not relevant in our experimental setup, which only involved binary classification tasks where the spurious correlation affected both classes.

In terms of computational overhead, a single epoch of fine-tuning with RR-ClArC was only moderately more expensive than a standard ERM epoch. The additional cost stems from the partial backpropagation required to compute the $L_{\mathrm{RR}}$ loss term, which increases epoch time by approximately 10% to 100% depending on the depth of the selected layer $l$. While this per-epoch increase aligns with the observations of Dreyer et al. (2024), the total training time in our experiments was substantially longer. Unlike the fixed 10-epoch fine-tuning schedule used by Dreyer et al., we fine-tuned until validation AGA converged, which took over 200 epochs in some cases, compared to a maximum of 25 for P-ClArC. This extended convergence time, in addition to the larger computational cost of a single epoch, makes hyperparameter tuning considerably more expensive than for P-ClArC. As the optimal hyperparameters values varied widely across datasets, an extensive and therefore costly grid search was unavoidable to identify the best-performing model configuration. This challenge was intensified by the fact that RR-ClArC introduces additional hyperparameters requiring optimization: the regularization strength $\lambda \in [0.1, 10^7]$ and the choice of loss function type (i.e. squared dot product or cosine similarity).

RR-ClArC was also susceptible to the same unreliable model selection process as P-ClArC, where validation AGA was often a poor proxy for test performance. To analyze this, we selected the top 20 corrected models for symmetric Squares based on their validation AGA and evaluated their corresponding test performance. Unlike the per-layer analysis used for P-ClArC, this approach accounts for the fact that other hyperparameters beyond layer $l$ can cause vastly different outcomes with RR-ClArC. The results for symmetric Squares, shown in Figure 22, exhibit a pattern similar to the P-ClArC evaluation. Although validation performance consistently overestimates test performance, the two metrics generally follow the same trend. This correlation suggests that, at least for simpler datasets like Squares, selecting the model with the highest validation AGA remains a viable strategy for identifying one of the better-generalizing configurations. For symmetric Squares with true confounder labels, the top three models identified by the grid search shared an identical validation AGA (Figure 22a). Their test AGAs were therefore averaged for reporting in Table 2. The outcome with SpRAy labels was stronger, as the model with the highest validation score also happened to achieve the best test performance (Figure 22b).

The unreliability of the model selection is aggravated by inaccurate SpRAy labels. This effect is illustrated by the results for symmetric CelebA Smiling (Figure 23). When using true confounder labels, the validation AGA successfully guides the selection to a model that also performs well on the test data (Figure 23a). However, with SpRAy labels, the test performance of the top 20 models widely varied, which was not at all reflected by their respective validation performances (Figure 23b). Consequently, the model with the highest validation AGA ultimately performed worse than P-ClArC and Group DRO. However, it still improved over the uncorrected student, and even outperformed DFR (cf. Table 3). Interestingly, this failure cannot be attributed to a suboptimal CAV derived from the noisy labels. The existence of another well-performing model within the top candidates (model #12), which would have achieved a similar test AGA as the best model with true group labels, proves that an effective correction was possible, despite noisy training labels. The main issue was therefore the inaccurately labeled minority samples in the validation set, which provided

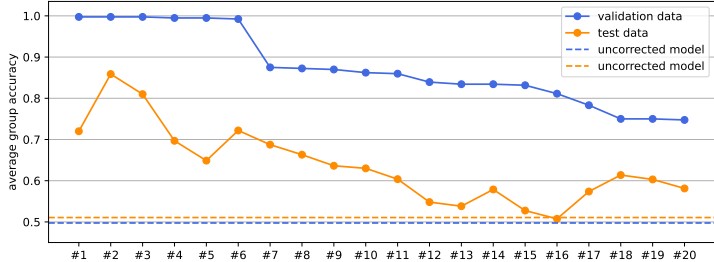

(a) RR-ClArC correction on symmetric Squares with true confounder labels

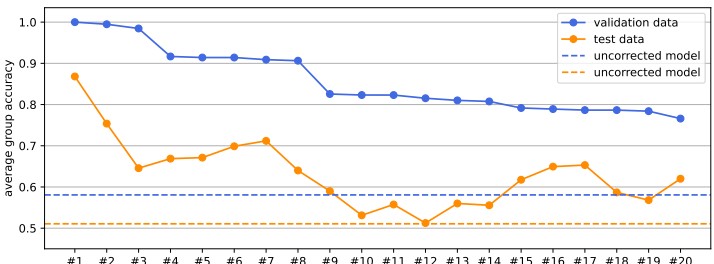

(b) RR-ClArC correction on symmetric Squares with SpRAy confounder labels

Figure 22: Comparison of the validation and test AGAs on symmetric Squares for the top 20 models, which were selected from a grid search based on their validation performance. The results are shown for corrections using (a) true confounder labels and (b) confounder labels provided by SpRAy.

a misleading signal and caused the selection of a poorly generalized model even when a better alternative exists.

An even more extreme example of the problems during model selection is provided by the Follicles dataset (Figure 24). Even with true confounder labels, the link between validation and test performance is fragile. Models with similar validation scores can differ noticeably regarding their test AGAs, with some even underperforming the uncorrected student (Figure 24a). Nevertheless, the model achieving the highest validation AGA was also the one that generalized best, and the three top-ranked models based on validation performance yield higher test scores on average. Hence, the selection process was still effective. However, when using SpRAy labels with Follicles, any meaningful relationship between validation and test AGA is lost, rendering the selection process basically as arbitrary as choosing a model at random. (Figure 24b).

It is plausible that SpRAy might have identified an entirely different spurious correlation, which would explain both the strong mismatch between validation and test scores and the low accuracy of the SpRAy labels (cf. Table 4). However, we cannot confirm or rule out this suspicion, as we possess no knowledge about the nature of this potential new confounder, and no ground-truth labels are available for the resulting data groups. Further investigations would be out of scope for this study. Nevertheless, during exploration with Virelay, the clusters seemed to be roughly distinguished by follicle sizes, i.e. the intended confounder.

Beyond the challenges in model selection, our experiments revealed a lack of robustness in the RR-ClArC method itself. We found that performance is highly sensitive to hyperparameter configurations, where minor adjustments can cause drastic changes in both validation and test AGAs. This sensitivity makes it difficult to establish a reliable starting point for tuning, as optimal configurations are not transferable between datasets. A set of hyperparameters that performs well in one context may fail completely in another. This problem persists even between closely related setups, such as the symmetric and asymmetric versions of the same dataset, and raises concerns about the reproducibility of results obtained with RR-ClArC.

Despite these difficulties, RR-ClArC is capable of producing results comparable to CFKD and superior to P-ClArC, Group DRO, and DFR when optimally tuned. Nevertheless, this high potential is consistently

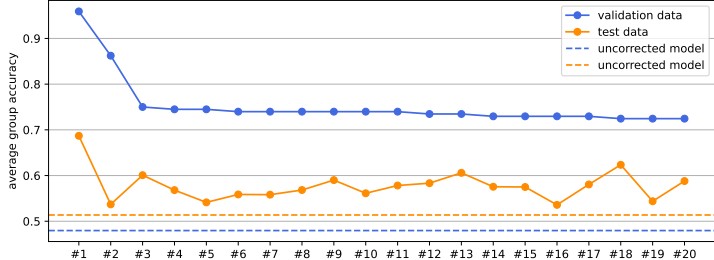

(a) RR-ClArC correction on symmetric CelebA Smiling with true confounder labels

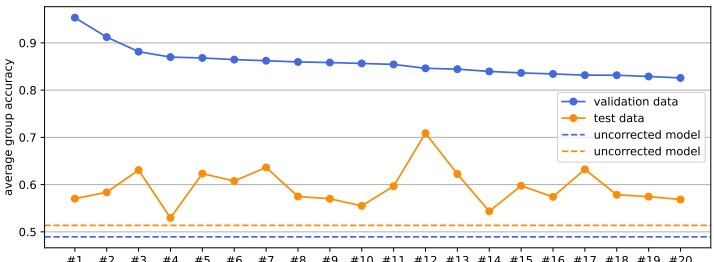

(b) RR-ClArC correction on symmetric CelebA Smiling with SpRAy confounder labels

Figure 23: Comparison of the validation and test AGAs on symmetric CelebA Smiling for the top 20 models, which were selected from a grid search based on their validation performance. The results are shown for corrections using (a) true confounder labels and (b) confounder labels provided by SpRAy.

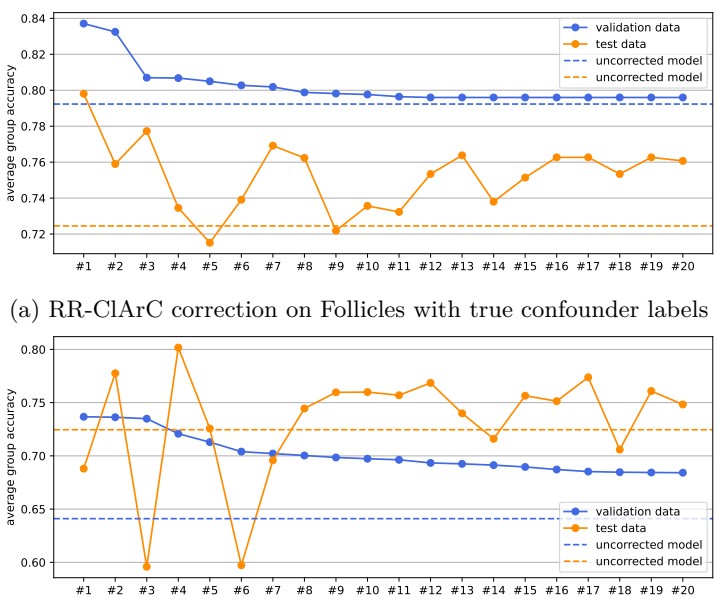

(a) RR-ClArC correction on Follicles with true confounder labels

(b) RR-ClArC correction on Follicles with SpRAy confounder labels

Figure 24: Comparison of the validation and test AGAs on Follicles for the top 20 models, which were selected from a grid search based on their validation performance. The results are shown for corrections using (a) true confounder labels and (b) confounder labels provided by SpRAy.

undermined by the primary challenge of unreliable model selection. This limitation makes RR-ClArC difficult to apply with confidence in scenarios defined by scarce data and extreme minority group underrepresentation, which were central to this study. Although Group DRO and DFR were also tuned using validation AGA and faced the same challenge, they circumvented this problem due to their simpler hyperparameter spaces, which did not require extensive grid searching. In other words, with fewer configurations to choose from, the probability of selecting a model that coincidentally fits the few minority samples in the validation set while still exhibiting severe Clever Hans behavior was significantly reduced. This probably helped Group DRO in particular to produce much more consistent results.

Once again, the decision boundaries of the Squares models after correction with RR-ClArC are visualized in Figure 25. The plots clearly show that RR-ClArC provides a more effective correction than the previously examined methods, with a larger portion of minority group samples now being correctly classified. Similar to P-ClArC, this is achieved without compromising the model's performance on majority groups. Notably,

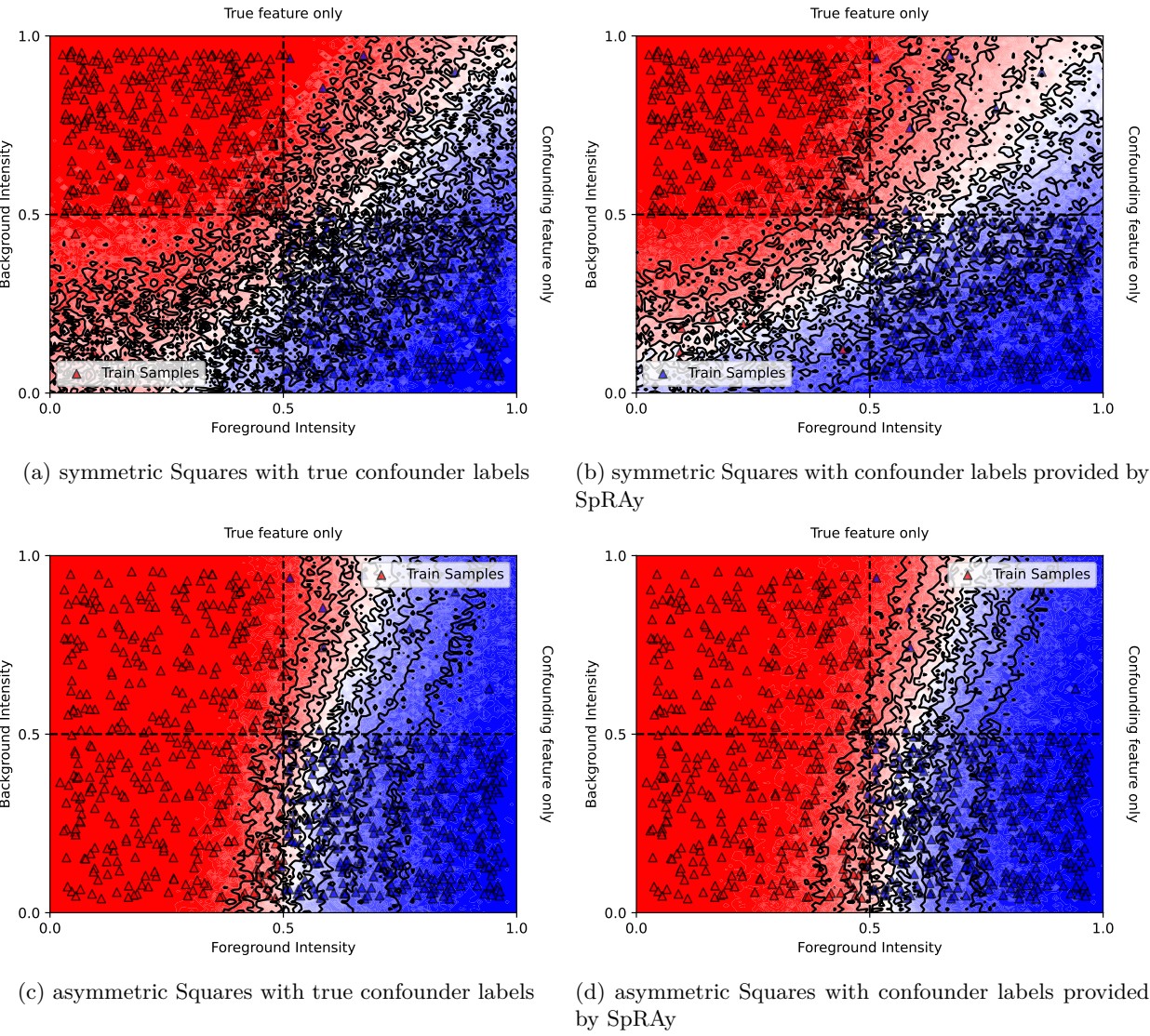

(a) symmetric Squares with true confounder labels

(b) symmetric Squares with confounder labels provided by SpRAy

(c) asymmetric Squares with true confounder labels

(d) asymmetric Squares with confounder labels provided by SpRAy

Figure 25: Decision boundaries and confidence regions for Squares after applying **RR-ClArC** to the students. The optimal decision boundary corresponds to a vertical line at 0.5 foreground intensity. Panels (a) and (b) depict results for symmetric Squares using true confounder labels and SpRAy labels, respectively. Panels (c) and (d) show the corresponding results for asymmetric Squares.

the decision boundary is also visibly smoother, and the regions of low model confidence are smaller and more constrained compared to the other methods. For the asymmetric case in particular, the corrected model's decision boundary nearly follows the theoretical optimum. This result again highlights RR-ClArC's potential, although it should be noted that asymmetric Squares was a dataset on which RR-ClArC performed particularly well (cf. Tables 2 and 3).

## 4.6 CFKD

Of all the correction methods evaluated, both XAI-based and non-XAI-based, CFKD most consistently delivered the best results. Specifically, for six of the nine datasets used in our experiments, the model corrected with CFKD achieved the highest average group accuracy. In two of the remaining three cases, it ranked second-best. And while it achieved the largest improvements on the simple Squares, it also produced high-quality results on the more complex Follicles dataset and even on CelebA Blond, where all other methods failed to achieve significant improvements over the uncorrected student (cf. Tables 2 and 3). This success highlights the advantage of CFKD's data augmentation strategy. By directly increasing the number of minority group samples, it strengthened model generalization to a degree the other approaches often could not match.

However, there is one notable outlier: for the asymmetric version of the Camelyon17 dataset, the corrected model achieved the lowest average group accuracy, with a test AGA that is (after rounding) identical to the uncorrected student. This is somewhat surprising given its otherwise strong results, as CFKD significantly improved generalization for symmetric Camelyon17. In general, a plausible explanation for CFKD might fail is a poor quality of the generated counterfactual explanations, as the subsequent fine-tuning on the augmented dataset is relatively trivial, with not much potential to fail. However, without expert-level domain knowledge, Camelyon17 is arguably the most difficult dataset for which to judge whether a generated counterfactual lies within the data distribution and whether the student and oracle models have classified it correctly. There is no ground-truth information available about what a specific counterfactual should ideally look like, or what its real class label would be. This ambiguity makes it difficult to diagnose the precise reason for CFKD's failure and, consequently, to find a solution.

Figure 26 shows a selection of counterfactual explanations for the asymmetric Camelyon17 samples. Visually, they seem to resemble original examples and could plausibly belong to the original data distribution. We

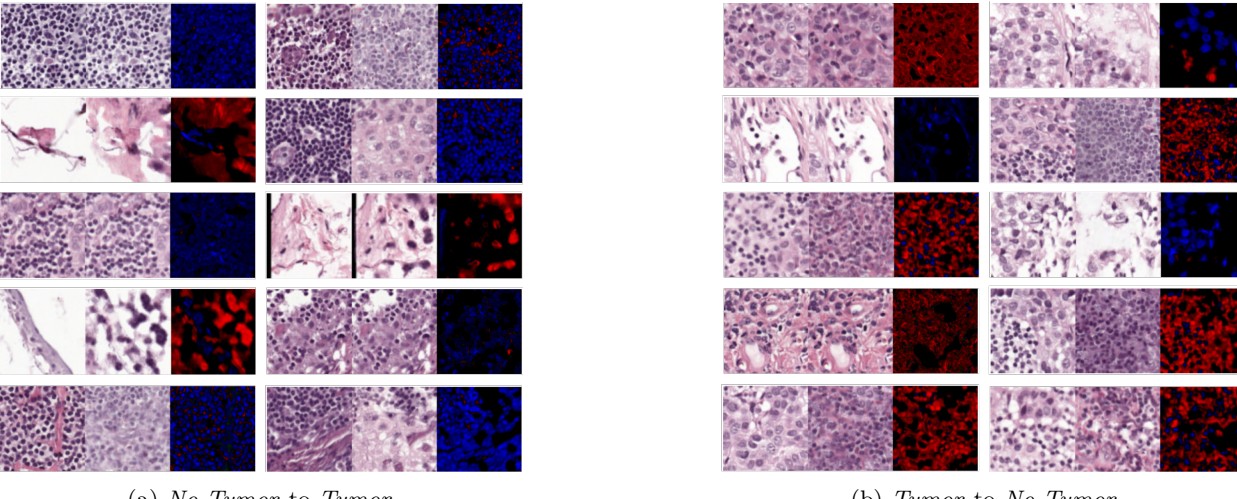

(a) *No Tumor* to *Tumor*     (b) *Tumor* to *No Tumor*

Figure 26: A selection of counterfactual explanations generated during the CFKD process for asymmetric Camelyon17. From left to right, each group shows the original image, its corresponding counterfactual, and the pixel-wise difference between the two. Panel (a) shows samples where the original was classified as *No Tumor* and the counterfactual as *Tumor* by the student, while panel (b) shows the reverse case.

can identify cases where the actual cell structure appears modified (true counterfactuals) and others where the changes primarily affect the color cast (false counterfactuals). While some changes are subtle and almost invisible to the human eye, these do not appear to be adversarial examples; the pixel-wise difference maps show targeted adaptations to the space between cells rather than seemingly random noise. It was therefore not possible for us to determine what might be wrong with the generated counterfactuals, leaving no clear path for hyperparameter tuning.

This challenge exemplifies a key drawback of CFKD: generating minimal yet meaningful counterfactuals is a non-trivial task. The absence of a ground truth for what an ideal counterfactual should be makes it difficult to judge their quality, a problem that is particularly acute in specialized domains like histopathology. Consequently, while CFKD performed well in most of our tests using default settings, resolving a failure case could require exhaustive and computationally expensive trial and error. This is further complicated by the vast hyperparameter space of the underlying counterfactual explainer (in this case, SCE) and the high cost of generating numerous explanations to uncover failure modes. For these reasons, even though we tried several different hyperparameter configurations, conducting an extensive grid search to find optimal values (as was done for P-ClArC and RR-ClArC) was not feasible.

As for the other methods, we again visualize the decision boundary and confidence scores for the Squares models after they have been corrected with CFKD (Figure 27). The visualizations align well with the results from Table 2, demonstrating that CFKD effectively eliminates the Clever Hans effect for both symmetric and asymmetric Squares, with decision boundaries that closely approximate the theoretical optimum. And while RR-CIArC yielded a slightly higher AGA and WGA for asymmetric Squares, CFKD, in turn, led to an even smoother decision boundary with smaller regions of low confidence (cf. Figures 27b and 25c). CFKD heavily benefits from augmenting its training dataset with counterfactuals. Increasing the minority group population with false counterfactuals not only mitigated the spurious correlation but also helped prevent overfitting and visibly promoted a more well-shaped decision boundary.

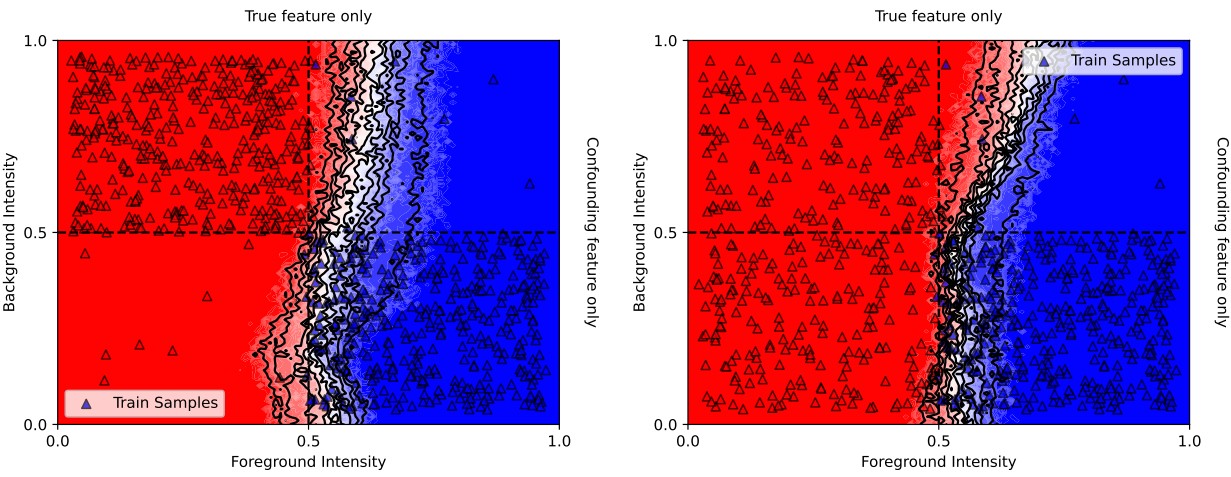

(a) symmetric Squares with true confounder labels

(b) symmetric Squares with confounder labels provided by SpRAy

Figure 27: Decision boundaries and confidence regions for Squares after applying **CFKD** to the students. The optimal decision boundary corresponds to a vertical line at 0.5 foreground intensity. Panels (a) and (b) depict results for symmetric and asymmetric Squares, respectively.

## 5    Conclusion and Outlook

This reproducibility study presented a comparative analysis of state-of-the-art methods designed to correct Clever Hans behavior in image classifiers, specifically under challenging conditions of limited data and severe spurious correlations. Our study centered on the recently proposed XAI-based correction methods P-ClArC, RR-ClArC, and CFKD, which have not yet been extensively and independently evaluated. We also included the well-established non-XAI methods DFR and Group DRO as baselines, whose effectiveness has already been demonstrated in more favorable settings with greater data availability and better minority group coverage. By evaluating both XAI-based and non-XAI-based approaches under these demanding conditions, this work offers insights into their practical effectiveness and inherent limitations.

A primary finding of our evaluation is that correction methods rooted in XAI generally outperformed the non-XAI baselines in mitigating model bias. Among them, CFKD proved to be the most consistently effective approach, achieving the highest AGA on the majority of datasets, even those with complex confounding features where other methods struggled. RR-CIArC also demonstrated strong potential, though its performance exhibited higher variance. In comparison, the non-XAI baselines, while simpler to implement, were typically unable to match the performance gains of the XAI techniques. Of the two, Group DRO was generally preferable to DFR, as its ability to use the entire training set and update all model parameters led to more substantial improvements, although this came at the cost of longer convergence times. Furthermore, our work again confirmed the inadequacy of empirical accuracy as a performance metric in the presence of strong dataset biases. Adopting group-aware metrics like AGA and WGA was essential for accurately diagnosing the Clever Hans effect and assessing correction quality.

Across our experiments, two central challenges consistently surfaced: the dependency on group labels and the instability of model selection in data-scarce environments. Many correction techniques, including DFR, Group DRO, and the ClArC family, require the availability of labels that identify the presence of confounding features in training and validation samples. Our attempt to automatically obtain group labels with SpRAy revealed its limitations. While effective for simple, salient confounders, SpRAy was unreliable for more complex features, often demanding extensive hyperparameter tuning, manual examination of the computed solution, and even manual clustering with the help of Virelay to yield usable results. In some cases, we were not able to produce usable results with SpRAy at all. The dependency on group labels thus represents a major hurdle to the practical deployment of these methods.

This difficulty was intensified by data scarcity, as our experimental design intentionally limited minority groups to only a few samples. The large size differences of the distinct data groups further impeded SpRAy's ability to accurately create clusters according to the different decision strategies. Moreover, with only a few minority instances in the validation set, the AGA became a high-variance and unreliable estimator for test performance, making the selection of the best model susceptible to chance and overfitting. This was especially evident after large hyperparameter searches for P-ClArC and RR-CIArC. However, the underrepresentation of minority groups in the training and validation data also limited the performance of Group DRO and, in particular, DFR. In this context, the typically superior performance of CFKD becomes understandable, as it avoids these problems by not requiring group labels and by creating additional minority group samples through counterfactual explanations.

Our experiments uncovered several other method-specific challenges. High-variance methods like RR-CIArC, for instance, suffer from questionable reproducibility, as their performance was extremely sensitive to hyperparameter configurations. Especially the choice of network layer $l$ strongly influenced the performance of P-ClArC, RR-ClArC, and SpRAy, with no one-size-fits-all solution for where to best extract feature representations or compute attribution maps. While the optimal layer was generally linked to the complexity of the confounding feature, it could still vary even between the two versions of the same dataset.

DFR, on the other hand, was often rendered inert by accuracy saturation. Here, the model had already overfitted the small fine-tuning set drawn from the training data, resulting in a negligible loss signal for correction. This highlights that DFR struggles to improve model behavior when no additional held-out data is available.

There also seems to be a trade-off between a method's performance and its computational complexity, as the top-performing method, CFKD, was also the most computationally expensive. And while its performance was generally stable, its failure on asymmetric Camelyon17 proved difficult to diagnose, because assessing the quality of the counterfactuals it relies on can be difficult, or even impossible, without deep domain expertise.

The findings of this study point to several promising avenues for future research to build more robust and practical solutions for correcting Clever Hans models. A critical area is the improvement of automated group discovery, either by further refining SpRAy or by developing alternative methods. Overcoming SpRAy's current limitations, particularly the need for extensive manual tuning and domain expertise, would broaden the applicability of many correction techniques. Another key direction is the development of more robust model selection criteria, as validation AGA proved unreliable in data-scarce settings. Specifically, metrics that are less sensitive to the small sample sizes of minority groups could be beneficial to increase the potential of existing correction methods.

Another possibility for improving current methods might be to explore their synergistic potential. For example, the fine-tuning stages of P-ClArC and RR-ClArC could be enhanced by incorporating Group DRO, especially since the necessary group labels already need to be available. Finally, a crucial direction is to develop methods that can detect and correct biases with minimal human supervision. All evaluated methods require an expert in the loop, either for interpreting SpRAy's outputs or for acting as a teacher in the CFKD framework. Future research that reduces this dependency could improve both scalability and ease of use.

Despite the comprehensive scope of this analysis, several limitations must be acknowledged, which offer additional next steps for future research. The datasets used in this study were intentionally small to emulate realistic data scarcity in high-stakes domains, which enabled controlled experimentation but reduced the statistical robustness of the results. Furthermore, the evaluation was restricted to image-based binary classification tasks, so the generalizability of the findings to other modalities (e.g. tabular or textual data) or multi-class problems remains uncertain. For methods requiring group labels, automatic clustering with SpRAy frequently failed, making manual clustering with ViRelAy necessary. This reliance on human judgment introduced a degree of subjectivity that may limit reproducibility. Moreover, all experiments were conducted using a single model architecture (ResNet-18), and a different architecture might have led to different results. Finally, each dataset contained only one confounding factor, whereas real-world scenarios often involve multiple interacting confounders. It remains an open question how this would have affected correction quality. Consequently, the reported results should be interpreted as indicative of practical trends rather than as absolute performance rankings.

Nevertheless, while significant challenges remain, the rapid evolution of methods to mitigate Clever Hans is highly encouraging. Continued progress in this field is an essential step toward the trustworthy deployment of machine learning in safety-critical domains, where robust, reliable, and interpretable models can provide substantial value to society.

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

## A    Detailed Model Architecture

```
layer 1: Conv2d(3, 64, kernel_size=(7, 7), stride=(2, 2), padding=(3, 3), bias=False)
layer 2: BatchNorm2d(64, eps=1e-05, momentum=0.1, affine=True, track_running_stats=True)
         ReLU(inplace=True)
layer 3: MaxPool2d(kernel_size=3, stride=2, padding=1, dilation=1, ceil_mode=False)
layer 4: BasicBlock(
  (conv1): Conv2d(64, 64, kernel_size=(3, 3), stride=(1, 1), padding=(1, 1), bias=False)
  (bn1): BatchNorm2d(64, eps=1e-05, momentum=0.1, affine=True, track_running_stats=True)
  (relu): ReLU(inplace=True)
  (conv2): Conv2d(64, 64, kernel_size=(3, 3), stride=(1, 1), padding=(1, 1), bias=False)
  (bn2): BatchNorm2d(64, eps=1e-05, momentum=0.1, affine=True, track_running_stats=True)
)
layer 5: BasicBlock(
  (conv1): Conv2d(64, 64, kernel_size=(3, 3), stride=(1, 1), padding=(1, 1), bias=False)
  (bn1): BatchNorm2d(64, eps=1e-05, momentum=0.1, affine=True, track_running_stats=True)
  (relu): ReLU(inplace=True)
  (conv2): Conv2d(64, 64, kernel_size=(3, 3), stride=(1, 1), padding=(1, 1), bias=False)
  (bn2): BatchNorm2d(64, eps=1e-05, momentum=0.1, affine=True, track_running_stats=True)
)
layer 6: BasicBlock(
  (conv1): Conv2d(64, 128, kernel_size=(3, 3), stride=(2, 2), padding=(1, 1), bias=False)
  (bn1): BatchNorm2d(128, eps=1e-05, momentum=0.1, affine=True, track_running_stats=True)
  (relu): ReLU(inplace=True)
  (conv2): Conv2d(128, 128, kernel_size=(3, 3), stride=(1, 1), padding=(1, 1), bias=False)
  (bn2): BatchNorm2d(128, eps=1e-05, momentum=0.1, affine=True, track_running_stats=True)
  (downsample): Sequential(
    (0): Conv2d(64, 128, kernel_size=(1, 1), stride=(2, 2), bias=False)
    (1): BatchNorm2d(128, eps=1e-05, momentum=0.1, affine=True, track_running_stats=True)
  )
)
layer 7: BasicBlock(
  (conv1): Conv2d(128, 128, kernel_size=(3, 3), stride=(1, 1), padding=(1, 1), bias=False)
  (bn1): BatchNorm2d(128, eps=1e-05, momentum=0.1, affine=True, track_running_stats=True)
  (relu): ReLU(inplace=True)
  (conv2): Conv2d(128, 128, kernel_size=(3, 3), stride=(1, 1), padding=(1, 1), bias=False)
  (bn2): BatchNorm2d(128, eps=1e-05, momentum=0.1, affine=True, track_running_stats=True)
)
layer 8: BasicBlock(
  (conv1): Conv2d(128, 256, kernel_size=(3, 3), stride=(2, 2), padding=(1, 1), bias=False)
  (bn1): BatchNorm2d(256, eps=1e-05, momentum=0.1, affine=True, track_running_stats=True)
  (relu): ReLU(inplace=True)
  (conv2): Conv2d(256, 256, kernel_size=(3, 3), stride=(1, 1), padding=(1, 1), bias=False)
```

```
      (bn2): BatchNorm2d(256, eps=1e-05, momentum=0.1, affine=True, track_running_stats=True)
      (downsample): Sequential(
        (0): Conv2d(128, 256, kernel_size=(1, 1), stride=(2, 2), bias=False)
        (1): BatchNorm2d(256, eps=1e-05, momentum=0.1, affine=True, track_running_stats=True)
      )
    )
    layer 9: BasicBlock(
      (conv1): Conv2d(256, 256, kernel_size=(3, 3), stride=(1, 1), padding=(1, 1), bias=False)
      (bn1): BatchNorm2d(256, eps=1e-05, momentum=0.1, affine=True, track_running_stats=True)
      (relu): ReLU(inplace=True)
      (conv2): Conv2d(256, 256, kernel_size=(3, 3), stride=(1, 1), padding=(1, 1), bias=False)
      (bn2): BatchNorm2d(256, eps=1e-05, momentum=0.1, affine=True, track_running_stats=True)
    )
    layer 10: BasicBlock(
      (conv1): Conv2d(256, 512, kernel_size=(3, 3), stride=(2, 2), padding=(1, 1), bias=False)
      (bn1): BatchNorm2d(512, eps=1e-05, momentum=0.1, affine=True, track_running_stats=True)
      (relu): ReLU(inplace=True)
      (conv2): Conv2d(512, 512, kernel_size=(3, 3), stride=(1, 1), padding=(1, 1), bias=False)
      (bn2): BatchNorm2d(512, eps=1e-05, momentum=0.1, affine=True, track_running_stats=True)
      (downsample): Sequential(
        (0): Conv2d(256, 512, kernel_size=(1, 1), stride=(2, 2), bias=False)
        (1): BatchNorm2d(512, eps=1e-05, momentum=0.1, affine=True, track_running_stats=True)
      )
    )
    layer 11: BasicBlock(
      (conv1): Conv2d(512, 512, kernel_size=(3, 3), stride=(1, 1), padding=(1, 1), bias=False)
      (bn1): BatchNorm2d(512, eps=1e-05, momentum=0.1, affine=True, track_running_stats=True)
      (relu): ReLU(inplace=True)
      (conv2): Conv2d(512, 512, kernel_size=(3, 3), stride=(1, 1), padding=(1, 1), bias=False)
      (bn2): BatchNorm2d(512, eps=1e-05, momentum=0.1, affine=True, track_running_stats=True)
    )
    layer 12: AdaptiveAvgPool2d(output_size=(1, 1))
    layer 13: Linear(in_features=512, out_features=2, bias=True)
```

