# OpenReview forum: "Reproducibility study on how to find Spurious Correlations, Shortcut Learning, Clever Hans or Group-Distributional non-robustness and how to fix them"
_TMLR — Under review for TMLR_

### Review · Reviewer_kpex · 2026-06-19

**Summary Of Contributions:**

The submission is a reproducibility study of methods for correcting Clever Hans/shortcut learning behavior in image classifiers under data scarcity, strong spurious correlations and severe subgroup imbalance. It compares the XAI-based methods P-ClArC, RR-ClArC, and CFKD, against DFR and Group DRO, and shows that in this setting the XAI-based method CFKD often achieves the strongest improvements. An important contribution the finding that practical deployment is bottlenecked by unreliable group-label discovery, unstable validation-based model selection and the reliance of an expert in the loop across all considered methods.

Strengths:
- the paper focuses on shortcut learning under realistic constraints such as scarce data, severe imbalance, and missing group labels
- discussion of limitations around Spray, unreliable hyperparameter tuning, supervision reliability and method-specific failure modes

Weaknesses:
- missing statistical evaluation/uncertainty estimation; especially in the studied low data setting findings might to some extednd be sensitive to e.g. data subsampling (only 10 minority samples per dataset) or initialization
- single architecture and single confounding factor
(both these weaknesses are already acknowledged in the limitations paragraph)

**Audience:**

Yes

**Audience Explanation:**

The paper would be specifically interesting to researchers working on robustness, shortcut learning, domain generalization, XAI, data bias, and empirical evaluation methodology. In my view, especially the practical findings and conclusions beyond the direct model comparisons are valuable.

**Claims And Evidence:**

Yes

**Claims Explanation:**

The submission provides convincing evidence for its practical findings, especially regarding Spray’s limitations and the instability of validation-based model selection. It also provides meaningful evidence for the broader comparative claim that XAI-based correction methods generally outperform the considered non-XAI baselines. However, "generally" here should be understood as referring to the specific data-scarce setting studied in this paper, rather than as a universal claim. Since CFKD's performance is largely stable across datasets, I do not think its strong performance can be dismissed as chance alone. Nevertheless, the evidence is weakened by the absence of a statistical evaluation and by the fact that both non-XAI baselines were applied in settings that may be somewhat disadvantageous to them. Moreover while computational effort (e.g. CFKD as the most demanding method) is mentioned multiple times in the text, the paper does not provide a systematic comparison of compute across methods. Overall, I answer this question with "yes", while noting that there are several straightforward ways to strengthen the provided evidence, and thereby the paper, as listed below.

**Requested Changes:**

Would strengthen the work:
- statistical evaluation of the presented results with regards to data subsampling, random initialization etc
- comparison to group DRO from scratch and DFR with held-out data to allow for assessing to which extend they are limited from these choices
- an comparison table of compute budget across methods
- an overview/comparison table during the method review on aspects like the type of necessary human supervision and/or the reliance on group labels
- addition of another architecture

Minor:
- Caption of figure 27 is contradicting; as cfkd is not using group labels I assume the panel captions should be changed
- In the paragraph on DFR implementation the sentence starting 'Since all ...' 'exact them students' should be 'exact same students'

---

### Review · Reviewer_hSd9 · 2026-07-12

**Summary Of Contributions:**

In this article, the authors survey and evaluate different explainable AI (XAI) based methods to identify and correct the "Clever Hans" effect, the exploitation of spurious correlations by machine learning models and its negative impact on generalization.

The authors first motivate and discuss the necessity of this mitigation, then introduce and discuss several techniques to perform this mitigation. Some of these depend on labels for potentially confounding effects, so the authors further discuss and evaluate Spectral Relevance Analysis (SpRAy) as a technique to automatically generate such labels. The authors evaluate their choice of methods on a broad spectrum of datasets, both synthetic and real and of different complexity.

The authors find that generally, more advanced XAI-based techniques outperform the simpler alternatives, especially Counterfactual Knowledge Distillation (CFKD). The authors also find the quality of SpRAy-generated confounder labels to deteriorate with the complexity of the confounder, further strengthening the case for CFKD, which does not rely on such labels.

**Audience:**

Yes

**Audience Explanation:**

Understanding when models rely on confounders rather than causal features, and which correction techniques actually address this, is a question of broad interest across the community.

**Claims And Evidence:**

Yes

**Claims Explanation:**

I cannot evaluate the choice of methods nor their technical explanation, as XAI is not my primary field of expertise. My assessment focuses on the motivation, the experimental design, and the conclusions drawn from the results.

The motivation is good and clearly stated. The dataset construction is overall quite good, the CelebA Smiling and Follicles cases in particular, and the spectrum of complexity across datasets is especially strong. The figures throughout the paper are all quite good, and Figure 1 works well as an introductory figure. The choice of metrics is good and well argued, and the discussion is detailed for each method. On balance the claims are well supported. Two points weaken the strength of the evidence, and a third is less a problem than a missed opportunity.

First, some claims are stated as settled without references. The statement that "In images, they often rely on low-level signals such as background textures, color casts, watermarks, or other dataset-specific artifacts while ignoring more complex but semantically relevant features" is asserted without support, as is the claimed simplicity bias of models. The latter is especially important as later arguments depend on it. In particular, Figure 1 panel A can be misleading at first, especially to readers who are not experts on this subject: one might ask why the confounder should push the decision boundary such that many samples are misclassified, if both the causal feature and the confounder are detectable. Figure 13 later shows the depiction to be accurate, but it comes much later, and the argument in the meantime rests on the simplicity bias point that is not well established. Panel D is also difficult to understand at the point where it is introduced.

Second, the experimental design leaves some room for noise. Some sample sizes are so small that multiple runs, seed variation, and similar measures are really needed to separate signal from noise. Using small datasets is fine, as the authors argue, but small datasets make a robust design critical. The authors further argue that sample size matters, yet it is unclear why the datasets are not sampled to a common size, as the different sample sizes themselves might introduce a distortionary effect on the cross-dataset results. This effect could then be investigated directly, but the authors neglect to do this, and it might distort the choice of datasets as a spectrum of task difficulty. The choice of model also seems likely to have a large effect, particularly on the simplicity bias, and is only briefly acknowledged. The train accuracies are a little weird: I would expect essentially all of them to be at or near 100. Follicles is understandable, but it is not clear why Smiling symmetric is so much worse than Smiling asymmetric.

Third, the conclusions do not fully use the setup. There is no serious discussion of the spectrum of complexity across the tasks, despite an interesting pattern: overall the more advanced the correction technique the better it performs, but on the most complex datasets the simpler methods seem to catch up and sometimes even outperform. In general, the authors' setup had more potential than they actually used.

**Requested Changes:**

The most valuable change would be a serious discussion of the spectrum of complexity already present in the results, in particular the pattern that advanced correction methods lead on the simpler tasks while simpler methods catch up or even overtake on the most complex ones. This is arguably among the paper's most interesting findings and it is currently left largely unexamined. Relatedly, discretizing the continuous confounders is fine for the authors' purposes, but it forecloses a deeper investigation, such as creating a spectrum of confounder strength and looking for a spectrum of effects rather than a binary one; a binary presence or absence of a watermark for CelebA Smiling would similarly avoid the discretization step entirely.

Second in importance are the experimental design points around the small datasets. The robustness of these experiments should be strengthened through multiple seeds and reported variation, and the datasets should either be sampled to a common size or the effect of sample size should be investigated directly.

Beyond that:

1. Add the missing citations, in particular for the reliance on low-level signals and for the simplicity bias, since later arguments depend on the latter.
2. Elaborate on Figure 1 panels A and D at the point where they are introduced, and bring the clarification currently deferred to Figure 13 forward.
3. Justify the method selection, which seems fairly small, and state whether more foundational XAI techniques such as LIME and SHAP, or even very basic methods like permutation feature effects, were considered and, if excluded, why.
4. Explain the train accuracy anomaly, in particular the gap between Smiling symmetric and Smiling asymmetric.

Finally, the in-text citation style is unusual, with citations rendered in parentheses where a textual form would be expected.

---

### Review · Reviewer_vadE · 2026-07-13

**Summary Of Contributions:**

**Strengths**

The paper addresses an important and practically relevant problem and brings together several previously separate lines of work on spurious correlations and Clever Hans behavior. It provides a broad empirical comparison of XAI-based and non-XAI-based correction methods under challenging conditions, including limited data, severe subgroup imbalance, and scarce minority examples. The paper is also relatively transparent about implementation choices and differences from the original studies.

**Weaknesses**

The main weakness is the lack of a sufficiently clear structure connecting the method taxonomy, experimental questions, result analysis, and conclusions. Although an initial categorization is introduced, the paper remains largely method-by-method, making the main cross-method takeaways difficult to extract and remember. Please see my detailed comments under "Requested Changes"

**Audience:**

Yes

**Audience Explanation:**

The paper concerns a very timely topic and a comparative analysis of several categories of method as done in this survey is highly valuable.

**Broader Impact Concerns:**

Broader impact statement is missing.

**Claims And Evidence:**

Yes

**Claims Explanation:**

I believe so but as I elaborated in the Requested Changes section, I would need to go through the paper again after the proposed restructuring to be more confident.

**Requested Changes:**

The paper addresses an important problem and contains a substantial amount of comparative experimental work. My main concern is not the absence of relevant material, but that the current organization makes it difficult to understand how the different parts fit together and what the main conclusions are. I therefore structure my comments around five dimensions: method categorization, experimental dimensions, dataset choices, result analysis, and conclusions.

### 1. Method categorization
Section 1 provides an initial categorization of the approaches and introduces the main methodological differences between the methods under comparison.

**What could be improved.**
This categorization should be developed into  more explicit taxonomy and then used consistently throughout the paper. The methods could, for example, be grouped into group-based robust optimization, representation-level concept removal, explanation-based regularization, and counterfactual knowledge-transfer approaches. The taxonomy should also compare the assumptions of the methods, including their need for group labels, concept annotations, counterfactual examples, retraining, and additional models. At present, the paper becomes predominantly method-by-method after the introduction, which makes the conceptual relationships between the approaches harder to follow.

### 2. Experimental dimensions
The experimental section evaluates several methods across multiple datasets and reports a wide range of method-specific results, implementation choices, and robustness observations.

**What could be improved.**
The section would benefit from being organized around a small number of explicit research questions rather than primarily around individual methods. Some examples are ( fee free to make your own):

* How effectively does each method reduce reliance on spurious features while preserving predictive performance?
* Which approaches are most reliable across datasets, confounder types, and imbalance settings?
* How do different intervention types compare?
* What are the practical costs in terms of retraining, tuning, annotations, and additional models?
* To what extent are the conclusions of the original papers reproduced?
* RQ concerning any new experimental settings for obtaining insights not present in the original paper

Organizing the results around such questions would make the experimental logic and the purpose of each analysis much clearer.

### 3. Dataset choices
The authors provide a broad rationale for the benchmark, including synthetic and real-world datasets, artificial and naturally occurring confounders, different confounder complexities, medical tasks, and severe minority-group underrepresentation.

**What could be improved.**
The dataset rationale should be made more systematic. The paper could first define the dimensions that the benchmark is intended to cover and then explain how each dataset contributes to that coverage. It should also state more explicitly whether each dataset was selected to reproduce an original experiment, enable a controlled cross-method comparison, or test a new setting. The scope of the conclusions should be qualified, since the experiments remain limited to image-based binary classification with a single confounder, and some datasets are closely related.

### 4. Result analysis
The paper reports many detailed observations and is  transparent about several differences between the present implementation and the setups used in the original studies.

**What could be improved.**
The analysis of the results could be reorganized around the research questions that you might want to introduce in the experimental section. Rather than discussing findings mainly method by method, each subsection should explicitly answer one research question using evidence across all relevant methods and datasets. For example, the paper could synthesize which methods most effectively reduce reliance on spurious features, which are most robust across datasets and confounder settings, how performance depends on supervision and resource requirements, and to what extent the original findings are reproduced and for what methods new insights are derived.

This would help distinguish broad comparative conclusions from isolated method-specific observations. It would also make it easier to understand which findings are general, which are dataset-dependent, and which may result from differences in supervision, model selection, hyperparameter tuning, or computational resources. A concise summary table mapping each research question to its main finding, supporting evidence, exceptions, and practical implications would further improve clarity.

Some implementation details, extended hyperparameter studies, secondary analyses, and additional per-dataset results could be moved to the appendix. The main text should focus on the central comparisons and findings.

### 5. Conclusions
The conclusion summarizes several observations from the experiments and discusses limitations.

**What could be improved.**
The conclusion would benefit from further internal structure. In particular, its subsections should correspond directly to the research questions posed in the experimental section. For each research question, the paper should state the main answer, the conditions under which it holds, important exceptions, and whether it agrees with the original literature.

Overall I hope the proposed alignment between method taxonomy, research questions, experimental results, and conclusion subsections would make the paper much easier to follow and remember. In its current form, I found it difficult to retain how observations in later sections connected to points made earlier. A substantial restructuring along these dimensions would therefore be helpful, and I would need to read the paper again carefully after such a revision to fully assess the strength and consistency of its conclusions.